# Distribution and diversity of classical deacylases in bacteria

Leonie G. Graf [1], Carlos Moreno-Yruela [2,6], Chuan Qin[1], Sabrina Schulze [1], Gottfried J. Palm [1], Ole Schmöker[1], Nancy Wang [3], Dianna M. Hocking [3], Leila Jebeli [3], Britta Girbardt[1], Leona Berndt[1], Babett Dörre[1], Daniel M. Weis[1], Markus Janetzky[1], Dirk Albrecht[4], Daniela Zühlke[4], Susanne Sievers [4], Richard A. Strugnell [3], Christian A. Olsen [2], Kay Hofmann [5] & Michael Lammers [1] ✉

Classical Zn$^{2+}$-dependent deac(et)ylases play fundamental regulatory roles in life and are well characterized in eukaryotes regarding their structures, substrates and physiological roles. In bacteria, however, classical deacylases are less well understood. We construct a Generalized Profile (GP) and identify thousands of uncharacterized classical deacylases in bacteria, which are grouped into five clusters. Systematic structural and functional characterization of representative enzymes from each cluster reveal high functional diversity, including polyamine deacylases and protein deacylases with various acyl-chain type preferences. These data are supported by multiple crystal structures of enzymes from different clusters. Through this extensive analysis, we define the structural requirements of substrate selectivity, and discovered bacterial de-D-/L-lactylases and long-chain deacylases. Importantly, bacterial deacylases are inhibited by archetypal HDAC inhibitors, as supported by co-crystal structures with the inhibitors SAHA and TSA, and setting the ground for drug repurposing strategies to fight bacterial infections. Thus, we provide a systematic structure-function analysis of classical deacylases in bacteria and reveal the basis of substrate specificity, acyl-chain preference and inhibition.

Acetylation of primary amines in proteins represents post-translational modifications known to act as sensors for the cellular metabolic state, and acetylation also occurs on small molecules such as polyamines[1–7]. In proteins, these amino groups are present as α-amino groups at the N-terminus or as ε-amino groups in lysine side chains. A common characteristic of these groups is that they are protonated and thus positively charged at physiological pH, and acetylation results in neutralization[8–11]. Next to protein acetylation it also occurs on the terminal primary amines of polyamines, i.e., the diamines cadaverine and putrescine, the triamine spermidine, and the tetraamine spermine[12–15]. These molecules are organic cations with a plethora of cellular functions in eukaryotes and prokaryotes: cell cycle progression, DNA packing, RNA stability, transcription and translation, biofilm formation, autophagy, and post-transcriptional regulation[16–42]. Polyamines are primordial molecules present at millimolar concentrations in prokaryotes and in eukaryotes playing important roles for cell

[1]Department Synthetic and Structural Biochemistry, Institute of Biochemistry, University of Greifswald, Greifswald, Germany. [2]Center for Biopharmaceuticals & Department of Drug Design and Pharmacology, Faculty of Health and Medical Sciences, University of Copenhagen, Copenhagen, Denmark. [3]Peter Doherty Institute for Infection and Immunity, Department of Microbiology and Immunology, The University of Melbourne, Melbourne, VIC, Australia. [4]Department of Microbial Physiology and Molecular Biology, Institute of Microbiology, University of Greifswald, Greifswald, Germany. [5]Institute for Genetics, University of Cologne, Cologne, Germany. [6]Present address: Institute of Chemical Sciences and Engineering (ISIC), School of Basic Sciences (SB), EPFL, Lausanne, Switzerland. ✉e-mail: michael.lammers@uni-greifswald.de

growth and cellular proliferation, i.e., spermidine is essential for cell viability in eukaryotes as it is needed for post-translational hypusination of the translation factor eIF5A[21,41–44].

The acetylation of primary amino groups on proteins or polyamines can be catalyzed enzymatically by acetyltransferases using acetyl-CoA as a donor molecule[45–54]. In bacteria, all acetyltransferases identified so far belong to the subfamily of Gcn5-related N-terminal acetyltransferases (GNATs)[55–60]. While acetylation of proteins regulates protein function using various mechanisms, acetylation of polyamines was shown to increase the metabolic flux of the polyamine biosynthetic pathway[5]. It is reported that acetylation of polyamines is essential to remove polyamines from cells and for interconversion of polyamines[61,62].

Next to this enzymatic acetylation of amino groups non-enzymatic acetylation was described to occur in eukaryotes and prokaryotes[63,64], depending on the intracellular concentration of acetyl-CoA and acetyl-phosphate, respectively[63,65–67]. Moreover, non-enzymatic acetylation of proteins also depends on the sequence context, the three-dimensional structure, i.e. the accessibility of the lysine side chain, and cellular conditions[7,58,66,68].

Lysine deac(et)ylases revert both enzymatic and non-enzymatic acetylation[7,69]. while most research has focused on eukaryotic deac(et)ylases, bacterial deacetylases are less well understood. In humans, eighteen deac(et)ylases can be distinguished based on their homology to *Saccharomyces cerevisiae* deacetylases[58,68,70]. Class I comprises the enzymes HDAC1–3 and 8 with homology to yeast transcriptional regulator RPD3. Class II enzymes show homology to yeast Hda1 and are subdivided into class IIa with HDAC4, 5, 7, and 9, and class IIb with HDAC6 and 10. Class IIa HDACs were shown to possess a low catalytic deacetylase activity, suggesting a role as a scaffolding protein rather than an enzyme due to a substitution of an active site tyrosine for histidine. Class III contains seven NAD$^+$-dependent sirtuin deacetylases (short: sirtuins; SIRTs), and class IV encompasses only a single enzyme, HDAC11, showing homologies toward class I and class II enzymes. The eleven enzymes of classes I, II, and IV are Zn$^{2+}$-dependent enzymes, sometimes referred to as the classical HDACs[70]. For mammals, it is reported that all class I HDACs except from HDAC8 are constituents of multi-protein complexes[71–78]. These complexes bind to sequence-specific DNA-transcription factors resulting in repression of transcription. Moreover, these complexes work in concert with other chromatin remodeling enzymes thereby also acting as epigenetic modulators[77]. Sirtuins are structurally and mechanistically, regarding the catalytic strategies used to achieve substrate deacylation, not related to classical HDACs[70,79,80]. Sirtuins and HDACs remove a range of diverse acylations, such as the aliphatic acylations: butyrylation, propionylation, lactylation, the charged acylations: malonylation, succinylation or glutarylation, and they can act as fatty acyl deacylases, capable to remove longer acyl-chains such as myristoyl groups or palmitoyl groups from lysine side chains, making it more appropriate to call the two deacetylase types sirtuins and classical HDACs in eukaryotes deacylases rather than deacetylases[76,81–98]. Moreover, classical HDACs and sirtuins have many non-histone substrates and even non-protein substrates such as carbohydrates, small molecules such as antibiotics, and polyamines[62,94,95,99–109]. So far, no deacetylase was discovered that acts as protein $N$-($\alpha$)-acetyl deacetylase, neither in eukaryotes nor in prokaryotes[110,111].

Classical Zn$^{2+}$-dependent HDACs were discovered in all domains of life[106,112–117]. Evolutionarily, this suggests that they constitute an ancient protein superfamily and their presence within their last common ancestor[106,115,116,118–120]. Structurally, these enzymes are composed of a central eight-stranded parallel β-sheet flanked by α-helices on each site, known as the α/β-arginase/deacetylase fold[113,121,122]. This shows homeostasis of L-arginine being of high importance during evolution[113,120]. From this precursor, classical Zn$^{2+}$-dependent protein lysine-, polyamine- and small molecule-deacetylases divergently developed during evolution[116]. The sporadic reports on bacterial

classical deacylases suggest that they can act as polyamine, small molecule, and/or protein deac(et)ylases[52,104,105,117,123–129]. The acetylpolyamine amidohydrolase from *Mycoplana ramosa* (*Mr*ApaH) deacetylates $N^8$-acetylspermidine[62,94,95,104,117], and the corresponding enzymes from *Pseudomonas aeruginosa* (*Pa*ApaHs) PA0321 and PA1409 deacetylate $N^1$-acetylputrescine and $N^1$-acetylcadaverine[104,105,117,124]. Based on sequence and structure, these enzymes relate to the eukaryotic polyamine deacetylase HDAC10 and the class II of mammalian classical deacylases[94,117]. The intracellular concentrations of polyamines were reported to reside in the millimolar range in prokaryotes and eukaryotes, however, the presence of de novo synthesized spermine is not well established in bacteria[130–136]. Few reports describe classical Zn$^{2+}$-dependent lysine deacetylases in bacteria, belonging to either the class I: ApaH (histone-deacetylase-like protein) from *Aquifex aeolicus* and AcuC (acetoin-utilization protein C) from the Gram-positive bacteria *Bacillus subtilis* and *Staphylococcus aureus*; or to class II: AcuC from *Aeromonas hydrophila*, Kdac1 of the multidrug-resistant pathogen *Acinetobacter baumannii*, *P. aeruginosa* PA3774, and *Alcaligenes/Bordetella* FB188 HdaH (histone-deacetylase-like amidohydrolase)[105,125–129,137]. So far, no bacterial enzyme belonging to class IV has been described. Structurally, it was shown that the presence of a loop-insertion in the N-terminal region, the so-called L1-loop, in the lysine deacetylases *P. aeruginosa* PA3774 and *Alcaligenes/Bordetella* HdaH mediates oligomerization contributing to the determination of substrate specificity[105,128,129]. Polyamine-specific deacetylases and *M. ramosa* ApaH have a loop-insert, the polyamine-specificity loop (PSL)/L2-loop, that drives dimer formation and contributes to substrate specificity toward acetylated-polyamines[117,124]. For the Gram-negative bacterial species *A. hydrophila*, it was shown that AcuC is needed for biofilm formation and for virulence suggesting that targeting classical deacetylases in bacteria might be a strategy to fight bacterial pathogens[125]. In addition, ApaH from *A. aeolicus* and HdaH from *Alcaligenes/Bordetella* are inhibited potently by mammalian HDAC inhibitors[129,138], which highlights their druggability. Recently, the *Legionella pneumophila* enzyme LphD (*Legionella pneumophila* deacetylase) was shown to be a para-effector secreted into host cells[139,140]. This is related to the enzyme Smh1 reported earlier[140].

Here, we show a comprehensive study on bacterial classical deacylases (DACs). We classify bacterial classical DACs into different clusters based on their amino acid sequences and provide extensive structural and functional data for molecular determinants of substrate specificity. We show that classical DACs are widely distributed across bacteria and that they display diverse activities including lysine delactylation and long-chain deacylation. We further report inhibition by established mammalian HDAC inhibitors, which sets the ground for future development of therapeutics.

## Results
### Bacteria encode a plethora of classical deacylases
Based on some reports describing the presence of classical deacetylases in Gram-positive and Gram-negative bacteria, we performed bioinformatics analyses to search for additional bacterial strains encoding classical lysine deac(et)ylases. A Generalized Profile (GP) was constructed from a multiple sequence alignment of known and validated classical deacetylases, and this was used for screen the UniProt database[141,142]. From the initial set of about 61,075 hits, among them about 37,989 bacterial sequences, 2557 hits from archaea, and 20,529 hits from eukaryotes, a reduced set of about 5973 representative members was selected by the 'cd-hit' program, by removing duplicates and highly similar (>60% identity) sequences (Supplementary Data 1–3)[143]. These sequences were used as input for clustering using the program clans (cluster analysis of sequences)[144], which performs all-against-all BLAST searches of unaligned sequences and clusters them by their similarity. In the output map, each sequence is represented as a dot, arranged on a two-dimensional plane so that their

a

b

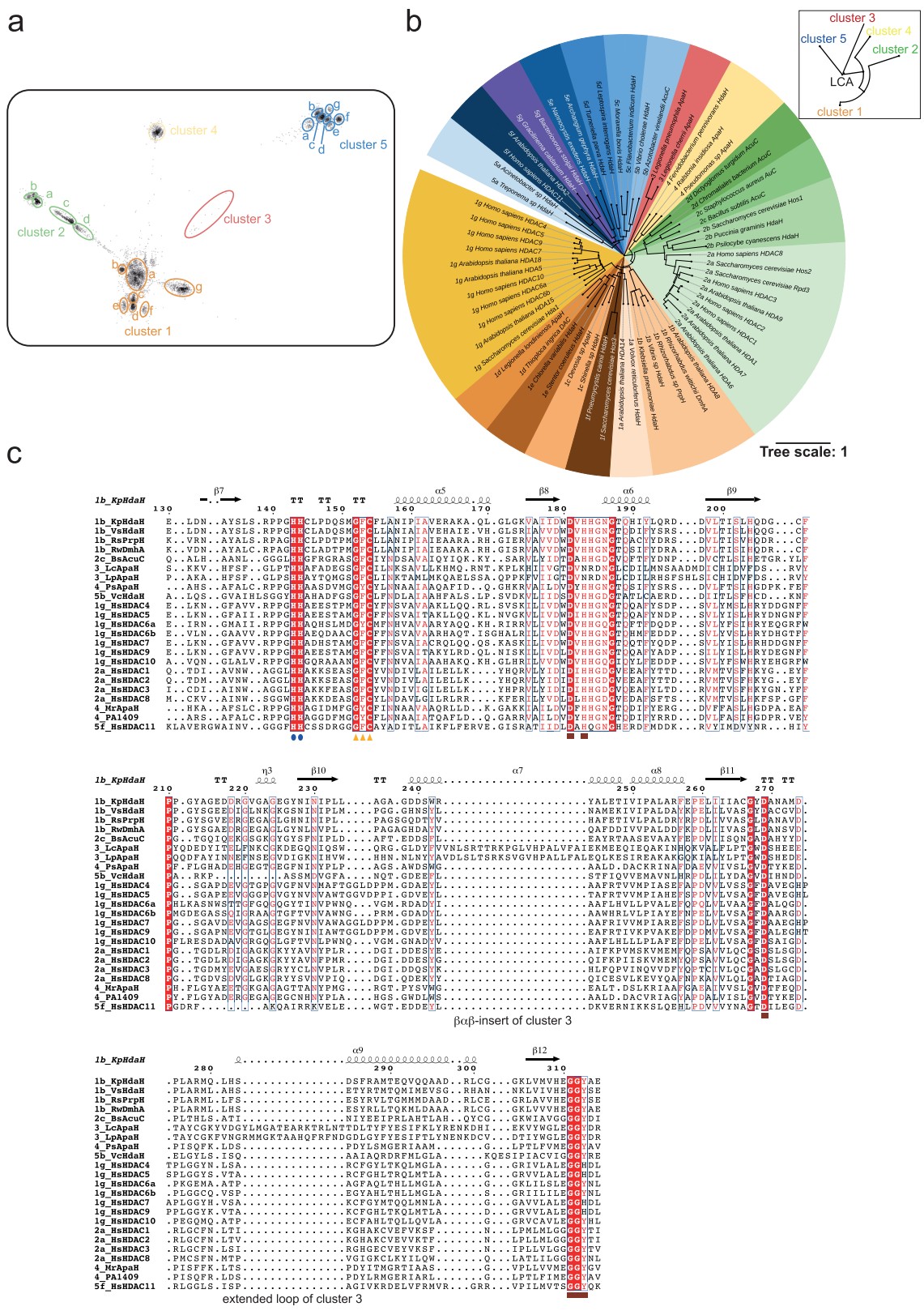

c

2D-distances approximately correspond to the sequence similarities (Fig. 1a). This resulted in a total of five major clusters, and clusters 1, 2, and 5 were further split into several sub-clusters due to their multi-lobe appearance. To visualize how these prokaryotic sequences cluster with known classical HDACs, the sequences from humans, *Saccharomyces cerevisiae*, and *Arabidopsis thaliana* were added prior to clustering. This analysis revealed that class I HDACs (human: HDAC1,2,3,8; yeast

Hos2, Rpd3; *A. thaliana*: HDA1,6,7,9) cluster in sub-cluster 2a, class II HDACs (human: HDAC4,5,7,9 (class IIa); HDAC6,10 (class IIb); yeast: Hda1; *A. thaliana*: HDA5,15) cluster in sub-cluster 1g, *A. thaliana* HDA14 is found in sub-cluster 1a, and the class IV (human: HDAC11; *A. thaliana*: HDA2) clusters in sub-cluster 5f. The clusters 3 and 4 do not contain any mammalian HDACs, suggesting these enzymes are structurally and/or functionally different. Within the highly-populated cluster 1, the

**Fig. 1 | Bacteria encode a plethora of Zn²⁺-dependent deacylases. a** Bacterial deacylases can be classified into five clusters, some with several sub-clusters. A Generalized Profile (GP) was constructed from a multiple sequence alignment (MSA) of classical deacetylases, which resulted in thousands of sequences upon screening the UniProt database. Clustering was done using the program clans (cluster analysis of sequences). Each sequence is represented as a dot on a two-dimensional plane, i.e. their 2D-distances correspond to sequence similarities. This resulted in a total of five major clusters (clusters 1–5). The clusters 1, 2, and 5 are subdivided into several sub-clusters, i.e., 1a–1g, 2a–2d, and 5a–5g. **b** Phylogenetic tree of classical Zn²⁺-dependent deacylases of selected bacterial deacylases representing all clusters. The human enzymes, the deacylases of *S. cerevisiae* and the classical deacylases from *A. thaliana* are highlighted. All human enzymes are categorized in cluster 1 (HDAC class IIa and IIb), cluster 2 (HDACs class I), and cluster 5 (HDAC11). The closeup shows the development of the clusters from the LCA (last common ancestor). The unrooted phylogenetic tree was created with iTOL using a multiple sequence alignment of the catalytic domains (deleted >90% of gaps) created by MAFFT. **c** Amino acid sequence alignment of selected classical Zn²⁺-dependent deacylases. The catalytic residues are totally conserved in enzymes from *Homo sapiens* and from bacteria. Shown are representative human enzymes of each class and bacterial enzymes representing each cluster (1–5) and the enzymes PA1409 and *Mr*ApaH characterized earlier[117,124]. The numbering and the secondary structure elements were shown for *Kp*HdaH (1b) above the alignment. Blue circles: double-His motif, with the second His acting as catalytic base/acid (*Kp*HdaH: His143-His144); yellow triangles: conserved GFC-motif lining the substrate binding channel; brown squares: Asp-His-Asp for coordination of the catalytic Zn²⁺-ion; brown rectangle: (E/S/G)GGY-motif lining the foot pocket for substrate release and the catalytic Tyr (*Kp*HdaH: Tyr313) important for orientation/polarization of the acetyl-group and for stabilization of the negative charged oxygen arising in the tetrahedral intermediate. The MSA was conducted with the T-Coffee algorithm and ESPript version 3.0 was used to create the figure[212,213,218,219].

sub-cluster 1a encompasses annotated bacterial HdaH (histone-deacetylase amidohydrolases) enzymes from a phylogenetically wide range of bacteria, while members of 1b are mainly from *Proteobacteria* and *Actinobacteria*. Sub-clusters 1c and 1d encompass mainly annotated ApaH (acetylpolyamine aminohydrolase) from α- and γ-*Proteobacteria*, respectively. Members of sub-clusters 1e–1g are HdaH enzymes from various eukaryotes, mostly from *Stramenopila*, *Alveolata*, and *Rhizaria* (SAR)-species (1e), fungi (1f) and animals/fungi/plants (1g). In cluster 2, sub-cluster 2a mainly comprises eukaryotic class I HDACs from animals, plants, and fungi, while sub-cluster 2b is mostly fungal-specific. The more divergent sub-cluster 2c encompasses the bacterial AcuC (acetoin-utilization proteins) of both Gram-positive and -negative bacterial taxa, while sub-cluster 2d groups additional AcuC-annotated proteins of γ-*Proteobacteria* with non-annotated archaeal proteins. The sparsely populated cluster 3 is found between clusters 1 and 5 and comprises enzymes from γ-*Proteobacteria*, mostly of the genus *Legionella*. Cluster 4 is formed by annotated ApaH-like (acetylpolyamine ami(n/d)ohydrolase-like) enzymes from all bacterial taxa. Finally, cluster 5 and its sub-clusters 5a–5g comprise a heterogenous group of bacterial deacylases from mainly α-, γ-*Proteobacteria*, *Chloroflexi*, *Spirochetes*, and *Bacteroides*. This clustering was also supported by phylogenetic analyses showing how the clusters were evolutionarily related (Fig. 1b). From the last common ancestor (LCA) of the ami(d/n)ohydrolases the cluster 5 enzymes evolved in one branch to form sub-clusters 5a–5f, and another branch split further into the branch of cluster 3 enzymes and the branch from which on the one hand cluster 4 enzymes and on the other hand cluster 1 (sub-clusters 1a–1g) and cluster 2 (sub-clusters 2a–2d) enzymes evolved (Fig. 1b; Supplementary Data 4, 5). Next, we selected representative enzymes to unravel whether these UniProt annotations reflect their activities, substrate specificities, and acyl-chain preferences.

## Bacterial DACs from oligomers and have conserved active site

We selected bacterial species encoding bacterial deacylases representing clusters 1–5 for subsequent functional and structural analyses. If possible, the criterion for the selection was the bacterial species encoding for the enzymes either being a human pathogen, potentially enabling therapeutically approaching the enzymes in a drug-repurposing strategy or being a bacterial model organism. To this end, we selected from cluster 1 *Vibrio* sp. HdaH (*Vs*HdaH (1b)), *Klebsiella pneumoniae* HdaH (*Kp*HdaH (1b)), DmhA (dimethoate hydrolase) of *Rhizorhabdus wittichii* (*Rw*DmhA (1b)) and PrpH (propanil hydrolase) of *Rhizorhabdus* sp. (*Rs*PrpH (1b)). From cluster 2, we selected AcuC of *B. subtilis* (*Bs*AcuC (2c)), and from cluster 3 we selected the *Legionella cherrii* and *Legionella pneumophila* acetyl-polyamine aminohydrolases (*Lc*ApaH (3); *Lp*ApaH (3)). Notably, the *L. pneumophila* enzyme was recently characterized as LphD, secreted into host cells during infection as a virulence factor acting as histone deacetylase[139,140]. The *Pseudomonas* sp. acetylpolyamine amidohydrolase ApaH (*Ps*ApaH (4)) was selected as the representative enzyme for cluster 4 and for cluster 5 we selected the deacylase of *Vibrio cholerae* HdaH (*Vc*HdaH (5b)). An amino acid sequence alignment and sequence logo representation of all selected enzymes showed that all essential active site residues are conserved, which points to catalytically active enzymes (Fig. 1c; Supplementary Fig. 1).

We established expression and purification strategies for these bacterial deac(et)ylases and could obtain pure enzymes in yields sufficient to perform further biochemical and structural analyses (Supplementary Fig. 2a). To characterize the enzymes' oligomeric states, we performed analytical size-exclusion chromatography (SEC) experiments. These data revealed that the deac(et)ylases of cluster 1, *Vs*HdaH (1b) and *Kp*HdaH (1b), *Rw*DmhA (1b), and *Rs*PrpH (1b), elute as an apparent trimer/tetramer (Supplementary Fig. 2b). Cluster 2 enzyme *Bs*AcuC (2c), cluster 3 enzymes *Lc*ApaH (3) *and Lp*ApaH (3), and the cluster 5 enzyme *Vc*HdaH (5b) elute as monomers, and cluster 4 enzyme *Ps*ApaH (4) elutes as apparent dimer from the analytical SEC column (Supplementary Fig. 2b). We suggest the findings regarding the oligomeric state observed for the representative enzymes of sub-clusters to be generalizable to the whole cluster, as structural alignments of AlphaFold2 models show high degree of structural similarity between members of different sub-clusters (Supplementary Data 6; Supplementary Fig. 3). With these representative enzymes in hand, we next analyzed their activity in vitro.

## Bacterial DACs act as lysine deacetylases

To understand substrate specificity of the bacterial deac(et)ylases, we performed a lysine deacetylation screening using Fluor-de-Lys assays (Fig. 2). We also analyzed catalytically inactive enzymes obtained by mutation of an active site catalytic His residue, as control (Fig. 2). Using a Boc-Lys(Ac)-AMC substrate, we identified the enzymes of cluster 3, *Lc*ApaH (3) and *Lp*ApaH (3), showing the highest activity amongst compared bacterial enzymes (Fig. 2a; Supplementary Fig. 4a). The class IIb human enzyme HDAC6 (*Hs*HDAC6 (1g)), representing cluster 1, was used for comparison as it was shown to efficiently deacetylate Boc-Lys(Ac)-AMC[145]. Moreover, the cluster 2 enzyme *Bs*AcuC (2c) showed deacetylase activity toward Boc-Lys(Ac)-AMC. Interestingly, except for the minor activity of *Vs*HdaH (1b), the cluster 1 enzymes [*Kp*HdaH (1b), *Rw*DmhA (1b), and *Rs*PrpH (1b)] were seemingly inactive compared to *Hs*HDAC6 (1g) (Fig. 2a). This result suggests that substrate selectivity of individual enzymes varies within the clusters.

We next performed Fluor-de-Lys assays with two commercially available peptides: peptide 2 and peptide 2a. Peptide 2 is a substrate for class I (*Hs*HDAC1,2,3,8) and class IIb (*Hs*HDAC6,10 (1g)) HDACs, and therefore we used *Hs*HDAC1 (2a) and *Hs*HDAC6 (1g) as references (Fig. 2b). To further validate the assay, we used the class IIa enzymes *Hs*HDAC7 (1g) and *Hs*HDAC9 (1g) as negative controls (Fig. 2b). Notably, we discovered activity of *B. subtilis* AcuC (2c), supporting that this

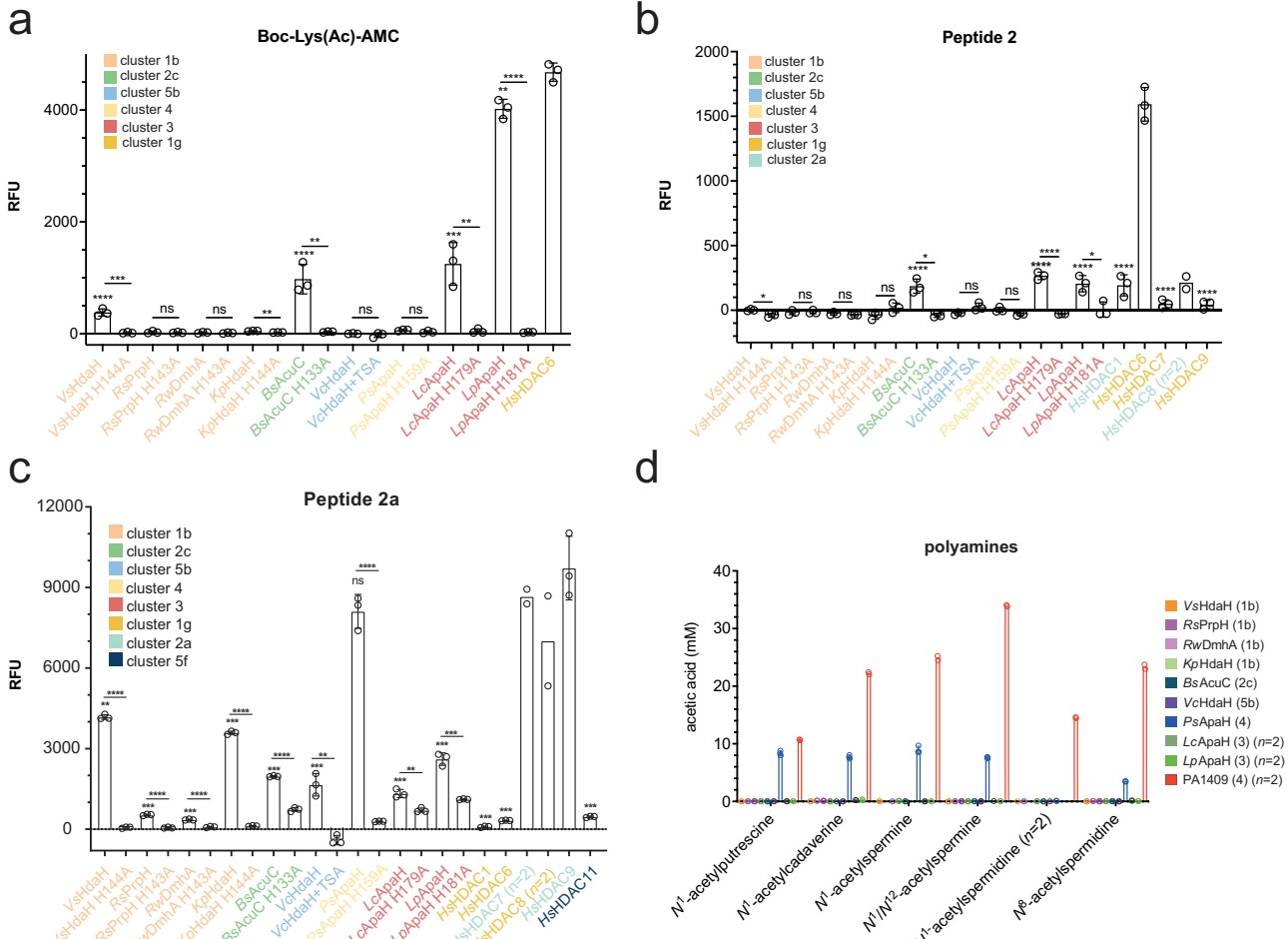

**Fig. 2 | Pre-screening of bacterial deacylases for lysine deacylase and polyamine deacylase activity.** **a** Activity of bacterial deacylases toward Boc-Lys(Ac)-AMC. *Hs*HDAC6 (1g) was used as reference. The enzymes *Vs*HdaH (1b), *Bs*AcuC (2c) and *Lc*ApaH (3) and *Lp*ApaH (3) are active in deacetylating Boc-Lys(Ac)-AMC. The catalytic inactive mutants were analyzed as controls. The experiments were performed in three replicates. Bars depict means ± standard deviation (SD). Significance was tested by unpaired, two-tailed t-tests (p < 0.05) (n = 3) either to the catalytically inactive mutant or to *Hs*HDAC6. Exact values can be found in the Source Data. Source data are provided as Source Data file. **b** Fluor-de-Lys Peptide 2 is deacetylated by *Bs*AcuC (2c), by *Lc*ApaH (3) and by *Lp*ApaH (3). As references, we show HDAC1 (2a; class I), HDAC8 (2a; class I), and HDAC6 (1 g; class IIb) are competent to deacetylate peptide 2. The inactivity of the catalytic inactive mutants confirms an enzymatic reaction as indicated. The experiments were performed in three replicates. Bars depict means ± SD. Significance was tested by unpaired, two-tailed t-tests (p < 0.05) (n = 3) either to the catalytically inactive mutant or to *Hs*HDAC6. Exact values can be found in the Source Data. Source data are provided

as Source Data file. **c** Fluor-de-Lys reporter peptide 2a is deacetylated by various bacterial enzymes. We observed the strongest activity for the cluster 1b enzymes *Kp*HdaH (1b) and *Vs*HdaH (1b) and for *Ps*ApaH (4) of cluster 4. Moderate activity is also observed for *Bs*AcuC (2c) and for the enzymes *Lp*ApaH (3) and *Lc*ApaH (3). The catalytic inactive mutants confirm the enzymatic reactions. The experiments were performed in three replicates (n = 3), except for human HDAC6 (n = 1), HDAC7 (n = 2), and HDAC8 (n = 2). Bars depict means ± SD. Significance was tested by t-tests (p < 0.05) to catalytically inactive mutant or to *Hs*HDAC9. Exact values can be found in the Source Data. Source data are provided as Source Data file. **d** Cluster 4 contains active polyamine deacetylases. *Ps*ApaH (4) is active in deacetylating the polyamines $N^1$-acetylputrescine, $N^1$-acetylcadaverine, $N^1$-acetylspermine, $N^1,N^{12}$-diacetylspermine with similar efficiency. *Ps*ApaH (4) weakly deacetylates $N^8$-acetylspermidine. The catalytic inactive mutants confirm an enzymatic reaction. As control, we used the *P. aeruginosa* enzyme PA1409. The experiments were performed in three replicates (n = 3). Bars depict means ± SD. Source data are provided as Source Data file.

is a class I enzyme. We also observed activity for *Lc*ApaH (3) and *Lp*ApaH (3) of cluster 3, supporting the separation of cluster 3 and cluster 1 enzymes in individual clusters (Fig. 1a).

Peptide 2a is preferentially deacylated by class IIa (*Hs*HDAC4,5,7,9 (1g)) HDACs and by the class I enzyme *Hs*HDAC8 (2a). Accordingly, we observed strong activity for *Hs*HDAC7 (1g) and *Hs*HDAC9 (1g) but neither for the class I enzyme *Hs*HDAC1 (2a) nor the class IIb enzyme *Hs*HDAC6 (1g), supporting the validity of the assay and of the clustering (Figs. 1a, b, 2c; Supplementary Data 1–3). In this assay, we observed strongest activity for the cluster 1b HdaH enzymes from *Vibrio* sp. and *K. pneumoniae*, as well as the cluster 4 enzyme *Ps*ApaH (Fig. 2c). Residual activity was also observed for the cluster 2c enzyme *Bs*AcuC, the cluster 5b enzyme *Vc*HdaH and the cluster 3 *Legionella* ApaH enzymes. These results suggest that commercial peptide

substrates allow the classification of bacterial deacetylases (Fig. 2b, c). The fact that observing activity toward peptide 2a only for the sub-cluster 1b enzymes *Vs*HdaH (1b) and *Kp*HdaH (1b) but not for the sub-cluster 1b enzymes *Rw*DmhA and *Rs*PrpH supports the notion that the latter two enzymes evolved toward activity as dimethoate hydrolase and propanil hydrolase, respectively[146,147]. We also observed a low activity of *Bs*AcuC (2c) toward peptide 2a. These results suggest that further mechanisms exist at the molecular level to determine substrate specificity such as the three-dimensional structure, the amino acid sequence of the substrate, and/or the acyl-chain.

## Cluster 4 contains polyamine deacetylases
The mammalian enzyme *Hs*HDAC10 (1g) is a polyamine deacetylase rather than a protein deacetylase, with substrate preference for $N^8$-

acetylspermidine over $N^1$-acetylcadaverine and $N^1$-acetylputrescine[94]. In *P. aeruginosa*, genes encoding the polyamine deacetylases (e.g. PA1409) are upregulated by the exogenous supply of acetylputrescine and agmatine but not putrescine[148], and their knockout slows down growth on acetylcadaverine or acetylputrescine as carbon sources[124]. To evaluate whether any of the expressed bacterial DACs are polyamine deacetylases, we analyzed their activity against $N^1$-acetylputrescine, $N^1$-acetylcadaverine, $N^1$-acetylspermidine, $N^8$-acetylspermidine, $N^1$-acetylspermine and $N^1,N^{12}$-diacetylspermine (Fig. 2d), with *P. aeruginosa* PA1409 as positive control[124]. Only the cluster 4 enzyme ApaH from *Pseudomonas* sp., *Ps*ApaH (4), showed activity as polyamine deacetylase (Fig. 2d; Supplementary Fig. 4b, c). While the *P. aeruginosa* enzyme PA1409 showed a general increase in activity with increasing polyamine chain length from acetylputrescine to acetylspermine, it showed slightly less efficiency for $N^1$- and $N^8$-acetylspermidine (10-atom backbone) compared to $N^1$-acetylcadaverine (7-atom backbone). In contrast, *Ps*ApaH (4) deacetylated $N^1$-acetylputrescine, $N^1$-acetylcadaverine and $N^1$-spermine/$N^1,N^{12}$-spermine with similar efficiency, while only marginally deacetylating $N^8$-acetylspermidine (Fig. 2d). For *Ps*ApaH (4), we performed Michaelis–Menten kinetics for the deacetylation of $N^1$-acetylputrescine (Supplementary Fig. 4c). This revealed a $K_M$ value of 0.96 mM and a $k_{cat}$ of 4.12 s$^{-1}$ (Supplementary Fig. 4c), which is in the same range as reported for PA1409 ($K_M$: 0.5 mM)[124]. While it was reported that *P. aeruginosa* enzymes PA1409 and PA0321 also deacetylate Boc-Lys(Ac)-AMC[124], our data on *Ps*ApaH (4) shows that it is only capable to deacylate peptide 2a but not Boc-Lys(Ac)-AMC (Fig. 2a, c). This suggests *Ps*ApaH (4) has a dual function acting as lysine deacetylase and polyamine deacetylase (Fig. 2c, d). Although the cluster 3 enzymes *Lc*ApaH and *Lp*ApaH were annotated as acetylpolyamine deacetylases (ApaHs), our analyses suggest that bacterial polyamine deacetylases are exclusively found in cluster 4 (Fig. 2d). Notably, *Hs*HDAC10 (1g) is present in cluster 1, suggesting that its activity has evolved independently following the separation of the branches containing cluster 4 and cluster 1/2 enzymes (Fig. 1b).

As a summary, our pre-screening data suggest the substrate preference of the protein deac(et)ylases depends on the substrates' amino acid sequence. Moreover, it is known that some enzymes show activity toward different types of acyl-chains rather than acting as pure deacetylases. To analyze to which extent the amino acid sequence affects the substrate specificity of bacterial deac(et)ylases and to gain insight into their acyl-chain preferences, we next performed Fluor-de-Lys assays with various peptide sequences and acyl-chains.

## Bacterial DACs show different acyl-chain type preferences

To analyze the impact of the substrate sequence and the capacity of the bacterial enzymes to remove various acyl-chain types from lysine side chains, we performed additional Fluor-de-Lys assay-based screenings. To this end, we used tri- or tetrapeptide sequences derived from histone H3 (APRK$_{acyl}$, H3$_{15-18}$ or TARK$_{acyl}$, H3$_{6-9}$), histone H4 (LGK$_{acyl}$, H4$_{10-12}$), tumor suppressor protein p53 (QPKK$_{acyl}$, p53$_{317-320}$) and DLAT [(dihydrolipoyllysine-residue acetyltransferase) component of pyruvate dehydrogenase complex; ETDK$_{acyl}$; DLAT$_{256-259}$], containing various acyl-chain types on the C-terminal lysine side chain, as substrates (Fig. 3a; Supplementary Fig. 5,6).

Our analyses revealed identification of substrates for all selected bacterial deacylases (Fig. 3a; Supplementary Fig. 6). The cluster 1 enzymes *Kp*HdaH (1b) and *Vs*HdaH (1b) showed a robust deacetylase activity toward the H3-derived peptide APRK$_{ac}$ and the p53-peptide QPKK$_{ac}$, i.e. both favor a positively charged residue at the −1 position (Fig. 3a, b). While *Kp*HdaH (1b) did not tolerate a negatively charged residue at the −1 position, *Vs*HdaH (1b) was capable to weakly deacetylate the DLAT peptide ETDK$_{ac}$ (Fig. 3a, b; Supplementary Fig. 4). Both enzymes showed an activity as demyristoylase on peptide TARK$_{myr}$. Interestingly, *Vs*HdaH (1b) was also active as de-L-lactylase as shown

using the p53-derived peptide QPKK$_{lac}$, as substrate (Fig. 3a, c; Supplementary Figs. 6 and 7). Of all the tested bacterial enzymes, *Vs*HdaH (1b) had in fact the highest de-L-lactylase activity, albeit being less efficient than *Hs*HDAC3 (2a) (Fig. 3a; Supplementary Figs. 5–7). Interestingly, the mammalian class I enzymes *Hs*HDAC1-3 (2a) are the main delactylases[98], while the bacterial enzyme *Vs*HdaH (1b) belongs to class II, suggesting that the delactylase activity is not restricted to class I in bacteria. For the cluster 1b enzyme *Rs*PrpH (1b) the deacylase activity was almost nondetectable, only for the H3-derived peptide APRK$_{ac}$ we observed a weak deacetylase activity. This supports the data obtained with the Boc-Lys(Ac)-AMC and the peptide 2 and 2a substrates (Fig. 2a–c; Fig. 3a). *Rs*PrpH (1b) was isolated from wastewater showing activity as propanil hydrolase for which Zn$^{2+}$ has inhibitory potential[147]. While not showing any lysine deacetylase activity in the previous assays, *Rw*DmhA showed activity as deacetylase toward the peptides H3 APRK$_{ac}$ and the p53-peptide QPKK$_{ac}$ although much less efficient compared to the cluster 1b enzymes *Kp*HdaH (1b) and *Vs*HdaH (1b) (Fig. 3a). For *Rs*PrpH (1b) and *Rw*DmhA (1b) the data suggest that these enzymes represent examples for divergent evolution developing toward an activity to convert specific substrates, i.e. propanil or dimethoate, while losing their efficiency to deacetylate lysine side chains, supporting that enzymes with different activities might be present within a cluster (Fig. 3a).

The enzyme *Bs*AcuC (2c) is known to be an efficient deacetylase for AMP-forming acetyl-CoA-synthetase *Bs*AcsA[126,149,150]. Our pre-screening suggested *Bs*AcuC (2c) being a rather promiscuous enzyme capable of removing a range of different acyl-chain types from lysine side chains; albeit, with relatively low efficiency (Fig. 3a, d). In agreement, *Bs*AcsA was shown to be modified by several acylations apart from acetylation, such as propionylation[151]. However, we found the peptide QPKK$_{oct}$ being most efficiently hydrolyzed by *Bs*AcuC (2c) compared to all other peptides tested including the the LGK$_{acyl}$ peptides (Fig. 3a). Moreover, we show *Bs*AcuC (2c) being the only bacterial enzyme able to remove an unsaturated acylation, acting as a decrotonylase (Fig. 3a, d). The high degree of acyl-chain type promiscuity on the one hand but low deacylation efficiency, on the other hand, might indicate *Bs*AcuC (2c) being able to remove a range of different acyl-chain types but being selected during evolution to have a very narrow substrate range, in fact only having members of the ANL (acyl/aryl-CoA-synthetases/ligases, the adenylation domains of non-ribosomal peptide synthetases (NRPSs) and firefly-luciferases)-family as substrates[126,152]. Members of the ANL family such as AMP-forming acetyl-CoA-synthetase, are acylated on a lysine present in a consensus sequence containing a highly conserved Gly-Lys dipeptide. Our data show *Bs*AcuC (2c) is capable of deacylated Gly-Lys-containing peptides with similar efficiency to peptides containing a positively charged residue at −1 position (Fig. 3a, d). Notably, the deacylation efficiency for the physiological substrate *Bs*AcsA might be substantially higher as the peptides analyzed here differ in their substrate sequences[150]. Furthermore, the three-dimensional structure might affect the deacylation efficiency.

Cluster 3 contains several enzymes from the genus *Legionella*. The enzymes *Lc*ApaH (3) and *Lp*ApaH (3) are closely related to the deacetylases LphD and Smh1 reported to be para-effectors released into the host cells by the type IV secretion system[139,140]. The reported data suggest a relation between Smh1 and class I/class II enzymes of mammalian HDACs. According to the classification presented here, class I enzymes belonging to cluster 2 and class II enzymes to cluster 1, we suggest to classify the *Legionella* ApaH enzymes into separate cluster 3. Notably, *Lc*ApaH (3) and *Lp*ApaH (3) are able to efficiently deacetylate several H3, H4, DLAT and p53-derived peptides either containing a positively charged, a negatively charged or a glycine residue at −1 position, showing a low degree of sequence specificity, which indicates that these enzymes are capable of deacylating substrates other than the previously proposed H3K14 (Supplementary

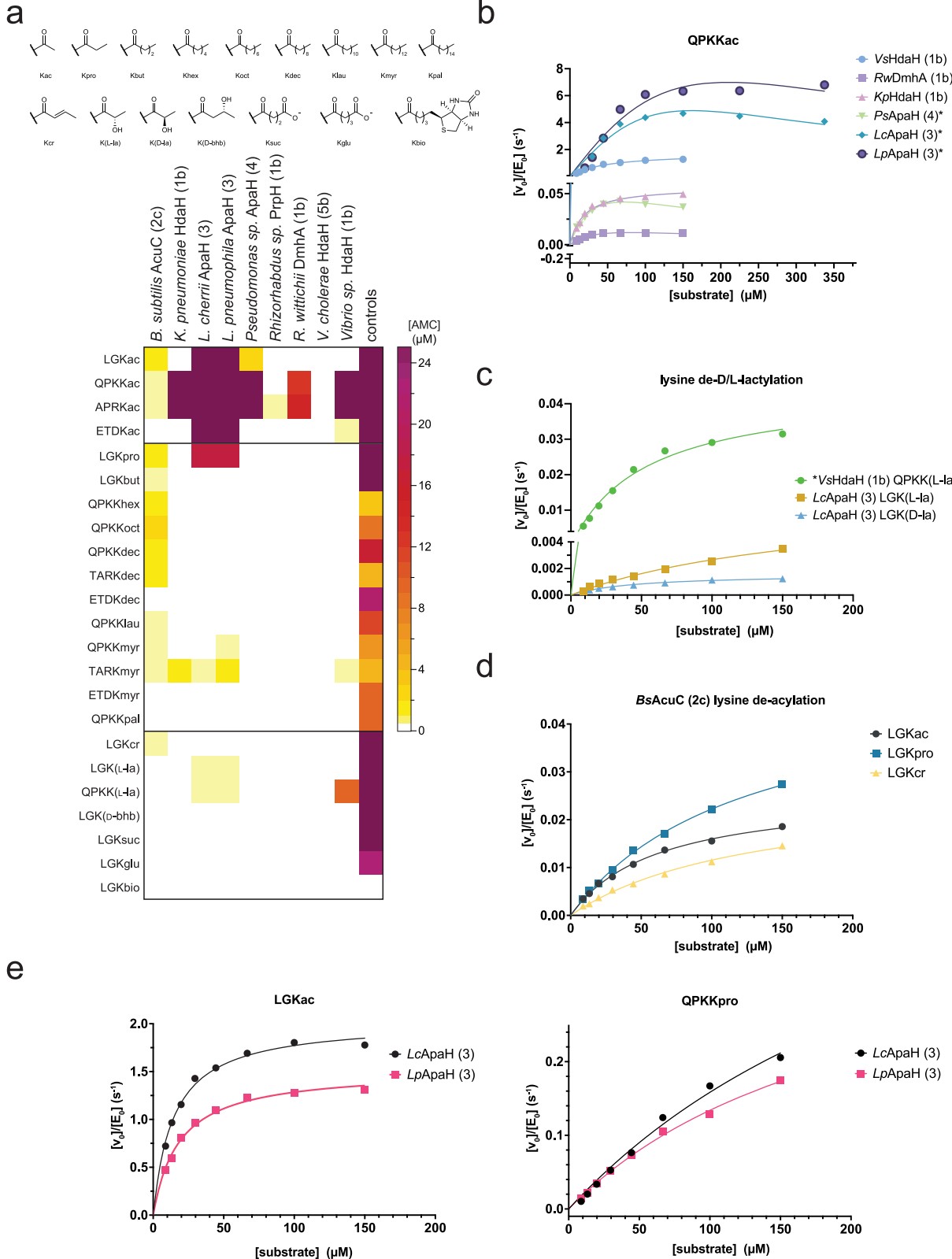

Figs. 6 and 7)[139]. We observed *Lc*ApaH (3) and *Lp*ApaH (3) act as lysine depropionylases and as lysine demyristoylases for the peptides containing a positively charged residue at the −1 position (Fig. 3a, e; Supplementary Fig. 7). Moreover, of all the compared enzymes, *Lc*ApaH (3) and *Lp*ApaH (3) were capable of removing ʟ- and ᴅ-lactylation from lysines (Kla) in both tested peptides, the H4 peptide LGK$_{lac}$ and the p53-peptide QPKK$_{lac}$ (Fig. 3a, c; Supplementary Fig. 7). Lactylation

was recently discovered as a modification important for metabolic reprogramming in both eukaryotes and prokaryotes[153–158].

Our data suggest the enzyme *Ps*ApaH (4) acts exclusively as deacetylase with a preference for a positively charged residue at the −1 position (Fig. 3a). Neither the DLAT peptide ETDK$_{ac}$ with a negatively charged residue at the −1 position nor the histone H4-derived peptide LGK$_{acyl}$ were efficiently deacetylated by *Ps*ApaH (4) (Fig. 3a). Moreover,

**Fig. 3 | Pre-screening to uncover acyl-chain preferences of bacterial deacylases.**
**a** Pre-screening of selected bacterial deacylases to assess their preferences for different acyl-chain types. A Fluor-de-Lys assay-based screening was performed using histone H3 (APRK$_{acyl}$, H3$_{15-18}$ or TARK$_{acyl}$, H3$_{6-9}$), histone H4 (LGK$_{acyl}$, H4$_{10-12}$), p53 (QPKK$_{acyl}$, p53$_{317-320}$) and DLAT derived peptide sequences. Depicted is the conversion of peptide in released [AMC] in μM. Positive controls as described in the "Methods" section. The graph depicts the means of both recorded independent replicates ($n = 2$). Source data are provided as Source Data file. **b** Michaelis–Menten kinetics for bacterial deacetylases (discontinuous assay). All selected bacterial deacylases showing robust deacetylase activity were summarized (QPKK$_{ac}$, p53$_{317-320}$). Notably, for $Rw$DmhA (1b), $Ps$ApaH (4), and $Lc$ApaH (3)/$Lp$ApaH (3) we observed substrate inhibition at higher substrate concentration. *: Data adjusted to kinetics with substrate inhibition at high concentration. The experiments were performed in two independent replicates ($n = 2$). Data are presented as means. Source data are provided as Source Data file. **c** Michaelis–Menten kinetics for the delactlases $Vs$HdaH (1b) and $Lc$ApaH (3). Notably, for $Vs$HdaH (1b) we observed a

stereoselectivity toward de-L-lactylation (QPKK$_{L-la}$, p53$_{317-320}$), while the $Lc$ApaH (3) converts both stereoisomers acting as de-D/L-lactylase (LGK$_{D-la/L-la}$, H4$_{10-12}$). The experiment was done in continuous assay format for $Vs$HdaH (1b) and $Lc$ApaH (3). The * indicates the discontinuous assay format used for $Vs$HdaH (1b). The experiments were performed in two independent replicates ($n = 2$). Data are presented as means. Source data are provided as Source Data file. **d** Michaelis–Menten kinetics for the deacylase activity for $Bs$AcuC (2c) (continuous assay). $Bs$AcuC (2c) is active as deacetylase, depropionylase, and decrotonylase (LGK$_{acyl}$, H4$_{10-12}$). The experiments were performed in two independent replicates ($n = 2$) in a continuous assay format. Data are presented as means. Source data are provided as Source Data file. **e** Michaelis–Menten kinetics for the deacylase and depropionylase activity of $Lp$ApaH (3) and $Lc$ApaH (3) (deacylase: continuous assay format; depropionylase: discontinuous assay format). $Lp$ApaH (3) and $Lc$ApaH (3) are active deacylases (LGK$_{ac}$, H4$_{10-12}$) and depropionylases (QPKK$_{acyl}$, p53$_{317-320}$). The experiments were performed in two independent replicates ($n = 2$). Data are presented as means. Source data are provided as Source Data file.

no longer acyl-chain was found to be removed from our model peptides by $Ps$ApaH (4) (Supplementary Fig. 5). However, the deacetylase activity was strongly reduced compared to the cluster 1b enzymes $Kp$HdaH and $Vs$HdaH, suggesting analogy to $Rs$PrpH (1b) and $Rw$DmhA (1b), in that the enzyme appear to have lost some of its lysine deacetylase efficiency, divergently evolving toward a different substrate. The preference to act as a deacetylase, being inactive in deacylating longer acyl-chain types and favoring strongly positively charged peptides, supported the finding that $Ps$ApaH (4) acts as deacetylase for the polycationic polyamines (Figs. 2d and 3a).

For $Vc$HdaH (5b) we were not able to identify a clear substrate under the pre-screening conditions. However, carefully inspecting the data and performing the experiments at higher enzyme concentration (1 μM versus 300 nM) allowed us to identify $Vc$HdaH (5b) as long-chain

deacylase, being the only enzyme of our collection acting as de-decanoylase (Supplementary Figs. 5 and 6).

Overall, we did not detect any deacylase capable of removing lysine myristoylation when a negatively charged Asp is present at the −1 position (Fig. 3a) and none of the deacylases were efficient depalmitoylases for the p53-peptide QPKK$_{pal}$ (Supplementary Figs. 5 and 6). Moreover, while the bacterial sirtuin deacylase CobB was shown to remove the negatively charged lysine malonylation and succinylation as well as removing β-hydroxyisobutyrylation, the bacterial deacylases studied here were neither capable of removing succinyl-/glutaryl-groups nor the β-hydroxyisobutyryl-group from lysine side chains of the histone H4-derived peptide LGK$_{suc}$/LGK$_{glu}$/LGK$_{bio}$ (Fig. 3a; Supplementary Figs. 5 and 6). For a better quantitative characterization of the enzymatic activities toward different peptides and acyl-chain types to be able to compare their activities, we next performed Michaelis–Menten kinetics.

## Michaelis–Menten kinetics of bacterial lysine deacylases

In order to be able to quantify enzyme efficiencies and to compare the bacterial deacylases with their mammalian counterparts, we next performed enzyme kinetics (Fig. 3b–e; Table 1; Supplementary Figs. 8–11; Supplementary Data 7). In the cases where we discovered an activity toward different peptides, we selected the peptides for which we found the highest deacylation activity in the pre-screening (Fig. 3a; Supplementary Figs. 6 and 7). We were not able to measure kinetics for $Kp$HdaH (1b), $Lc$ApaH (3) $Lp$ApaH (3) against myristoylated peptide substrates nor for the deacylase $Vc$HdaH (5b) against decanoylated substrates due to low substrate conversion rates and high $K_M$ values. However, $Kp$HdaH (1b) showed a $K_M$-value for the p53-peptide QPKK$_{ac}$ of 18 μM and a turnover number, $k_{cat}$, of 0.057 s$^{-1}$ resulting in an overall catalytic efficiency of $k_{cat}/K_M$: $3.1 \times 10^3$ M$^{-1}$s$^{-1}$ (Fig. 3b; Table 1; Supplementary Fig. 8).

For the enzyme $Vs$HdaH (1b) we found an almost 12-fold increase in efficacy, $k_{cat}/K_M$: $37 \times 10^3$ M$^{-1}$ s$^{-1}$, due to the almost 30-fold higher turnover (1.73 s$^{-1}$ versus 0.057 s$^{-1}$) for deacetylation of the p53-peptide QPKK$_{ac}$ (Fig. 3b; Table 1). Next to its deacetylase activity, $Vs$HdaH (1b) is also active as delactylase for the L-lactylated p53-peptide QPKK$_{L-la}$ (Fig. 3c; Table 1; Supplementary Figs. 7 and 8). However, for the delactylase activity the catalytic efficiency is reduced 46-fold ($k_{cat}/K_M$: $0.8 \times 10^3$ M$^{-1}$ s$^{-1}$ versus $37 \times 10^3$ M$^{-1}$ s$^{-1}$) compared to its deacetylase activity (Table 1; Supplementary Figs. 7 and 8). These values are in good agreement with reported values for mammalian HDACs[159]. Supporting the idea that the enzyme $Rw$DmhA (1b) has divergently evolved from a lysine deacetylase toward an enzyme with altered substrate specificity, we observed an overall 60-fold reduced catalytic efficiency for deacetylation of the p53-derived peptide QPKK$_{ac}$ compared to the $Vs$HdaH (1b) ($k_{cat}/K_M$: $0.6 \times 10^3$ M$^{-1}$ s$^{-1}$ versus $37 \times 10^3$ M$^{-1}$ s$^{-1}$), which is mainly due to the 70-fold decreased turnover

## Table 1 | Michaelis–Menten kinetics of deacylation of peptides by selected bacterial deacylases

| Enzyme | Substrate | $K_M$ (μM) | $K_i$ (μM) | $k_{cat}$ (s$^{-1}$) | $k_{cat}/K_M$ (M$^{-1}$·s$^{-1}$) |
|---|---|---|---|---|---|
| $Bs$AcuC (2c) | LGKac[a] | 83 | – | 0.022 | 0.3·10$^3$ |
| | LGKpro[a] | ~160 | – | 0.04 | ~0.2·10$^3$ |
| | LGKcr[a] | ~170 | – | 0.02 | ~0.1·10$^3$ |
| $Kp$HdaH (1b) | QPKKac | 18 | – | 0.057 | 3.1·10$^3$ |
| $Lc$ApaH (3) | QPKKac[b] | ~2000 | ~13 | ~123 | ~63·10$^3$ |
| | QPKKpro | ~300 | – | 0.6 | ~2·10$^3$ |
| | LGKac[a] | 15.2 | – | 2.10 | 139·10$^3$ |
| | LGKpro[a] | ~280 | – | 0.16 | 0.6·10$^3$ |
| | LGK(L-la)[a] | 170 | – | 0.006 | 0.03·10$^3$ |
| | LGK(D-la)[a] | 62 | – | 0.0022 | 0.04·10$^3$ |
| $Lp$ApaH (3) | QPKKac[b] | ~1000 | ~44 | ~74 | ~74·10$^3$ |
| | QPKKpro | ~200 | – | 0.42 | ~2·10$^3$ |
| | LGKac[a] | 21 | – | 1.56 | 75·10$^3$ |
| | LGKpro[a] | 150 | – | 0.065 | 0.43·10$^3$ |
| $Ps$ApaH (4) | QPKKac[b] | 43 | 122 | 0.09 | 2.1·10$^3$ |
| $Rw$DmhA (1b) | QPKKac[b] | 42 | 154 | 0.024 | 0.6·10$^3$ |
| $Vs$HdaH (1b) | QPKKac | 47 | – | 1.73 | 37·10$^3$ |
| | QPKK(L-la) | 53 | – | 0.044 | 0.8·10$^3$ |

Shown are the enzymes, the peptide sequences with the acyl-modifications on the lysine side chains, and the results obtained for $K_M$, turnover number $k_{cat}$, and catalytic efficiency $k_{cat}/K_M$. The experiments were performed in two independent replicates ($n = 2$). Data are presented as means. Source data are provided as Source Data file.
[a]Data acquired with continuous assays.
[b]Data adjusted to enzyme kinetics with substrate inhibition at high concentration, with $K_i$ as indicated.

number ($k_{cat}$: 0.024 s$^{-1}$ versus 1.73 s$^{-1}$) while the $K_M$-values are almost identical (42 μM versus 47 μM) (Fig. 3b; Table 1). We observed an inhibition of enzyme activity for *Rw*DmhA (1b) for higher substrate concentrations (Table 1; Fig. 3b; Supplementary Fig. 8). We assume this to be the case regardless of $K_i$ or $K_M$, since inhibition by the deacylated product (or by acetate/acyl-group) would otherwise result in loss of reaction linearity over time, which we did not observe. To this end, we determined an inhibition constant of $K_i$: 154 μM for the substrate inhibition of *Rw*DmhA (Table 1).

*Bs*AcuC (2c) was very unstable in our hands and especially under the assay conditions, being active only for 10–15 min. This did not allow us to measure kinetics against the peptides QPKK$_{ac}$ and QPKK$_{oct}$ in the end-point assay format. Instead, for *Bs*AcuC (2c) we turned to a continuous assay format that uses the H4 peptide LGK$_{acyl}$ substrates, which allowed the determination of initial conversion rates for this enzyme (Fig. 3d; Supplementary Fig. 9). We observed a similar catalytic efficiency ($k_{cat}$/$K_M$ values) for deacylation of acetylated, propionylated and crotonylated histone H4 peptides (Fig. 3d; Table 1). Moreover, compared to *Lp*ApaH (3) the $k_{cat}$/$K_M$ value for the deacetylation of the H4 peptide LGK$_{ac}$ is more than three orders of magnitude reduced (Table 1; Fig. 3d, e; Supplementary Figs. 9 and 10). These data are in agreement with the pre-screening results showing a higher level of substrate promiscuity for *Bs*AcuC (2c); however, at the expense of a lower enzyme efficiency compared to other bacterial deacylases in our collection (Fig. 3a, d). This supports our model in which *Bs*AcuC (2c) evolutionary developed toward an enzyme with a high level of substrate acyl-chain promiscuity, however, with a very narrow substrate range for which it shows a high catalytic efficiency[126,149].

The enzyme *Ps*ApaH (4) shows a similar kinetic profile as deacetylase as *Kp*HdaH (1b) ($K_M$: 43 μM versus 18 μM; $k_{cat}$: 0.09 s$^{-1}$ versus 0.057 s$^{-1}$; $k_{cat}$/$K_M$: 2.1 × 10$^3$ M$^{-1}$ s$^{-1}$ versus 3.1 × 10$^3$ M$^{-1}$ s$^{-1}$) (Fig. 3b; Table 1). However, for *Ps*ApaH (4) we discovered a substrate inhibitory effect in the deacetylation reaction of the p53-dervied peptide QPKK$_{ac}$ at high concentrations as described above for *Rw*DmhA (1b), resulting in an inhibition constant of $K_i$: 122 μM for substrate inhibition (Table 1; Fig. 3a; Supplementary Fig. 8).

For the enzymes *Lc*ApaH (3) and *Lp*ApaH (3), we discovered the highest deacetylation efficiency of all compared enzymes with $k_{cat}$/$K_M$-values of 139 × 10$^3$ M$^{-1}$ s$^{-1}$ and 75 × 10$^3$ M$^{-1}$ s$^{-1}$ for the histone H4 peptide LGK$_{ac}$, respectively (Fig. 3e; Table 1; Fig. 3e; Supplementary Fig. 11). For deacetylation of the peptide QPKK$_{ac}$, catalyzed by *Lc*ApaH (3) and *Lp*ApaH (3), we obtained high $K_M$-values for both enzymes ($K_M$: 2000 μM and 1000 μM) and high turnover numbers ($k_{cat}$: ~123 s$^{-1}$ and ~74 s$^{-1}$), resulting in catalytic efficiencies of $k_{cat}$/$K_M$: 63 × 10$^3$ M$^{-1}$ s$^{-1}$ and 74 × 10$^3$ M$^{-1}$ s$^{-1}$, respectively (Fig. 3a; Table 1; Supplementary Fig. 10). Moreover, we obtained low inhibition constants ($K_i$: 13 μM and 44 μM, respectively) for substrate inhibition, suggesting a potential allosteric regulation of enzyme activity by substrates (Table 1; Supplementary Fig. 10). The depropionylase activities against LGK$_{pro}$ are 65-fold and 175-fold reduced compared to deacetylation of LGK$_{ac}$ for *Lc*ApaH (3) [$k_{cat}$/$K_M$ (LGK$_{ac}$): 139 × 10$^3$ M$^{-1}$ s$^{-1}$ vs $k_{cat}$/$K_M$ (LGK$_{pro}$): 0.6 × 10$^3$ M$^{-1}$ s$^{-1}$] and for *Lp*ApaH (3) [$k_{cat}$/$K_M$ LGK$_{ac}$: 75 × 10$^3$ M$^{-1}$ s$^{-1}$ vs $k_{cat}$/$K_M$ LGK$_{pro}$: 0.43 × 10$^3$ M$^{-1}$ s$^{-1}$], respectively (Table 1; Fig. 3e; Supplementary Fig. 11). *Lc*ApaH (3) converts the lactylated peptides LGK$_{D-la}$ and LGK$_{L-la}$ with comparable catalytic efficiency for the two stereoisomers ($k_{cat}$/$K_M$ (K$_{L-la}$): 0.03 × 10$^3$ M$^{-1}$ s$^{-1}$ and $k_{cat}$/$K_M$ (K$_{D-la}$): 0.04 × 10$^3$ M$^{-1}$ s$^{-1}$) but more than three orders of magnitude less efficient than its deacylation of LGK$_{ac}$ ($k_{cat}$/$K_M$ (K$_{ac}$): 139 × 10$^3$ M$^{-1}$ s$^{-1}$) (Table 1; Fig. 3c; Supplementary Fig. 11).

Overall, our data show bacterial deacylases having preferences for substrate sequence, some favoring a positively charged residue or a glycine residue at the −1 position, and different acyl-chain type preferences. As an example, the *Ps*ApaH (4) only acts as deacetylase, exclusively removing acetyl-groups but no longer acylations from lysine side chains (Fig. 3a). Moreover, it only deacetylates peptides containing a Lys or Arg at the −1 position in agreement with our finding

that *Ps*ApaH (4) also acts as polyamine deacetylase (Figs. 2d and 3a; Supplementary Fig. 4b, c). The cluster 1 enzymes *Kp*HdaH (1b), *Vs*HdaH (1b), *Rw*DmhA (1b), and *Rs*PrpH (1b) show the strongest activities as deacetylases (Table 1; Fig. 3a; Supplementary Fig. 8). After screening of selected enzymes for a potential activity as protein deacetylase or polyamine deacetylase our next question was what governs substrate specificity. To this end, we structurally characterized selected enzymes.

### Overall structures of bacterial ami(n/d)ohydrolases

We solved the crystal structures of seven bacterial classical deacetylases: three enzymes representing cluster 1 (*Kp*HdaH (1b);, *Rw*DmhA (1b), *Rs*PrpH (1b)), two enzymes of cluster 3 (*Lc*ApaH and *Lp*ApaH), one enzyme of cluster 4 (*Ps*ApaH) and one enzyme representing cluster 5 (*Vc*HdaH (5b)) by X-ray crystallography (Fig. 4). Even under intensive trials, we were not able to obtain crystals for the cluster 2 enzyme *Bs*AcuC (2c). To this end, our analyses on cluster 2 enzymes will rely on AlphaFold2 structure prediction of *Bs*AcuC (2c).

The overall structures resembled the typical α/β-fold of the arginase-deacetylase family similar to classical mammalian deacetylases (Fig. 4a). The core contained a central eight-stranded parallel β-sheet flanked by three and five α-helices on each side, respectively (Fig. 4a). All active site residues were conserved, as shown by the multiple sequence alignment and the structural alignment, suggesting a general acid/base catalytic mechanism for the bacterial enzymes as described for the eukaryotic counterparts (Figs. 1c and 4b–d). These enzymes mediate catalysis using a penta-coordinated catalytic Zn$^{2+}$ ion coordinated by two aspartates and a histidine (*Kp*HdaH (1b): Asp181, Asp269, His183) (Fig. 4c). In bacterial deacylases of cluster 3, *Lp*ApaH (3) and *Lc*ApaH (3), the *Kp*HdaH His183 is replaced by an Asn (*Lp*ApaH (3): Asn221; *Lc*ApaH (3): Asn218) both being functionally conserved (Fig. 4b). The coordination is completed by two solvent molecules in the apo state, the substrates acetyl-group and the attacking water molecule in the substrate-bound state, or the product carboxylate coordinating the Zn$^{2+}$ in a bidentate fashion in the post-catalysis state (Fig. 4c)[160–162]. A double-His motif, (*Kp*HdaH (1b): His143/His144) plays an important role during catalysis. The first His (*Kp*HdaH: His143) of this tandem His motif acts as an electrostatic catalyst during catalysis, i.e. it orients and polarizes the catalytic water molecule, and it is itself oriented and polarized by an Asp (*Kp*HdaH (1b): Asp179). The second His (*Kp*HdaH: His144) acts as a general base abstracting a proton from the catalytic water molecule, thereby activating the water molecule for nucleophilic attack on the carbonyl carbon of the acetyl/amide-group, i.e. the acetyl-lysine, the acetylated polyamine or the small molecule amide (Fig. 4d). This second His acting as general base is oriented and polarized either by an Asn (*Kp*HdaH: Asn186) or an Asp (*Hs*HDAC8: Asp183) (Fig. 4b). A mutation of the second His in *Kp*HdaH (*Kp*HdaH H144A) leads to an inactivation of the enzyme while not affecting the overall conformation (Fig. 2; Supplementary Fig. 12; Supplementary Table 5). For all structures we identified additional potassium ions next to the catalytic Zn$^{2+}$ ion (Fig. 4c). Two potassium ions were also described for mammalian classical deacetylases[163,164]. We observed additional potassium ions for some bacterial deacylases, i.e. *Kp*HdaH (1b) and *Vc*HdaH (5b), binding in the unoccupied substrate/inhibitor binding tunnel may be due to high concentrations of potassium present in the crystallization condition, and due to the highly negatively charged surface area as explained above. Potassium ion 1 (K$^+$1) interacts with the side and main chain of an Asp (*Kp*HdaH (1b): Asp179) and the main-chain carbonyl groups of an Asp (*Kp*HdaH (1b): Asp181) and of a His (*Kp*HdaH: His183) (Fig. 4c). To this end, the K$^+$1 ion is indirectly involved in catalysis and coordination of the catalytic Zn$^{2+}$ ion. For K$^+$1, the full hexa-coordinated geometry is established by further coordination of the side chain of a conserved Ser and the main chain of a Leu (*Kp*HdaH (1b): Ser202; Leu203) (Fig. 4c). The second potassium ion, K$^+$2, binds distantly from the active site and has a pure structural role

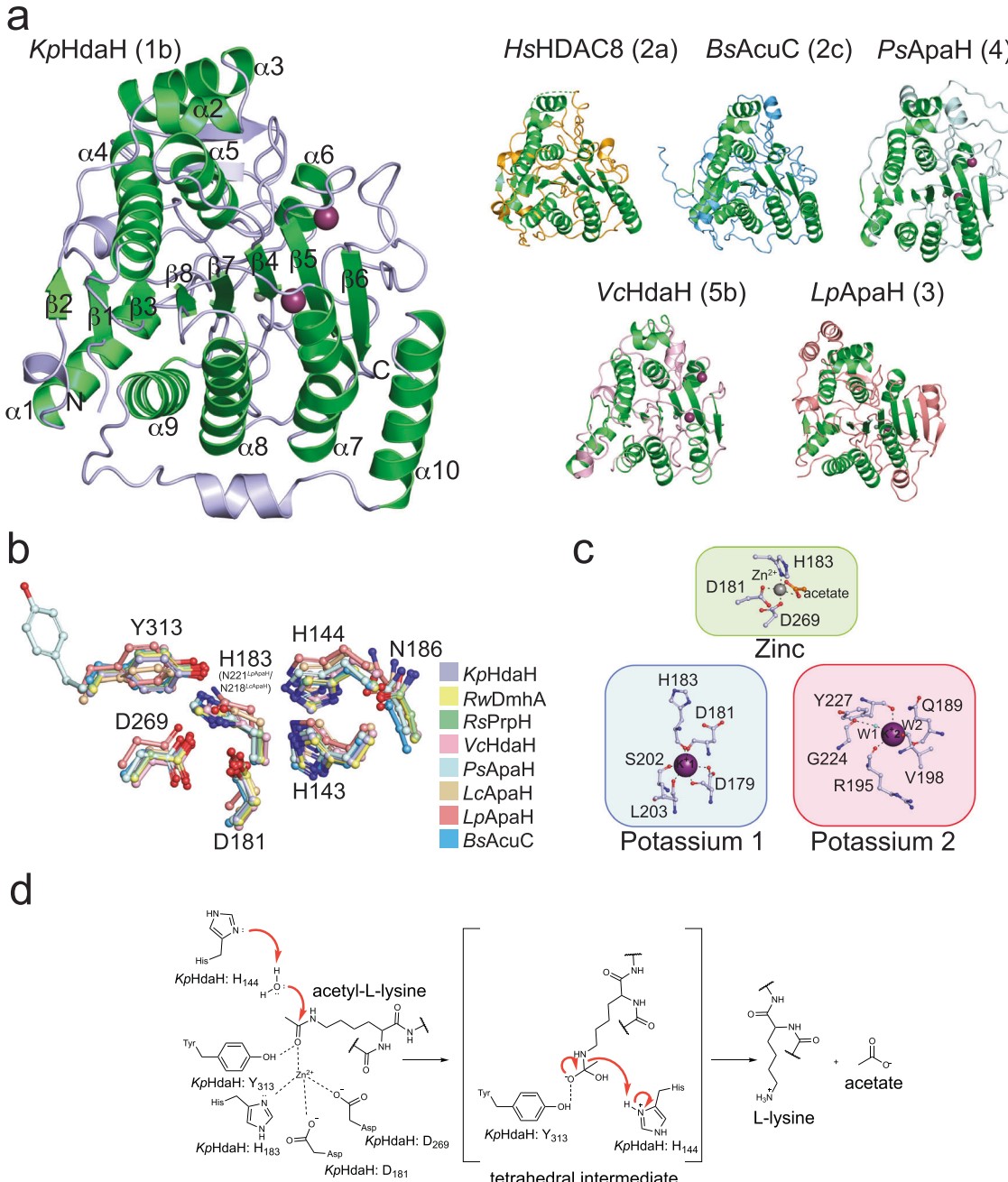

**Fig. 4 | Bacterial Zn²⁺-dependent deacylases share a highly conserved structural α/β-arginase/deacetylase fold. a** Ribbon representation of the catalytic cores of bacterial deacylases. For *Bs*AcuC (2c) we show the AlphaFold2 model for the other crystal structures. All structures share a highly similar catalytic core domain (green) consisting of an eight-stranded parallel β-strand surrounded by α-helices. Additional structural features emanate from the catalytic core domain. Cluster 1 enzymes have the extended N-terminal L1-loop and an additional short C-terminal α-helix (light blue). Cluster 2 and 5 enzymes contain an extended foot pocket and additional small α-helices and loops (*Bs*AcuC (2c): blue; *Vc*HdaH (5b): pink). In cluster 3 enzymes the central parallel β-sheet is extended by two additional anti-parallel β-strands as shown here for *Lp*ApaH (3) and by an additional pair of α-helices emanating from the core domain (red). Cluster 4 enzymes contain an extended L2-loop (PSL) (light blue). **b** The active site architecture in bacterial Zn²⁺-dependent deacylases is totally conserved. The active site Zn²⁺-ion (not shown here) is coordinated by two Asp-residues and a His (*Kp*HdaH (1b): Asp181, Asp269,

His183). The His acting as general base/acid (*Kp*HdaH (1b): His144) is part of a double-His motif (*Kp*HdaH (1b): His143, His144). Both His form charge-relay systems with two Asp or an Asp and an Asn/Gln (*Kp*HdaH (1b): Asp179, Asn186), respectively. In agreement with eukaryotic HDACs, we found an Asn as part of the second charge-relay system in cluster 1 (*Kp*HdaH (1b): Asn186), cluster 3 (*Lc*ApaH (3): Asn221) and cluster 4 (*Ps*ApaH (4): Asn200). **c** Metal ions are important for the structure and catalysis of bacterial deacylases. As their eukaryotic counterparts also bacterial deacylases use an active site Zn²⁺-ion for catalysis. Moreover, bacterial deacylases contain one or two potassium ions. Potassium ion 1 (K⁺1) is indirectly involved in catalysis while the second potassium ion, K⁺2, is a structural metal ion. Numbering is shown for *Kp*HdaH (1b). **d** Bacterial deacylases exert a conserved catalytic mechanism for substrate deacylation. A catalytic base His abstracts a proton from a catalytic water molecule as a nucleophile attacking the acyl-group. A tetrahedral intermediate is formed which collapses to form the deacylated substrate. Numbering is shown for *Kp*HdaH (1b).

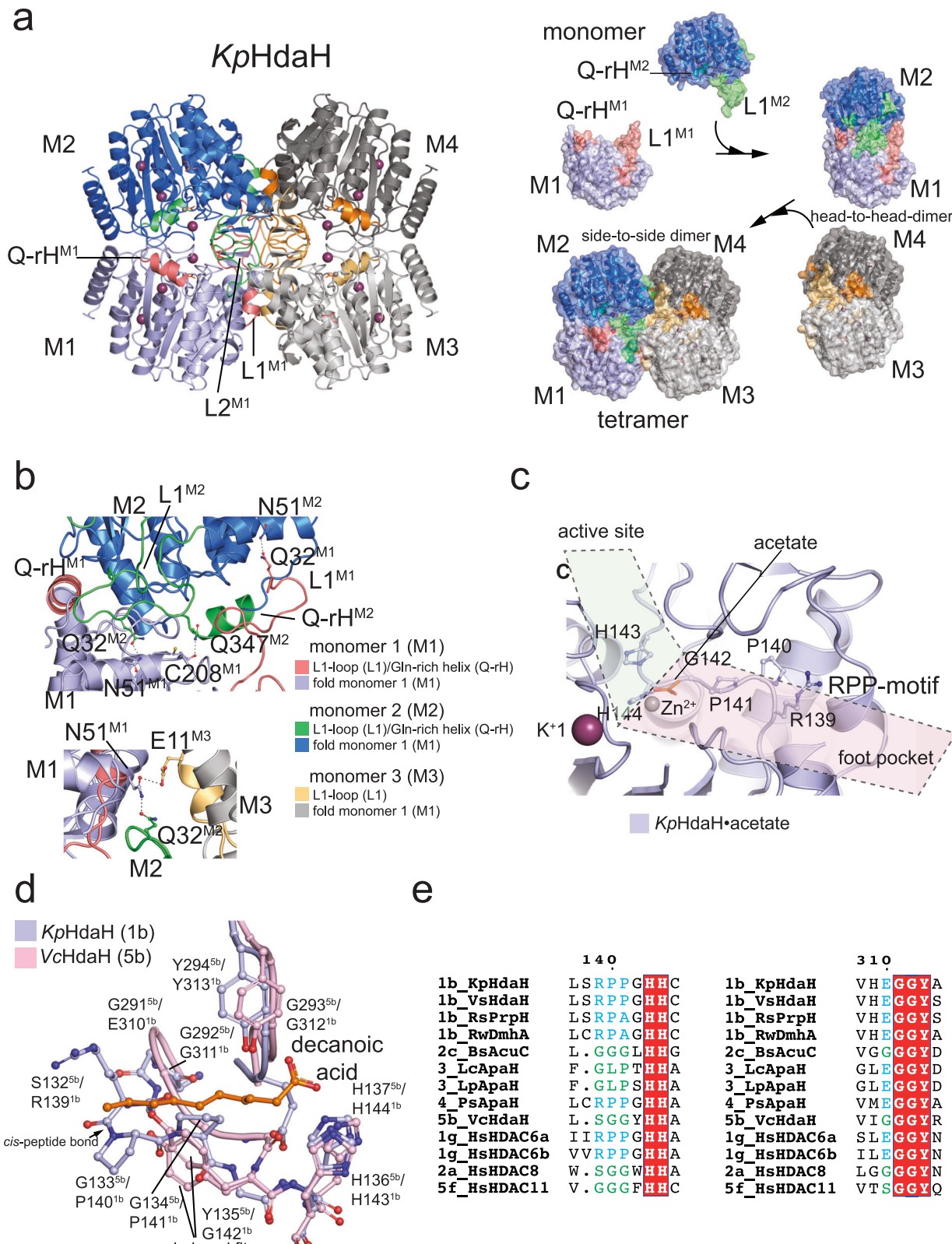

(Fig. 4c). It is hexa-coordinated by the main-chain carbonyl oxygens of several residues (KpHdaH: Val189, Tyr192 (not shown in Fig. 4c), Arg195, Tyr227) and by two additional water molecules (Fig. 4c). The mammalian HDACs are sub-classified into class IIa, in which the catalytic Tyr is replaced by a His and class IIb containing the catalytic Tyr. This Tyr, (KpHdaH (1b): Tyr313; HDAC8: Tyr306), is involved in the stabilization of the tetrahedral oxyanion intermediate during catalysis

(Fig. 4b, d). The Tyr (KpHdaH (1b): Tyr313) follows a glycine-rich sequence that is conserved in all classical deacetylases, containing two strictly conserved Gly side chains (KpHdaH (1b): Gly311, Gly312) (Fig. 1c; Supplementary Fig. 1). This sequence confers conformational flexibility for substrate binding and catalysis in agreement with earlier studies performed on human HDAC8 (Fig. 1c)[165]. For the bacterial enzymes, we exclusively observed the presence of a Tyr at this

**Fig. 5 | Bacterial sub-cluster 1b enzymes are robust deacetylases forming tetramers in solution. a** Bacterial sub-cluster 1b enzymes form tetramers in solution mediated by the N-terminal L1-loop and by an additional α-helix in an extended loop leading to the C-terminal α-helix (Gln-rich helix Q-rH in *Kp*HdaH (1b)). Left panel: The tetramers consist of two head-to-head dimers arranged side-by-side. The L1-loop of each monomer subunit (M1-M4) contacts all remaining subunits of the tetramer and of the substrate. The L2-loop encompasses a β-hairpin motif placed in the interface of the tetramer. Right panel: Analyses of the interface area and mutational studies suggest two head-to-head dimers form the tetramer by side-to-side arrangement. **b** Closeup of the tetramer interface area reveals interactions needed for the composition of the integral tetramer. As an example, head-to-head dimer formation is mediated by the main-chain carbonyl of Cys208/Cys207 of one subunit (monomer: M) forming hydrogen bonds to the side chains of Gln347/ Trp346 of the other subunit of *Kp*HdaH (1b), respectively. Moreover, Gln32 of the L1-loop of one monomer forms a direct interaction with Asn51 of the other

monomer. **c** The active site is almost perpendicular to the foot pocket. The architecture of the foot pocket needed for substrate release explains the preference of *Kp*HdaH (1b) to act as a robust deacetylase. **d** The foot pocket is extended in *Vc*HdaH (5b) compared to *Kp*HdaH (1b) allowing it to accommodate the decanoic acid. The RPP-motif is replaced in *Vc*HdaH (5b) by 132-SGGYHH-137 and the XGGY-motif by 290-GGGY-294 explaining structurally how the foot pocket in *Vc*HdaH (5b) is able to accommodate the longer acyl-chain. The RP-peptide bond in the RPP-motif is in *cis*-configuration resulting in restricting the volume of the foot pocket. Tyr135$^{5b}$ (superscript 5b: *Vc*HdaH (5b); superscript 1b: *Kp*HdaH (1b)) lines the alkyl chain of the fatty acid (induced fit mechanism) and the extended Gly-rich GGGY-motif in *Vc*HdaH (5b) allows structural flexibility for the active site Tyr294$^{5b}$. **e** Multiple sequence alignment of the RPP- and XGGY-motif lining the foot pocket. All robust deacetylases show a conserved RPP- and XGGY-motif as shown for bacterial deacetylases and selected classical HDACs.

position. After analyzing the common structural features of bacterial ami(n/d)ohydrolases, we next studied the structures of each cluster to derive molecular determinants for the observed activity, substrate selectivity, and acyl-chain preferences.

## Oligomerization and substrate specificity in cluster 1 DACs

We solved the crystal structures of three enzymes representing cluster 1 amidohydrolases: *Kp*HdaH (1b), *Rw*DmhA (1b), and *Rs*PrpH (1b) (Figs. 3a and 4; Supplementary Fig. 12). This cluster contains enzymes with efficient lysine deacetylase activity, but also evolutionary divergent enzymes as *Rw*DmhA (1b) and *Rs*PrpH (1b) developed to be active as small molecule amidohydrolases (Fig. 3a, b; Table 1; Supplementary Fig. 6).

All cluster 1 enzymes eluted as apparent trimer or tetramer from the analytical SEC column (Supplementary Fig. 2a, b), and our crystal structures confirmed the formation of tetramers for sub-cluster 1b enzymes (Fig. 5a). Overall, the structure of *Kp*HdaH (1b) resembles the structure of *P. aeruginosa* HdaH solved earlier with an overall r.m.s.d. of 0.41 Å[105]. All cluster 1b enzymes contain the so-called L1-loop at the N-terminus as a structural feature mediating the formation of a tetramer via side-by-side arrangement of head-to-head dimers supporting earlier investigations[105]. The L1-loop is important for the side-to-side arrangement of two head-to-head dimers in order to form the tetramer (Fig. 5a, b). The L1-loop of each monomer, acting as molecular glue, contacts all three remaining monomers of the tetramers. The L1-loop is also involved in mediating substrate specificity by restricting access to the enzymes' active sites (Fig. 5a, b). Notably, we deleted part of the L1-loop (Δ20-37 *Kp*HdaH), which resulted in a dimeric enzyme showing enzymatic activity albeit slightly reduced compared to full-length *Kp*HdaH (1b), suggesting tetramer formation is needed for full activity (Supplementary Fig. 13). Analysis of the interface area in the head-to-head dimer versus the side-by-side dimer suggests deletion of the L1-loop abrogates side-by-side dimer formation while leaving the head-to-head dimer intact. The interface area of the head-to-head dimer within the full tetramer is 13,160 Å$^2$, while the interface area of the side-by-side dimer is just 4400 Å$^2$ as determined by the PISA server[166]. To understand the structural mechanisms restricting the acyl-chain preference of cluster 1 enzymes to act as robust deacetylases, we analyzed the substrate binding tunnel and the product release channel (foot pocket). We postulate restricting the volume of the foot pocket, compared to other bacterial deacylases does not allow longer acyl-chains to be accommodated contributing to restricting the activity to remove acetyl-groups rather than longer acyl-chains (Fig. 5c). This is mechanistically achieved by the presence of a proline-rich loop, the RPP-motif (general: RPPXHH; *Kp*HdaH: 139-RPPGHH-144; *Rw*DhmhA: 136-RPAGHH-143; *Rs*PrpH: 148-RPAGHH-153) (Fig. 5d). The RPP-motif is totally conserved in cluster 1 enzymes (Supplementary Fig. 1a, b; Supplementary Fig. 14). Only for *Rw*DmhA (1b) and *Rs*PrpH (1b), which evolved toward non-protein substrates, the second proline is substituted for an Ala

suggesting this position being important for substrate specificity to accommodate the product (Fig. 5d, e). The peptide bond preceding Pro140 in *Kp*HdaH (1b) connecting it to Arg139 is in *cis*-configuration resulting in twisting the loop thereby reducing the volume of the product release cavity, which is therefore not able to accommodate longer acyl-chains than acetyl-groups (Fig. 5d). This foot pocket is lined on the other side by the highly conserved XGGY-motif (Fig. 5d, e). The Tyr in this motif is the catalytic Tyr (*Kp*HdaH: Tyr313) involved in the polarization of the acetyl-group and stabilization of the tetrahedral intermediate during catalysis, which is following the double-Gly motif needed to convey flexibility for orientation of the active site Tyr and for providing the volume of the foot pocket (Figs. 4d and 5e; Supplementary Fig. 1b, d). The presence of a small side chain N-terminal in the XGGY-motif, i.e. X is Gly, Ser/Thr, etc., indicates the possibility of accommodating longer acyl-chains. In *Kp*HdaH (1b), *Rw*DhmhA (1b) and *Rs*PrpH (1b), being robust deacetylases, the sequence is EGGY (*Kp*HdaH (1b): 310-EGGY-313) the presence of glutamate further restricting the volume of the foot pocket (Figs. 3a and 5e; Supplementary Fig. 14). The fact that cluster 1b enzymes discriminate between peptides with positively charged residues at −1 position over negatively charged residues or glycine at −1 position suggest that the positive charge at this position is an important feature for substrate recognition (Fig. 3a). Analyses of the surface area surrounding the entry of the active site tunnel shows a highly acidic negatively charged electrostatic surface potential explaining the preference for Arg or Lys at −1 position (Supplementary Fig. 15a). As a support, we observed binding of a potassium ion at the negatively charged active site entry in the structure of *Kp*HdaH (1b). Notably, for all sub-cluster 1b enzymes, except for *Rw*DmhA (1b), we found an acetate molecule in the active site coordinating the active site Zn$^{2+}$ ion. For *Rw*DmhA (1b) we identified an all-*trans* octanoic acid in the substrate binding tunnel coordinating the active site Zn$^{2+}$ ion with its carboxylate moiety (Supplementary Fig. 12). We did neither supply the acetate nor the octanoic acid in the crystallization condition, which might therefore originate from protein production and the activity of the enzymes in *E. coli*. *Rw*DmhA (1b) is a dimethoate amidohydrolase, i.e. the substrate specificity changed from acetylated lysine side chains to a small molecule phosphorothionate. For *Rs*PrpH (1b) acting as arylamidase converting the pesticide propanil to 3,4-dichloraniline rather than lysine deacetylase. For all cluster 1 enzymes, the active site architecture and the overall-fold is totally conserved (Supplementary Data 6; Supplementary Fig. 3). We identified a specific structural feature of cluster 1 deacetylases, i.e. an additional hydrophobic C-terminal α-helix (*Kp*HdaH (1b): 340-LLEFIQQQQ-348 (Gln-rich helix; Q-rH; Fig. 5a, b); *Rw*DmhA (1b): 339-VLEMAEAW-346; *Rs*PrpH (1b): 349-ELEMFALWQ-357) structurally contributing to head-to-head dimer formation, amongst others by formation of a hydrogen bond between the main-chain carbonyl of C208/C207 and the side chain of Q347/W346 for *Kp*HdaH (1b)/ *Rw*DmhA (1b), respectively (Fig. 5a, b). This α-helix furthermore restricts substrate access to the active site in transforming the edge of the active

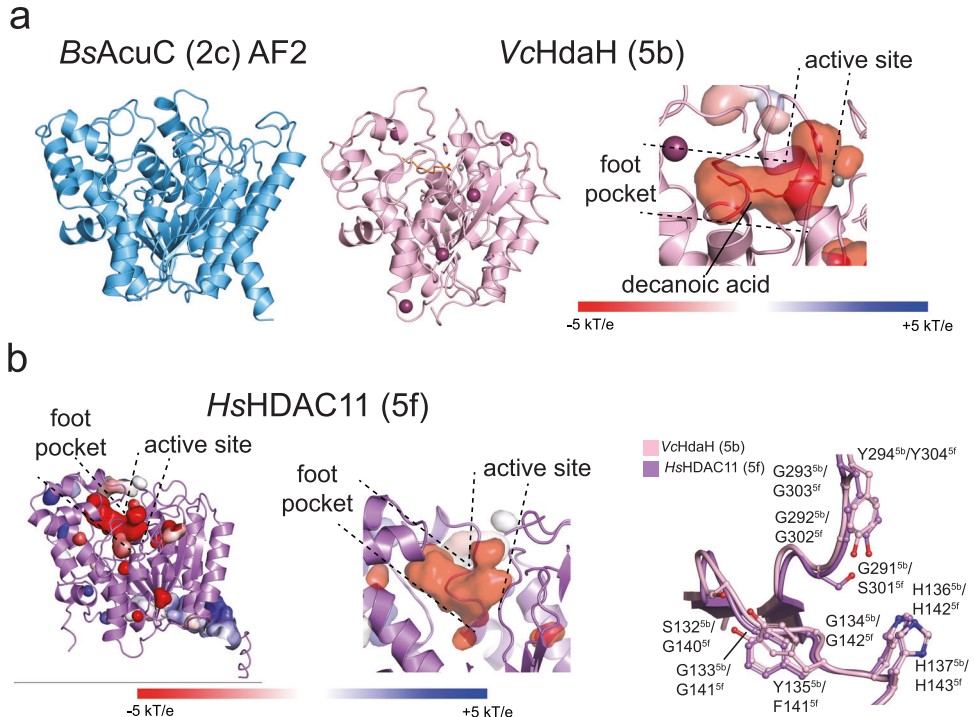

**Fig. 6 | VcHdaH belongs to sub-cluster 5b and is a de-decanoylase.**
**a** AlphaFold2 structure prediction of *Bs*AcuC (2c) shows a typical arginase-deacetylase fold with a central eight-stranded parallel β-sheet surrounded by α-helices. Right panel: The crystal structure of the enzyme *Vc*HdaH (5b) reveals the binding of a decanoic acid in the foot pocket. The foot pocket is arranged almost perpendicular to the substrate binding tunnel leading to the active site. The structure shows coordination of the active site $Zn^{2+}$-ion by the decanoic acid carboxylate in a bidentate fashion. The electrostatics were plotted onto the interior surface of the *Vc*HdaH (5b) structure using the APBS plugin in PyMOL[220]. This shows an extended foot pocket to accommodate the decanoic acid. This pocket is negatively charged. **b** Structural superposition of the structure of *Vc*HdaH (5b) and human HDAC11 (5f) suggest similar molecular mechanisms underlying the observed activities as de-fatty acylases. As no experimental structure is known for human HDAC11, an AlphaFold2 model was used here. For both, an extended foot pocket is present allowing to release the products of the deacylation reaction. Both enzymes are capable of acting as deacylases for longer acyl-chains. The structural superposition shows the structural similarity of the foot pockets in human HDAC11 (5f) and *Vc*HdaH (5b).

site entry (Fig. 5a, b). Sequence differences particularly in the N-terminal region containing the L1-loop, and in the C-terminal region can explain the differences in substrate specificity. For *Rw*DmhA (1b) and *Rs*PrpH (1b) we discovered a hydrophobic 22-LFL-24/32-LYF-34 motif in the L1-loop lining the substrate binding tunnel, the sequence 98-GHLAP-102/107-GMLAP-111 as well as the sequence 342-MAEAW-346/352-MFALW−356 in the C-terminal α-helix lining the active site rim. These residues form a hydrophobic active site and contribute to the binding of the observed octanoic acid in *Rw*DmhA (1b). Those residues in *Rw*DmhA are replaced by polar residues in *Kp*HdaH (1b) (*Kp*HdaH (1b): 23-VTL-25; *Kp*HdaH: 99-GKEAP-102; 343-FIOOQ−347) contributing to substrate specificity (Supplementary Fig. 15b). These observations point to the importance of the residues lining the substrate binding tunnel in mediating substrate specificity of enzymes even within the same sub-cluster. As a summary, these data suggest the L1-loop, L2-loop, and the C-terminal α-helix are important structural features to mediate oligo-merization and to determine substrate specificity of sub-cluster 1b enzymes.

## Cluster 2 contains enzymes homologous to *B. subtilis* AcuC
We were not able to obtain crystals for an enzyme of cluster 2, although we extensively tried and used *B. subtilis* AcuC and *Staphylococcus aureus* AcuC. The AlphaFold2 model of BsAcuC (2c) shows the typical arginase-deacetylase fold (Fig. 4a; 6a). Enzyme kinetics suggest *Bs*AcuC (2c) being a promiscuous enzyme deacylating different substrate peptides modified with different acyl-chain types. However, *Bs*AcuC (2c) shows a reduced enzymatic efficiency compared to other deacylases (Fig. 3a, d; Table 1; Fig. 3d; Supplementary Fig. 9). In enzymes of cluster 2 the RPP- and XGGY-motifs are substituted by

XGGLHH and (G/A)GGY (*Bs*AcuC (2c): 128-GGGLHH-133; 298-GGGY −301), respectively, structurally supporting the ability of the foot pocket to accommodate various acyl-chain types (Fig. 5e; Supplementary Fig. 1a, c; Supplementary Fig. 16). *Bs*AcuC (2c) is known to be important for regulation of AMP-forming acetyl-CoA-synthetase (AcsA). *Bs*AcuC (2c) deacetylates *Bs*AcsA at the C-terminal K549 being part of a highly conserved sequence motif, i.e. 546-RSGKIMR-552 (Supplementary Fig. 17)[150]. This deacylation results in the activation of AcsA activity[126]. The observed high acyl-chain promiscuity but low overall enzymatic efficiency observed here for the tested peptide substrates might indicate that *Bs*AcuC (2c) evolved to be highly substrate specific toward *Bs*AcsA but still being able to remove various different acyl-chain types (Fig. 3a, d; Supplementary Figs. 6 and 10). Our data suggest that *Bs*AcuC (2c) is capable of deproionylate *Bs*AcsA, if *Bs*AcsA was propionylated in vivo. However, this needs further investigation.[151] The low catalytic efficiency of *Bs*AcuC (2c) to deacylated histone H4-derived peptides ($LGK_{acyl}$, $H4_{10-12}$) containing a GK-dipeptide motif indicates these peptides cannot mimic the physiological substrate at the molecular level (Fig. 3a, d; Supplementary Fig. 6). We and others have shown that besides the sequence context also the three-dimensional structure of acylated lysine side chains is an important feature for substrate recognition by lysine deacylases[167,168].

## The foot pocket mediates long-chain deacylase activity
We solved a crystal structure of *Vibrio cholerae* HdaH (5b) (*Vc*HdaH (5b)) (Figs. 4a and 6a). The enzyme is a monomer in solution (Supplementary Fig. 2b). We were not able to determine any deacylation activity apart from a weak de-decanoylase activity for which we were not able to perform quantitative Michaelis−Menten kinetics

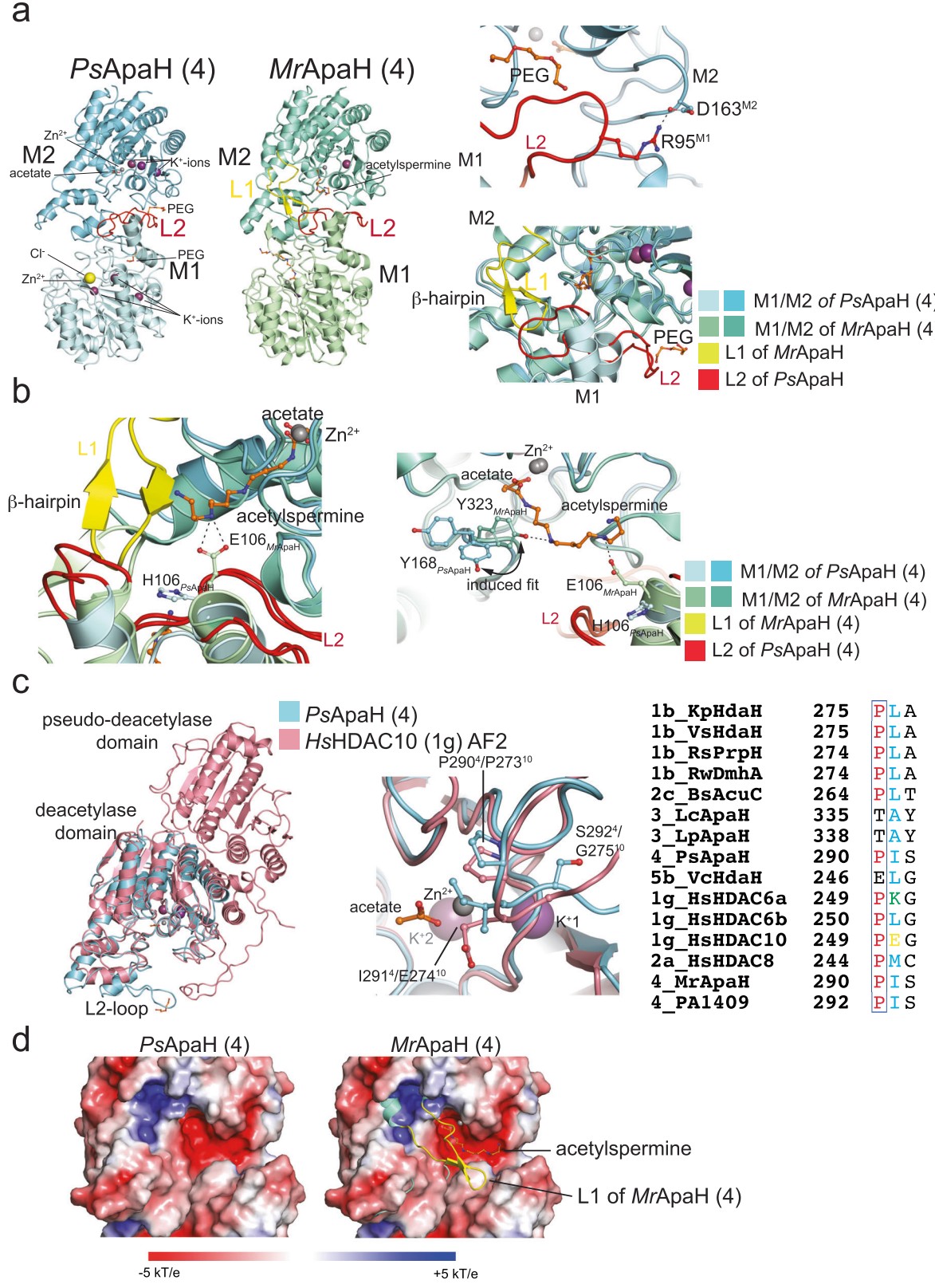

**a**

*Ps*ApaH (4) *Mr*ApaH (4)

**b**

**c**

| | | | |
|---|---|---|---|
| 1b_KpHdaH | 275 | P | L A |
| 1b_VsHdaH | 275 | P | L A |
| 1b_RsPrpH | 274 | P | L A |
| 1b_RwDmhA | 274 | P | L A |
| 2c_BsAcuC | 264 | P | L T |
| 3_LcApaH | 335 | T | A Y |
| 3_LpApaH | 338 | T | A Y |
| 4_PsApaH | 290 | P | I S |
| 5b_VcHdaH | 246 | E | L G |
| 1g_HsHDAC6a | 249 | P | K G |
| 1g_HsHDAC6b | 250 | P | L G |
| 1g_HsHDAC10 | 249 | P | E G |
| 2a_HsHDAC8 | 244 | P | M C |
| 4_MrApaH | 290 | P | I S |
| 4_PA1409 | 292 | P | I S |

**d**

*Ps*ApaH (4) *Mr*ApaH (4)

−5 kT/e +5 kT/e

(Supplementary Fig. 6). When inspecting the crystal structure, we observed a clear electron density for a decanoic acid approaching the catalytic $Zn^{2+}$ ion with its carboxylate in a bidentate fashion (Supplementary Fig. 18). Importantly, the position of the decanoic acid is almost perpendicular to the direction of substrate binding. This corresponds to the localization of acetate we observed for several crystal structures (Supplementary Figs. 12 and 18). We propose these carboxylic acids correspond to the deacylation products of the deacylation reaction binding to the foot pocket (Figs. 5d and 6b; Supplementary Fig. 18). For release, the pocket must open by an unknown mechanism which may involve binding of the next substrate molecule. In *Vc*HdaH (5b), acting as de-decanoylase, the RPP-motif lining the foot pocket is substituted by 132-SGGYHH-137 and the XGGY-motif by 291-GGGY−294 (Fig. 5d, e; Supplementary Fig. 1a, d; Supplementary

**Fig. 7 | Structural features of cluster 4 containing polyamine deacetylases.**
**a** The structure of *Ps*ApaH (4) confirms cluster 4 enzymes forming head-to-head dimers. Dimer formation proceeds via two structural features, i.e. an extended L2-loop (PSL, red) and the N-terminal L1-loop forming a two-stranded β-hairpin (yellow). The L1-loop was not resolved in the *Ps*ApaH (4) apo structure. Comparison with the structure of *M. ramosa*•$N^8$-acetylspermidine (PDB: 3Q9C) suggests an induced fit mechanism upon substrate binding stabilizing the L1-loop. Dimer formation is mediated by interactions such as the salt bridge between Arg95 of the PSL-loop of monomer 1 (M1) and Asp163 of monomer 2 (M2). **b** Superposition of the structure of *Ps*ApaH (4) and the structure of *M. ramosa* ApaH (4) in complex with $N^8$-acetylspermidine reveals molecular mechanisms of substrate specificity. The L1-loop mediates dimer formation and forms a lid of the active site *in cis* limiting active site access of the substrate also in trans, i.e. for the other monomer. The selectivity for $N^8$-acetylspermidine compared to $N^1$-acetylspermidine is created by *M. ramosa* ApaH (4) (PDB: 3Q9E) forming a salt bridge to Glu106 in the C-terminal end of

L2-loop with the $N^1$-amino group of the $N^8$-acetylspermidine/$N^4$-amino group of acetylspermine. This Glu106 in *Mr*ApaH is replaced by His106 in *Ps*ApaH (4) explaining the higher acetylpolyamine promiscuity. **c** The PEG-motif (yellow) in human HDAC10 (1g) is replaced by PIS (blue) in *Ps*ApaH (4). For *Hs*HDAC10 (1g), the PEG-motif (273-PEG-275) contains the Glu gatekeeper creating selectivity for $N^8$-acetylspermidine forming an electrostatic interaction with the secondary $N^4$-amino group. This sequence is replaced by 290-PIS-292 in *Ps*ApaH (4) and other bacterial polyamine deacetylases. Shown is a superposition of the structure of *Ps*ApaH (4) and *Hs*HDAC10 (1g) AF2. For CD1 of human HDAC6 (1g), *Hs*HDAC6a, a Lys (green) is present at this position in the PKG-motif, which explains the preference to deacylate C-terminal Lys-side chains. **d** Electrostatic surface representation of *Ps*ApaH (4) and *M. ramosa* ApaH (4) in complex with acetylspermine (PDB: 3Q9E) shows sterically restricted access to the active site. The electrostatics were plotted onto the surface of the *Ps*ApaH (4) using the APBS plugin in PyMOL[220].

Fig. 19). In agreement, for the class IV enzyme HDAC11 categorized in cluster 5, it was reported to be rather a fatty-deacylase for longer acyl-chains, such as lysine demyristoylase, than a deacetylase (Fig. 1a, b; Supplementary Data 3)[93,96,98]. The AlphaFold2 structure of HDAC11 supported earlier assumptions that it contains a hydrophobic pocket near the catalytic $Zn^{2+}$ ion, which corresponds to the foot pocket described here (Fig. 6b). In analogy to *Vc*HdaH (5b), in human HDAC11 the RPP-motif is replaced by 138-GGGFHH-143 and the XGGY-motif by 301-SGGY−304 enabling the formation of an extended product release cavity capable to accommodate longer acyl-chains (Fig. 5e). Interestingly, we observed an electron density in the substrate binding tunnel leading to the active site, in which we build an imidazole molecule as this was present during purification. This suggests bulkier substrates being able to enter the active site tunnel leaving the possibility of *Vc*HdaH (5b) having other small molecule substrates.

## Active site access and gatekeeper motif in polyamine DACs

To explain the molecular basis for classical deacylases to act as polyamine deacetylases, we structurally characterized the enzyme *Ps*ApaH (4) (Fig. 7; Supplementary Fig. 18). The crystal structure of *Ps*ApaH (4) confirmed the analytical SEC experiments supporting cluster 4 enzymes forming head-to-head dimers (Fig. 7a; Supplementary Fig. 2b). Dimer formation is mediated by a structural feature distinct to polyamine deacetylases, i.e. L2-loop (PSL: polyamine-specificity loop) (Fig. 7a; Supplementary Fig. 20). An important specificity-determining interaction is a salt bridge between R95 of the L2-loop of one monomer and D163 of the other monomer of the dimer (Fig. 7a).

The available structures on bacterial acetylpolyamine deacylases suggest several mechanisms creating specificity for acetylpolyamines and selectivity toward $N^8$-acetylsperimidine. Firstly, polyamine deacetylases have a sterically restricted active site formed upon dimer formation. The N-terminal L1-loop interacts in trans with the L2-loop, occluding the active site and thereby restricting the access of acetylated peptides (Fig. 7b). This L1-loop is highly flexible in the apo form, as it was not resolved in our apo structure of *Ps*ApaH (4). However, the L1-loop is visible and well-defined in the structures of *Mr*ApaH (4) in complex with $N^8$-acetylspermidine (PDB: 3Q9C) and acetylspermine (3Q9E). The interactions of the side chains of Phe27, Tyr168, Phe225 and Ile29 in *Mr*ApaH (4) with the polyamine alkyl chain stabilize the L1-loop (Supplementary Fig. 19). Comparing the complex structures of *Mr*ApaH (4) with our apo structure of *Ps*ApaH (4), the Tyr of the conserved XGGY-motif (*Ps*ApaH (4): Tyr168: *Mr*ApaH (4): Tyr323) is rotated by app. 90° upon substrate binding, suggesting an induced fit mechanism (Fig. 7b). Tyr323 of *Mr*ApaH (4) forms a hydrogen bond with the $N^7$ of the acetylspermine (Fig. 7b). In our *Ps*ApaH (4) structure, Y168 exists in two alternative conformations supporting the flexibility in the unliganded state (Fig. 7b). Secondly, selectivity toward polyamines is created by E106 located at the

C-terminal end of the L2-loop (PSL) in *Mr*ApaH (4), forming a salt bridge in trans with the $N^1$-amino group of acetylspermine or $N^4$ of $N^8$-acetylspermidine[117]. In *Ps*ApaH (4), this Glu106 is replaced by His106 explaining the higher acetylpolyamine substrate promiscuity for *Ps*ApaH (4) (Supplementary Fig. 20). The sequences of the L2-loop and the N-terminal loop vary considerably in cluster 4 enzymes, also contributing to different deacetylation efficiencies toward acetylpolyamines. Thirdly, the presence of the Glu gatekeeper in the 271-PEG-273 motif defined for *Hs*HDAC10 (1g) conveys a preference for polyamines and creates selectivity toward $N^8$-acetylspermidine by electrostatically interfering with $N^4$ of acetylspermidine (Fig. 7c)[94,103,169]. *Hs*HDAC10 (1g) was reported to act as polyamine deacetylase with a preference to deacetylate $N^8$-acetylspermidine over $N^1$-acetylcadaverine and $N^1$-acetylputrescine but not capable of deacetylating neither $N^1$-acetylspermidine nor acetylspermine[94]. This specificity is mediated by the surface electrostatics at the entrance of the active site and at the base of the active site (Fig. 7d)[95]. The active site is highly negatively charged supporting the binding of polycations such as polyamines. The glutamate gatekeeper in the PEG-motif (*Hs*HDAC10: 271-PEG-273) contributes to this electrostatic profile interacting with the positively charged amino groups at $N^4$ of $N^8$-acetylspermidine/$N^1$ of acetylspermine (Fig. 7c)[170]. This interaction was shown to increase the efficiency of polyamine deacetylase activity over lysine deacetylase activity and to mediate selectivity of $N^8$-acetylspermidine over $N^1$-acetylspermidine (Fig. 7a)[95]. In contrast, *Mr*ApaH (4) lacks the Glu gatekeeper (290-PIS-292) and has a broader substrate specificity (Fig. 7c). *Ps*ApaH (4) and the *P. aeruginosa* enzymes PA1409 and PA0321 also contain the PIS-motif (Fig. 7c; Supplementary Fig. 1c; Supplementary Fig. 19; Supplementary Data 6), which explains the higher level of promiscuity observed for bacterial polyamine deacetylases (Fig. 2d and 7c). Since substitution of the Glu gatekeeper by Leu in *Hs*HDAC10 (1g) resulted in a remarkable two orders of magnitude increase in the lysine deacetylase activity, with a concomitant decrease in its polyamine deacetylase activity, our structures further explain the dual function of bacterial deacylases as polyamine and lysine deacetylases[94,105,124]. This is in good agreement with our data showing *Ps*ApaH (4) with PIS-motif having both, and polyamine deacetylase and lysine deacetylase activity. For *Ps*ApaH (4) peptides with positively charged residues in −1 position are preferred substrates (Fig. 3a). In fact, the peptide with negatively charged Asp at −1 is not deacetylated at all and the peptide with Gly at −1 is weakly deacetylated (Fig. 3a). This is in good agreement with the highly negatively charged active site. We also identified *Ps*ApaH (4) as active in deacetylating Lys549-acetylated *Bs*AcsA suggesting that it is also competent to deacetylate proteins apart from the Fluor-de-Lys peptides tested here (Supplementary Fig. 17). For HDAC10, the presence of a $3_{10}$-helix in the N-terminus with the sequence P(E/A)CE (*Dr*HDAC10: 23-PECE-26; *Hs*HDAC10: 21-PECE-23) was suggested to mediate substrate specificity by restricting active site access[94,95,103,171]. This sequence motif and the $3_{10}$-helix is missing in

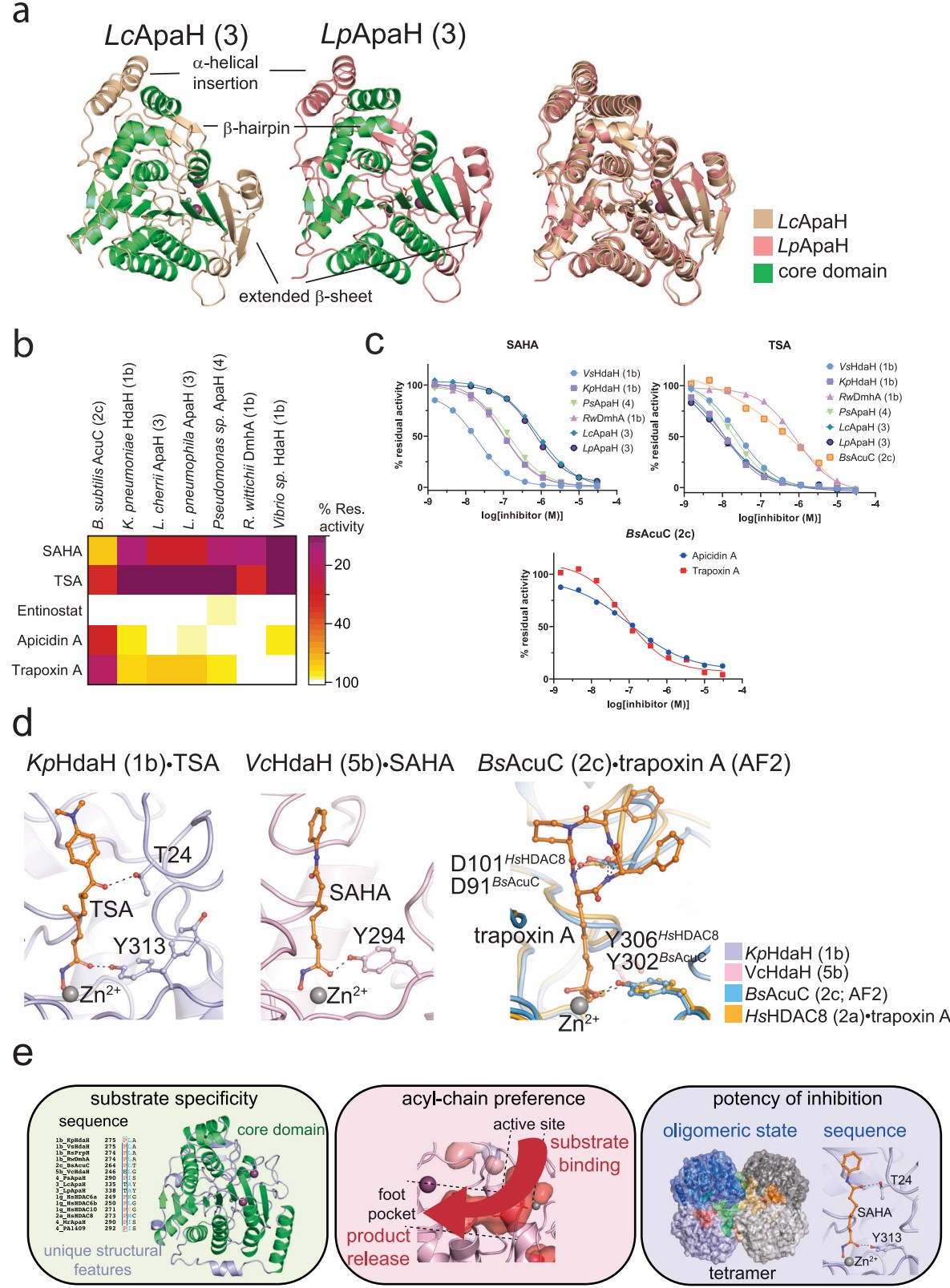

bacterial polyamine deacetylases, and substrate access is instead restricted by head-to-head dimer formation and presence of the L1- and L2-loops as stated above (Fig. 7b).

Notably, the bacterial polyamine deacetylases also contain a conserved RPP-motif (*Ps*ApaH (4)/*M. ramosa* ApaH: 154-RPPGHH-159) and an XGGY-motif (*Ps*ApaH (4)/*M. ramosa* ApaH: 320-EGGY−323), identified for robust deacetylases such as *Kp*HdaH (1b) but not present

in deacylases with activity for longer acyl-chains, suggesting these enzymes prefer to remove acetyl-groups from polyamines but no longer acyl-chains (Supplementary Fig. 1). Apart from the polyamines studied here, further polyamines are described such as thermospermine, caldopentamine and caldohexamine. Further studies are needed to characterize deacetylation of these polyamines by polyamine deacetylases.

**Fig. 8 | Structural features of cluster 3 containing *Legionella* virulence factors and inhibition of bacterial deacylases by hydroxamate inhibitors and cyclic peptides. a** The cluster 3 enzymes *Lc*ApaH (3) and *Lp*ApaH (3) are bacterial virulence factors. Structurally the cluster 3 enzymes contain two distinct features. The eight-stranded parallel β-sheet is extended by two antiparallel β-strands (*Lc*ApaH (3): brown; *Lp*ApaH (3): red) and there are two additional α-helices in the catalytic core domain. **b** Heatmap of the pre-screening of selected bacterial deacylases by the hydroxamate inhibitors SAHA and trichostatin A (TSA), the benzamide inhibitor entinostat (MS-275), and the cyclic peptides apicidin A and trapoxin A. The color code represents the residual activity after 1 h incubation at 37 °C. The experiments were performed in two independent replicates ($n = 2$). Shown are the means. Source data are provided as Source Data file. **c** Dose-response curves obtained for inhibition of bacterial deacylases with the hydroxamate inhibitors SAHA, TSA, and the cyclic peptides apicidin A and trapoxin A. The experiments were performed in two independent replicates ($n = 2$). Shown are the means. Source data are provided

as Source Data file. **d** The hydroxamate inhibitors SAHA and/or TSA use an identical inhibitory mechanism for bacterial deacylases and for mammalian HDACs. For *Bs*AcuC (2c) an AlphaFold2 model was created to show interaction with trapoxin A by superposition with the structure of *Hs*HDAC8 (2a)•trapoxin A (PDB: 5VI6). For binding of the cyclic inhibitors formation of hydrogen bonds between the main-chain amide of the cyclic peptides and the side chain of a conserved Asp (*Bs*AcuC (2c): Asp91; *Hs*HDAC2 (2a): Asp100; *Hs*HDAC8 (2a): Asp101) is important. **e** Model of the molecular mechanisms underlying the observed substrate and acyl-chain type preferences of bacterial deacylases. Bacterial deacylases apply three major mechanisms for their activity and for the determination of substrate specificity. Left: Oligomerisation determines the accessibility of substrates to the active site. Middle: The architecture of the foot pocket determines acyl-chain preference. Right: Differences in the sequences and presence of additional structural features determine substrate promiscuity, acyl-chain preference, and the inhibitory potency by different classes of inhibitors.

## Cluster 3 encompasses *Legionella* virulence factors

We were able to solve two crystal structure for two *Legionella* enzymes of cluster 3, *L. cherrii* ApaH and *L. pneumophila* ApaH (Fig. 8a; Supplementary Fig. 18). For the homologous *L. pneumophila* enzymes LphD and Smh1, it was reported recently these being para-effectors important for *L. pneumophila* virulence acting as histone deacetylase with preference to deacetylate histone H3K14 (10-STGG(KAc)APRK-18)[139,140]. LphD was furthermore shown to directly interact with the H3K14 methyltransferase RomA and to be present in a complex with the lysine acetyltransferase KAT7[139]. To this end, LphD and RomA were shown to translocate into the nucleus where they act as epigenetic modulators adjusting gene expression of the host cell to allow an efficient infection process[139]. We show that both *Legionella* enzymes studied here are capable to act as efficient deacylases for histone H4 (LGK$_{acyl}$, H4$_{10-12}$) and p53 (QPKK$_{acyl}$, p53$_{317-320}$) peptides. For the histone H4 (LGK$_{acyl}$, H4$_{10-12}$) the $K_M$ values are more than 50-fold lower compared to the p53 (QPKK$_{acyl}$, p53$_{317-320}$) peptide (Table 1; Fig. 3b,c,e). As histone tails are unstructured the presence of a Gly-Lys dipeptide motif in the amino acid sequence is a major determinant for

substrate selectivity. For the H3H14 peptide an $K_M$ value of 95 µM was reported. For the histone H4 (LGK$_{acyl}$, H4$_{10-12}$) peptide we determined $K_M$ values of 15 µM and 21 µM for *L. pneumophila* and *L. cherrii* ApaH (3), respectively. Moreover, both also act as depropionylase and de-D-/L-lactylase suggesting these enzymes can remove other acylations than acetyl-groups from lysine side chains and have more physiologically important substrates than H3K14. Along that line, we observed that both *Legionella* enzymes are capable to deacetylate *B. subtilis* AcsA, acetylated at a lysine residue within a sequence motif (546-RSG(KAc) IMR-552) strictly conserved in mammalian acetyl-CoA synthetase Acss2 (cytosolic/nuclear; RSG(KAc)IMR). In *Lc*ApaH (3) and *Lp*ApaH (3) the XGGY-motif is conserved (*Lc*ApaH: 388-EGGY−391/*Lp*ApaH: 391-EGGY −394). The RPP-motif is substituted by the motif 176-GLPSHH-181 (*Lc*ApaH)/176-GLPSHH-181 (*Lp*ApaH), however, with highly similar main-chain traces (Fig. 5e; Supplementary Fig. 21). To this end, differences in the sequences in the region of the RPP-motif might explain the observed highest activity as deacetylase while also being active as depropionylase and de-D-/L-lactylase (Fig. 3a−c,e; Supplementary Fig. 21). Structurally, compared to the arginase-deacetylase core, *Lp*ApaH (3) and *Lc*ApaH (3) contain an additional pair of α-helices and the central parallel β-sheet is extended by a two-stranded antiparallel β-sheet (Fig. 8a). These features might be needed for recognition of the enzymes by the type IV secretion system for secretion into the host cell, or important for interaction with other proteins such as the methyltransferase RomA or the lysine acetyltransferase KAT7 in the host cells[139], which calls for further investigations. To understand whether these bacterial classical deacylases with different oligomeric states, substrate specificities and acyl-chain preferences can be inhibited by known mammalian classical HDAC inhibitors, we next performed inhibition studies to evaluate potential drug-repurposing strategies to fight bacterial infections.

## Table 2 | Inhibition of selected bacterial deacylases by the hydroxamate inhibitors SAHA and TSA, the benzamide inhibitor etinostat (MS-275) and the cyclic peptides trapoxin A and apicidin A

| Enzyme | Inhibitor | Log[IC$_{50}$(M)] | IC$_{50}$ (nM) | $K_i$ (nM)[a] |
|---|---|---|---|---|
| *Bs*AcuC (2c) | TSA | −6.2 | 593 | 400 |
| | Apicidin A | −7.0[b] | ≤107 | ≤72 |
| | Trapoxin A | −7.1[b] | ≤80 | ≤54 |
| *Kp*HdaH (1b) | SAHA | −7.01 | 97.1 | 22.4 |
| | TSA | −7.97[b] | ≤10.7 | ≤2.47 |
| *Lc*ApaH (3) | SAHA | −6.09 | 809 | 307 |
| | TSA | −7.98 | 10.4 | 3.95 |
| *Lp*ApaH (3) | SAHA | −6.18 | 669 | 230 |
| | TSA | −8.1 | 7.17 | 2.47 |
| *Ps*ApaH (4) | SAHA | −6.84 | 143 | 59.7 |
| | TSA | −7.74[b] | ≤18.1 | ≤7.56 |
| *Rw*DmhA (1b) | SAHA | −7.00 | 101 | 41.6 |
| | TSA | −6.06 | 865 | 356 |
| *Vs*HdaH (1b) | SAHA | −7.69 | 20.3 | 8.90 |
| | TSA | −7.60 | 25.4 | 11.1 |

Shown are the enzymes, the inhibitors used and the results obtained for log[IC$_{50}$], the IC$_{50}$ and $K_i$. As the deacylase activity for *Vc*HdaH (5b) was low, we could not analyze the inhibition. The experiments were performed in two independent replicates ($n = 2$). Data are presented as means. Source data are provided as Source Data file.
[a]$K_i$ data calculated using the Cheng–Prusoff equation.
[b]Calculated IC$_{50}$ close to the concentration of enzyme in the assay (i.e., stoichiometric inhibition).

## Bacterial DACs are inhibited by eukaryotic HDAC inhibitors

In an initial screen, we analyzed the inhibitory potency of the known class I/class II HDAC inhibitors suberoylanilide hydroxamic acid (SAHA) and trichostatin A (TSA), the class I benzamide inhibitor entinostat (MS-275) and the cyclic peptides apicidin A and trapoxin A, for their potency to inhibit the bacterial deacylases (Table 2; Fig. 8b; Supplementary Figs 22 and 23; Supplementary Data 8). The initial screen revealed inhibition of all bacterial deacylases by TSA and SAHA, albeit with different potencies (Table 2; Supplementary Fig. 23). Notably, the pre-screening suggests that *Bs*AcuC (2c) is only weakly inhibited by the hydroxamates (TSA and SAHA) compared to other bacterial deacylases (Table 2; Fig. 8c; Supplementary Fig. 23). Instead, *Bs*AcuC (2c) was the only deacylase being inhibited by the cyclic peptides apicidin A and trapoxin A, which selectively inhibit mammalian class I and class IV HDACs[172–175]. None of the bacterial deacylases were inhibited by the class I benzamide inhibitor entinostat (Table 2; Fig. 8b; Supplementary Figs 22 and 23). Notably, it was reported that

*Hs*HDAC11 (5f) possesses a similar inhibition profile as we observed here for *Bs*AcuC (2c). *Hs*HDAC11 (5f) was reported to be inhibited by trapoxin A and analogs thereof in the sub-micromolar to the nanomolar range while being only weakly inhibited by SAHA and/or TSA[96,173]. The lower potency of inhibition (in sub-micromolar range) for trapoxin A against human *Hs*HDAC11 (5f) compared to class I human HDACs is due to the lack of an aspartic acid residue [Asp101 in *Hs*HDAC8 (2a); Asp100 in *Hs*HDAC2 (2a); Asp91 in *Bs*AcuC (2b)], which was shown for *Hs*HDAC8 (2a) to be essential for inhibition by trapoxin A and apicidin A (Fig. 8d; Supplementary Fig. 24)[176,177].

We were not able to perform inhibitor studies with *Vc*HdaH (5b) because the observed de-decanoylase activity was neither strong enough to perform Michaelis–Menten kinetics nor to determine $K_i$/ $IC_{50}$ values (Supplementary Fig. 6). Whether *Vc*HdaH (5b) is inhibited by the HDAC11-selective trapoxin A analog TD034[173] or recently reported *Hs*HDAC11-selective hydroxamates needs further investigation[178,179].

We performed dose-response inhibition assays to determine $IC_{50}$-values and inhibitory constants, $K_i$ values, for all enzyme-inhibitor combinations of which we observed inhibition in the pre-screening (Table 2; Fig. 8b,c; Supplementary Figs. 22 and 23). $K_i$ values were determined by applying the Cheng–Prusoff equation as a model to fit the data. For *Bs*AcuC (2c), we determined $K_i$ values for TSA of 400 nM, while not detecting any inhibition by SAHA (Table 2). Inhibition of *Bs*AcuC (2c) by apicidin A and trapoxin A was more potent with $K_i$ values of 72 nM and 54 nM, respectively (Table 2; Fig. 8c). This is in agreement with the mammalian class I HDACs shown to be potently inhibited by apicidin A and trapoxin A[172,173].

For *Rw*DmhA (1b), we obtained $K_i$ values of 356 nM and 41.6 nM for TSA and SAHA, respectively (Table 2; Fig. 8c). *Vs*HdaH (1b) was potently inhibited by both hydroxamates, SAHA and TSA ($K_i$ (TSA): 11.1 nM and $K_i$ (SAHA): 8.9 nM) (Table 2; Fig. 8c). All other enzymes were potently inhibited by TSA with $K_i$ values of <10 nM, while $K_i$ values for SAHA were 5–10-fold higher (Table 2; Fig. 8c). These data show that the archetypical inhibitors for eukaryotic classical deacylases were similarly potent and selective for inhibition of bacterial enzymes.

This strong inhibitory potential of SAHA and TSA fed our idea to apply these inhibitors in a drug-repurposing strategy to fight pathogenic bacteria. For *L. pneumoniae*, the deacylase LphD was recently shown to constitute a para-effector supporting virulence[139,140]. We tested this hypothesis for the *K. pneumoniae* deacetylase *Kp*HdaH (1b) by testing the impact of presence *Kp*HdaH (1b) on bacterial growth in complex and minimal medium, on biofilm formation and on its virulence by performing infection experiments with C57ZBL/6 mice and by genomically deleting the *hdaH* gene in the *K. pneumoniae* B5055 strain (Supplementary Fig. 25). The data showed a slightly improved growth of *K. pneumoniae* B5055 Δ*hdaH* in minimal medium supplemented with glucose and acetate as carbon sources. Genomic deletion of *hdaH* did not affect growth in complex medium or growth with glucose or acetate as sole carbon source. Further, neither biofilm formation nor *K. pneumoniae* B5055 virulence were affected in infection experiments with C57ZBL/6 mice (Supplementary Fig. 25). Thus, further studies are required to determine if other $Zn^{2+}$-dependent bacterial deacylases are candidates for development of anti-microbial therapeutic strategies. Next, we wondered whether similar mechanistic similarities of these inhibitors exist between the inhibition of bacterial deacylases the mammalian enzymes. To address this question, we performed structural analyses to characterize various enzyme-inhibitor complexes.

## Mechanisms underlying inhibition of bacterial DACs

To show how the hydroxamate inhibitors TSA and SAHA as well as the cyclic peptide inhibitors apicidin A and trapoxin A inhibit bacterial deacylases, whether they apply similar molecular mechanisms to inhibit deacylase activity compared to mammalian HDACs and to understand differences in the observed potencies of inhibition of

different bacterial deacylases, we performed structural analyses. We solved several X-ray crystal structures of bacterial deacylases in complexes with TSA and/or SAHA (Fig. 8d; Supplementary Figs. 26–28; Supplementary Tables 5–10). For the cyclic peptide inhibitors, we used the AlphaFold2 model of *Bs*AcuC (2c) and superimposed this with the complexes solved earlier of *Hs*HDAC2·apicidin A (PDB: 7LTG) and *Hs*HDAC8·trapoxin A (PDB: 5VI6) (Fig. 8d; Supplementary Fig. 23a, b). The enzyme *Kp*HdaH (1b) was soaked with SAHA and TSA (Fig. 8d; Supplementary Figs. 26–28). We solved the X-ray crystal structures of *Kp*HdaH (1b), *Rw*DmhA (1b), and *Vc*HdaH (5b) in complexes with SAHA to a resolution of 1.95 Å, 1.75 Å, and 1.64 Å, respectively (Fig. 8d; Supplementary Figs. 26–28; Supplementary Tables 6–8). Moreover, we also successfully solved the structure of *Kp*HdaH·TSA to a resolution of 2.18 Å (Fig. 8d; Supplementary Figs. 26–28; Supplementary Table 6).

As described for mammalian $Zn^{2+}$-dependent HDACs, the hydroxamates coordinated the catalytic $Zn^{2+}$-ion in a bidentate fashion, the carbonyl oxygen replacing the carbonyl oxygen of the acetamide in the substrate and the hydroxyl group replacing the position of the catalytic water molecule (Fig. 8d; Supplementary Fig. 26b).

For all cluster 1 enzymes, we observed a potent inhibition by SAHA, and in most cases even higher by TSA (Table 2). *Kp*HdaH (1b) forms a hydrogen bond with the side chain of Thr24 of the L1-loop with the carbonyl/amide oxygen present in the linker of TSA/SAHA connecting the head group of SAHA and TSA and the hydoxamate warhead. Thr24 contributes to the observed high potency observed for SAHA and TSA to inhibit *Kp*HdaH (1b) activity (Fig. 8c; Supplementary Fig. 26a, b). This Thr24 is not conserved in bacterial deacylases, exemplifying differences in substrate specificity or inhibition due to differences in structure and sequence of bacterial deacylases (Supplementary Fig. 1). Inspection of the structures of the *Kp*HDAH (1b) ·SAHA, *Kp*HDAH (1b)·TSA and *Rw*DmhA (1b)·SAHA showed that the binding site of the inhibitors is created by part of the L1-loop supplied *in cis*, and by the C-terminal α-helix supplied in trans, i.e. by another monomer of the oligomer (Supplementary Fig. 24a). The lack of polar interactions such as hydrogen bonds with the inhibitor linker parts suggest that the differences in potency between SAHA and TSA might derive from van-der-Waals interactions or entropic effects. The unsaturated TSA alkyl chain has a lower conformational flexibility compared to SAHA, which might explain a different entropic contribution to the binding. Moreover, a more potent inhibitory activity for TSA can be expected if the binding site is primed for TSA binding. *Rw*DmhA (1b) was almost ten-fold less potently inhibited by TSA compared to SAHA (Table 2). *Kp*HdaH (1b) inhibition followed the opposite trend, and *Vs*HdaH (1b) was inhibited equally by both inhibitors (Table 2). To understand these differences in the inhibitory potencies, we superimposed the structures of the bacterial deacylases with structures solved in complexes with SAHA/TSA to model the binding (Supplementary Fig. 24a). Comparing the hydrophobicity of SAHA and TSA showed that SAHA is more hydrophobic than TSA (Lipinski hydrophobicity index: SAHA 2.31 ± 0.21; TSA 1.82 ± 0.57), as determined by SciFinder[n]. A major difference in the inhibitor binding area is the hydrophobicity of a sequence motif supplied in trans to complete the active site entry site. The hydrophobic sequence motif 345-AW-346 in *Rw*DmhA (1b) is substituted by the less hydrophobic sequence motif 356-AQ-357 in *Vs*HdaH (1b) and the hydrophilic sequence motif 346-QQ-347 in *Kp*HdaH (1b) (Supplementary Fig. 24a). Notably, in *Kp*HdaH (1b) Phe343 (*Rw*DmhA (1b): Met342; *Vs*HdaH (1b): Ile354) is closer to the aromatic cap of TSA compared to SAHA explaining the higher inhibitory potency observed for TSA as described also for *P. aeruginosa* PA3774[137]. This variation in the hydrophobicity might explain the differences observed for potency of inhibition by SAHA and TSA in sub-cluster 1b enzymes.

Interestingly, we observed SAHA and TSA only weakly inhibiting *Bs*AcuC (2c). We analyzed the underlying mechanism explaining the observed low potency of the hydroxamates TSA and SAHA to inhibit

*Bs*AcuC (2c). Superposition of the AlphaFold2 structure of *Bs*AcuC (2c) with the structures of *Kp*HdaH (1b) solved in complexes with TSA and SAHA shows the monomeric *Bs*AcuC (2c) lacks the hydrophobic binding site for the TSA/SAHA head group formed by the extended L1-loop and the C-terminal helix in the oligomers of cluster 1 and cluster 4 enzymes (Supplementary Fig. 24a, b). Asp91 of *Bs*AcuC (2c) needed for binding to apicidin A and trapoxin A is too far away to interact with SAHA/TSA (Supplementary Fig. 24b).

Notably, for the cluster 3 enzymes *Lp*ApaH (3) and *Lc*ApaH (3) we discovered the highest discrepancy in the inhibitory potency between SAHA and TSA, with TSA inhibiting the enzymes almost 80–95-fold stronger (Table 2). The structures show an open active site entry due to absence of oligomer formation and, as a consequence, the missing in trans supply of the active site lid as observed for cluster 1 and cluster 4 enzymes (Supplementary Fig. 24a). The fact that TSA still shows a low nanomolar inhibitory potency might be due to the differences in the conformational flexibility of the unsaturated TSA compared to the saturated SAHA alkyl chain, resulting in an entropically favored binding of TSA.

For all structures, we confirmed the binding of the inhibitors within the substrate binding tunnel with the catalytically important Y of the XGGY-motif forming a hydrogen bond to the carbonyl oxygen of the hydroxamate group (Fig. 8d; Supplementary Fig. 26a, b). The structural model of *Bs*AcuC (2c) prepared in complexes with apicidin A or trapoxin A suggested formation of a hydrogen bond to Tyr302 of *Bs*AcuC (2c) with the gem-diolate group of the apicidin A and trapoxin A, respectively (Fig. 8d; Supplementary Fig. 26b). The only enzyme inhibited by cyclic peptides trapoxin A and apicidin A was *Bs*AcuC (2c). Trapoxin A and apicidin A were reported to be potent inhibitors for class I HDACs and a sub-micromolar inhibitor of the class IV enzyme HDAC11 (5f)[172,173,176,177]. *Hs*HDAC11 (5f) was shown to be reversibly inhibited by the trapoxin A analog TD034, while for the class I enzyme HDAC8 trapoxin A acts as an irreversible nanomolar inhibitor, although not forming a covalent enzyme-inhibitor complex with its epoxide moiety[172,173]. Mechanistically, the epoxide moiety undergoes nucleophilic attack by a water molecule resulting in a gem-diolate coordinating the catalytic $Zn^{2+}$ ion mimicking the tetrahedral intermediate and the transition states resulting in formation and breakdown of this intermediate[173].

A conserved acidic residue [Asp100 in *Hs*HDAC2 (2a) or Asp101 in *Hs*HDAC8] at the active site pocket surface, which contributes to substrate binding, is a prerequisite for the inhibition by the cyclic peptide inhibitors, as mutation of Asp101 to A in *Hs*HDAC8 abolishes inhibition by trapoxin A[172,174,175,177,180]. This Asp/Glu side chain forms important hydrogen bonds with the peptide bonds of the substrate or inhibitors (Fig. 8d; Supplementary Fig. 26b)[161,172,173,177,180]. *Bs*AcuC (2c) contains Asp91 at the analogous position, suggesting a similar mechanism for the inhibition by trapoxin A and apicidin A (Fig. 8d; Supplementary Fig. 1). However, the presence of a negatively charged residue at this position is not sufficient for cyclic peptide inhibition. In fact, we found that Asp/Glu is conserved in members of all main clusters (*Kp*HDAH (1b): Glu101; *Bs*AcuC (2c): Asp91; *Lp*ApaH (3): Asp146; *Lc*ApaH (3): Asp144; *Hs*HDAC2: Asp100; *Hs*HDAC8: Asp101), except from the polyamine deacylases of clusters 4 and 5 (*Ps*ApaH (4): Gly117; *Vc*HdaH (5b): Gly97), and *Bs*AcuC (2c) was the only of these bacterial enzymes that was potently inhibited by trapoxin A or apicidin A (Fig. 8a; Supplementary Fig. 1; Supplementary Fig. 24c; Table 2). This shows further mechanisms must exist apart from presence of the D/E needed to convey inhibition by the cyclic peptide inhibitors. Next to the Asp/Glu-residue needed for trapoxin A/apicidin A binding, analyses of crystal structures suggested that the oligomeric bacterial deacylases of cluster 1b and 4 would not allow binding of the cyclic peptides due to steric clashes with the L1-loop in sub-cluster 1b enzymes, or by monomer 2 in cluster 4 enzymes, respectively (Supplementary Fig. 24d). Overall, our structural data complemented with the

inhibitory profiles provide evidence that mammalian HDAC inhibitors are potent and potentially selective for inhibition of bacterial classical deacylases.

## Discussion

In this study, we performed comprehensive structure-function analyses on bacterial classical deacylases. We performed a bioinformatics analysis to identify thousands of potential bacterial classical deacylases by applying a Generalized Profile (GP) to screen the UniProt database. These bacterial deacylases were classified into five clusters (clusters 1–5), some with several sub-clusters. The mammalian class I classical $Zn^{2+}$ dependent deacylases group in cluster 2 and class II deacetylases in cluster 1. All bacterial cluster 1 enzymes contain the catalytic tyrosine used to classify mammalian deacylases into class IIb. We identified no bacterial enzyme with this tyrosine (HDAC8: Tyr306) replaced by histidine, which would classify them into class IIa and resulting in low deacylase activity[181]. The presence of additional clusters of enzymes not containing any mammalian HDAC suggest presence of enzymes in bacteria being different in their sequences, structural features and/or activities, i.e. substrate specificities and acyl-chain preferences (clusters 3 and 4). Phylogenetically, mammalian and plant classical deacylases are present in clusters 1, 2 and 5, while cluster 3 and 4 only contain bacterial deacylases (Fig. 1b; Supplementary Data 1–3). This suggests that some activities and specificities were independently developed during evolution in a convergent evolution, i.e. bacterial polyamine deacetylases present in cluster 4 and mammalian polyamine deacetylase *Hs*HDAC10 (1g) in cluster 1.

An important molecular mechanism determining substrate specificity of mammalian classical HDACs is the formation of protein complexes[114]. Except for HDAC8, all class I enzymes were shown to be part of multi-protein complexes, the so-called co-repressor complexes involved in transcriptional repression[72,74,182]. Often, these complexes contain several HDAC domains, their presence in complexes stimulates their deac(et)ylase activity[183,184]. For mammalian HDACs, it was postulated that the distance and the relative orientation of the HDAC catalytic sites determine specificity of the deac(et)ylase activity targeting different conformations on chromatin. The activities of some HDACs within the complexes were shown to be regulated by inositol phosphates second messengers not present in bacteria[72,75]. However, a regulation by inositol phosphates is imaginable for bacterial decylases acting as virulence factors as they are secreted into host cells. Also, class IIa and class IIb HDACs were part of protein complexes and many protein-protein interactions were reported[182,185–187]. For the class IV enzyme HDAC11, the presence in multi-protein complexes could not been shown so far. Interactions and formation of multi-protein complexes are often mediated via the HDAC catalytic core domains but also via additional regions N- and/or C-terminal to the catalytic HDAC domains[188]. Our data on the bacterial enzymes indicated cluster 1 enzymes containing an extended L1-loop can form tetramers and the polyamine deacetylases of cluster 4 form dimers in solution. These interactions are mediated directly via the catalytic domains arranged in a head-to-head orientation and by additional loop regions and sometimes additional secondary structure elements as observed for cluster 1b enzymes. Our structural data suggest this oligomerization being of functional importance to determine substrate specificity and catalytic activity as the formation of an oligomer restricts access of substrate to the active site. Although we could not detect it with our reported substrates, for cluster 1b enzymes the position of the L1-loop of each monomer, as a molecular glue, mediating the contact to all remaining three molecules within the tetramer and also interacting with the substrate, might open the possibility of cooperativity of enzyme activity. Further studies in vitro and in vivo will reveal if this is the case. We provide evidence that deletion of a part of the L1-loop abolishes the integrity of the side-by-side dimer while leaving the head-to-head dimer intact. We show the individual head-to-head dimer still

being active, albeit with reduced activity compared to the tetramer showing tetramer formation is important for full catalytic activity. For the cluster 3 enzyme LphD, a para-effector affecting the virulence of *L. pneumophila*, a direct interaction with the *L. pneumophila* methyltransferase RomA and with the lysine acetyltransferase KAT7 in the host cell was experimentally validated[139]. Our structural analysis on the related enzymes *Lp*ApaH (3) and *Lc*ApaH (3) revealed presence of an additional insertion of two α-helices and by two antiparallel β-strands extending the central β-sheet of the catalytic core. These additional structural features might mediate these interactions, i.e. interaction with the secretion machinery or interactions with RomA and/or KAT7. Future studies are needed to show if and to which extend bacterial classical deacylases are part of protein complexes and if they have further interaction partners important for modulating catalytic activity or substrate specificity.

We were able to solve a total of twelve crystal structures of bacterial deacylases in their apo form and in complexes with hydroxamate inhibitors SAHA and/or TSA (Supplementary Tables 5–10). We observed the binding of the reaction product acetate in the foot pocket in several crystal structures, suggesting that these structures present the state after catalysis. We identified *Vc*HdaH (5b) being an active de-decanoylase. For *Rw*DmhA (1b), we discovered it being an active lysine deacylase, which might suggest that this 1b enzyme is capable to act as lysine deacylase additionally to the reported dimethoate hydrolase activity[146].

The structure-function analyses performed here suggest different molecular mechanisms exist in bacterial deacylases to create substrate specificity and acyl-chain preference: 1. Oligomerization, i.e. formation of head-to-head oligomers to restrict access of substrates to the active site, 2. Distinct structural features, i.e. different N- and C-terminal extensions to create substrate specificity, 3. Conserved sequence motifs, i.e. sequences such as the RPP-motif lining the foot pocket mediating acyl-chain preference or the PEG/PKG/PIS sequence determining substrate specificity (PEG: polyamine deacetylase activity of *Hs*HDAC10 (1g); PKG: activity for C-terminal acetyl-lysines of catalytic domain 1 (CD1) of *Hs*HDAC6 (1g); PIS: dual activity as polyamine and lysine deacetylase in bacterial ApaHs (4)) and 4. The surface electrostatics, i.e. determining the substrate sequence preference, such as *Kp*HdaH (1b) with an acidic surface potential showing preference for positively charged residues at −1 position.

For *Rw*DmhA (1b) and *Rs*PrpH (1b), we characterized two examples of deacetylases that divergently evolved from protein lysine deacetylases to enzymes with substrate specificity for small molecules, i.e. dimethoate and propanil. For *Rw*DmhA (1b), we identified an all-*trans*-octanoic acid in the crystal structure, suggesting *Rw*DmhA (1b) being capable of releasing this fatty acid upon deacylation from their substrate. As a consequence, we propose *Rw*DmhA (1b) having additional yet undefined substrates apart from dimethoate. We and others provide structural evidence differences in the sequences of the L1-loops and in the amino acid composition lining the tunnel leading to the active site being the major factor for determining substrate specificity for cluster 1b enzymes. We identified another structural feature for the cluster 1b enzymes, i.e. a C-terminal α-helix, mediating contacts to the other monomers structurally contributing to tetramer formation. We observed an efficient deacetylation of the H4-derived peptide LGK$_{ac}$ by the cluster 1b lysine deacetylases. However, the K549-acetylated AcsA protein of *B. subtilis* with the conserved sequence (546-RSG(KAc)IMR-552) was not deacetylated suggesting the substrate specificity of 1b enzymes depend also on the substrate amino acid sequence, of residues C-terminal of the acetyl-lysine and maybe on the three-dimensional structure of the substrates.

With *B. subtilis* AcuC of sub-cluster 2c, *Bs*AcuC (2c), we characterized an enzyme with a very narrow substrate specificity[126]. At first glance, this enzyme shows a quite promiscuous activity in deacylating various peptide sequences and different acyl-chains. However, when

analyzing the enzymatic specificity constants, $k_{cat}/K_M$, it became obvious that this occurs at a rather low enzymatic specificity and efficiency. This might represent an example of the co-evolution of an enzyme, *Bs*AcuC (2c), with its substrate, *Bs*AcsA. *Bs*AcuC (2c) is encoded by the *acu*-operon, which gene products are essential for regulation of AMP-forming acetyl-CoA synthetase (AcsA). AcsA is known to be regulated by lysine acetylation at a C-terminal lysine side chain present in a sequence (546-RSG(KAc)IMR-552) highly conserved in enzymes of the ANL family[126,150]. Our studies show AcuC being a monomer in solution. Whether AcuC has further interaction partners or binds to other regulators needs further investigation.

For cluster 5, the enzyme of *Vibrio cholerae* HdaH of sub-cluster 5b, *Vc*HdaH (5b), was a monomer in solution, as confirmed by the crystal structure and by analytical size-exclusion chromatography. We detected its unprecedented and selective activity as lysine de-decanoylase. In contrast to the cluster 1b enzymes, *Vc*HdaH (5b) favors a negatively charged residue at −1 position while not deacylating peptides with a positively charged residue at this position. This data was supported by structural analyses, revealing the presence of an all-*trans*-decanoic acid in the active site. As we did not supply it in the crystallization condition this fatty acid must have derived from the expression in *E. coli*. Moreover, the fatty acid binds with a rather high occupancy, suggesting the activity of de-decanoylase might be of physiological importance. Future investigations will reveal the physiological substrates of *Vc*HdaH (5b). In mammals, acylation of lysines with longer acyl-chains as acetylation was reported to regulate subcellular localization of proteins and protein-protein interactions[89]. The possibility to remove these acyl-chain types from lysine side chains makes the reaction reversible comparable to lipidation on Cys side chains by thioester formation but in contrast to thioether formation on Cys side chains by prenylation[89]. Whether lysine acylation regulates the subcellular localization of proteins in bacteria needs further investigation.

Importantly, as observed for acetate in some structures (*Kp*HdaH (1b), *Rs*PrpH (1b), *Lp*ApaH (3), *Ps*ApaH (4)), in *Vc*HdaH (5b) we observed the decanoic acid to be bound in the foot pocket almost perpendicular to the orientation of the active site tunnel suggesting it as product of the deacylase reaction. Our structural data provide mechanistic data on how *Vc*HdaH (5b) can accommodate the decanoic acid product while other deacylases cannot. How the release of the decanoic acid into the solvent is achieved is not clear as the cavity is not solvent exposed in this state. For *Hs*HDAC8 it was proposed the deacetylation product acetate can exit the enzyme either using the acetyl-lysine substrate binding tunnel in the substrate unbound form or it is expelled via an internal cavity, the so-called foot pocket or acetate release channel[189]. As this internal cavity is not solvent exposed mechanisms must exist that open the cavity to the solvent such as conformational rearrangements upon substrate binding. Moreover, efficient substrate release for hydrophobic acyl-chain might depend on the medium into which the product is released, i.e. cytosol or lipid membrane. The occupation of the product release cavity might explain the low deacylase activity measured for *Vc*HdaH (5b) as the reaction cannot proceed when the product is not released. The de-decanoylase activity being multiplicatively higher in reality as the binding to the active site competes with the binding of the substrate. This example shows that the specificity of a deacylase does not only depend on the initial step, i.e. accommodation of the substrate and formation of the enzyme-substrate complex, but also on efficient product release.

The crystal structure of the bacterial polyamine deacetylase *Ps*ApaH (4) confirmed earlier studies suggesting polyamine-specificity is created by dimerization in a head-to-head arrangement and by the amino acid sequence supporting to form interactions to discriminate between different polyamines[117,190]. The dimerization is mediated structurally by the L2-loop (PSL) and by an N-terminal loop, the L1-loop, including a β-hairpin motif. This N-terminal loop adapts a stable

conformation only in the substrate-bound form as concluded from the *Pseudomonas* sp. ApaH apo structure solved here compared to the structures of *Mr*ApaH (4) in complexes with $N^8$-acetylspermidine (PDB: 3Q9C) and acetylspermine (PDB: 3Q9E). This supports an induced fit mechanism as the enzyme induces a conformation including the formation of a β-hairpin upon substrate binding rather than selecting a conformation preexisting without substrate, which must be a prerequisite for a conformational-selection binding mechanism[191]. In mammals, HDAC10 was recently identified as polyamine deacetylase. As we observe here for the bacterial polyamine deacetylases, substrate specificity was shown to be created by a restricted active site compared to lysine deacylases and an acidic surface at the edge of the active site promoting electrostatic steering of the polycationic polyamines. For *Hs*HDAC10 a Glu in a 273-PEG-275-motif was shown to act as a gatekeeper for polyamine selectivity electrostatically binding to the $N^4$-of acetylspermidine[94,103,170]. This is replaced by an Ile in the 290-PIS-292-motif in bacterial polyamine deacetylases explaining the higher degree of polyamine substrate selectivity observed for human *Hs*HDAC10. For the *Pseudomonas* sp. ApaH we observed that it can act as lysine deacylases for peptides having a positively charged residue at −1 position. Moreover, we also observed a deacetylase activity with the peptide containing a Gly-Lys(Ac) sequence supporting the activity observed in deacetylating acetylated *B. subtilis* AcsA. This is in agreement with data on *M. ramosa* ApaH being active against Boc-Lys(Ac)-AMC substrate next to its polyamine deacetylase activity[117]. As a support for the importance of the PEG/PIS-motif, the substitution of the gatekeeper Glu in 273-PEG-275 of *Hs*HDAC10 with Leu is very similar in its physicochemical properties to an Ile in the 290-PIS-292-motif of bacterial polyamine deacetylases, impaired its polyamine deacetylase activity but increased its deacetylase activity[94]. We observed the formation of a stable dimer for bacterial polyamine deacetylases. Head-to-head dimer formation via the PSL-loop (L2-loop) and the N-terminal loop (L1-loop) is an important molecular mechanism for bacterial deacylases to convey activity toward acetylpolyamines. *Hs*HDAC10 is structurally composed of an N-terminal catalytically active polyamine deacetylase domain and a C-terminal catalytically inactive pseudo-deacetylase domain. However, dimerization is not obviously essential for polyamine deacetylase activity as both domains are arranged in a tail-to-tail fashion[94,103,169]. However, besides the important sequence motifs described above, selectivity toward polyamines is also achieved by the presence of additional secondary structure elements not present in bacterial polyamine deacetylases such as a $3_{10}$-helix (25-CEI-27) in the N-terminal region forming part of the *Hs*HDAC10 active site[94,103,169]. The fact that two mechanistic strategies being realized in bacteria and eukaryotes to achieve efficient polyamine deacetylation suggests these mechanisms evolved by convergence.

Polyamine biosynthesis is evolutionary and highly conserved present in eukaryotes and bacteria. Moreover, except from the two orders of *Methanobacteriales* and *Halobacteriales* also archaea contain polyamines. Polyamines were shown to play diverse roles in cells, ranging from cell proliferation, gene expression, and cellular stress response. In bacteria, the role of polyamines on biofilm formation and biosynthesis of natural products was shown[134]. Deletion of polyamine levels was shown to have a negative impact on these processes. Polyamine acetylation is important to liberate the polycations from anionic storages and for interconversion of polyamines, extracellular transport, and turnover of polyamines. To this end, polyamine acetyltransferases developed to specifically acetylate polyamines[12]. We and other labs show bacteria and eukaryotes encoding for classical deacetylases competent to deacetylate polyamines. While *Hs*HDAC10 (1g) was shown to be a specific polyamine deacetylase with an almost undetectable lysine deacetylase activity, the bacterial polyamine deacetylases characterized so far apply a different molecular strategy to achieve activity as polyamine deacetylase. Moreover, the bacterial enzymes show a lower degree of polyamine subtype specificity and

they retain an activity to act as lysine deacetylase. This furthermore supports that eukaryotic and bacterial polyamine deacetylases evolutionary convergently developed independently after separation from their last common ancestor.

We were able to solve two crystal structures of cluster 3 enzymes, i.e. *Lp*ApaH (3) and *Lc*ApaH (3), which are closely related to the recently reported virulence factor LphD of *L. pneumophila*[139,140]. LphD was shown to bind to the methyltransferase RomA, as virulence factors both being secreted by the type IV secretion system into the host cells and targeting histone H3K14. Within the host cell, LphD and RomA interact with the lysine acetyltransferase KAT7[139,140]. We identified an additional pair of α-helices in the structures of *L. pneumophila* and *L. cherrii* ApaH (3) compared to other bacterial deacylases suggesting that these enable the interactions with RomA and/or KAT7 or they are needed for the interaction with the secretion machinery. This needs further investigation. The kinetic experiments suggest cluster 3 enzymes being capable of also acting as delactylases. We discovered both *Legionella* enzymes deacetylate AMP-forming acetyl-CoA deacetylase (AcsA) from *B. subtilis* within a highly conserved C-terminal sequence motif (546-RSG(KAc)IMR-552), which is also conserved in mammalian acetyl-CoA synthetases Acss2 (cytosolic/nuclear; RSG(KAc)IMR) and highly conserved in Acss1 (mitochondrial; RSG(KAc)VMR), supporting that the Gly-Lys dipeptide motif and presence in a structurally accessible region are both essential determinants mediating substrate specificity. This supports a model according to which *Legionella* ApaH/LphD/Smh1 are capable of deacetylating Acss2 in the nucleus/cytosol resulting in its activation and thereby generation of nuclear acetyl-CoA which can be used by KAT7 to acetylate histone H3K14 resulting in modulating of gene expression to promote an efficient infection.

Comparison of enzymes of all clusters suggests the RPP-motif being a major molecular determinant for deacetylase activity for either lysine side chains or polyamines and the potency the remove longer acyl-chains from lysines. All robust deacetylases show the presence of the RPP-motif within a conserved RPPXHH sequence in which the peptide bond preceding the first Pro residue is in *cis*-configuration resulting in a steric restriction of the foot pocket or product/acyl release channel only capable of accommodating acetate but no longer acyl-groups. In contrast, all lysine deacylases with the capability to remove longer acyl-chains from lysines have a conserved GGGXHH sequence replacing the RPP-motif containing sequence RPPXHH resulting in the opening of the foot pocket allowing to accommodate longer acyl-chains. Variations in the sequences in direct vicinity to this GGGXHH-motif might be responsible for mediating acyl-chain type specificity.

As a summary, we provide a detailed structure-function analyses on bacterial classical deacylases. We show classical deacylases being widely distributed among Gram-positive as well as Gram-negative bacteria. Our data suggest bacterial deacylases have developed in a divergent evolution process resulting in deacylases with specificity toward different substrates, either protein lysine deacylases with different acyl-chain preferences, including de-D-/L-lactylases, de-decanoylases, polyamine deacetylases or small molecule deacetylases. Enzyme substrate selectivity is achieved by different oligomeric states of the enzymes mediated by the presence of distinct structural features and by differences in the enzymes' amino acid sequence and by the surface electrostatics recognizing the substrates. We show classical deacylases, except for the cluster 2 enzyme *B. subtilis* AcuC (2c), can selectively and potently be inhibited by the mammalian HDAC hydroxamate inhibitors SAHA and/or TSA. *B. subtilis* AcuC (2c) is the only enzyme that can be inhibited by the benzamide inhibitor apicidin A and by the cyclic peptide inhibitor trapoxin A (Fig. 8e). Future studies will reveal whether bacterial deacylases are present in multi-protein complexes and regulated by post-translational modifications as described for the mammalian counterparts, their physiological

substrates and interaction partners. The presence of bacterial deacylases in several important human pathogens suggests the possibility of targeting these in a drug-repurposing strategy to develop therapeutics to fight bacterial infections.

# Methods

## Generalized profile of bacterial deacylases

A Generalized Profile (GP) was constructed from a multiple sequence alignment (MSA) of known and validated classical deacetylases. The MSA was generated with MAFFT (L-Ins-I modus). The GP was subsequently applied to screen the UniProt database[141,142]. Beginning with a set of about 61,075 hits, (37,989 hits from bacteria sequences, 2557 hits from archaea, and 20,529 hits from eukaryotes), a reduced set of about 5973 representative members were selected by the 'cd-hit' program by removing duplicates and highly similar (>60% identity) sequences (Supplementary Data 1–3)[143]. These sequences were used as input for clustering using the program clans (cluster analysis of sequences)[144]. Clans perform all-against-all BLAST searches of unaligned sequences and cluster them by their similarity. The output map shows each sequence is represented as a dot, arranged on a two-dimensional plane so that their 2D-distances approximately correspond to the sequence similarities (Fig. 1a). This resulted in a total of five major clusters (clustera 1–5). Clusters 1, 2, and 5 were split into several sub-clusters due to their multi-lobe appearance. To visualize how these prokaryotic sequences cluster with known classical deacetylases, sequences of the human classical HDACs, HDACs from *Saccharomyces cerevisiae*, and the HDACs from *Arabidopsis thaliana* were added prior to clustering. The set of sequences covered by the clustering analysis is virtually identical to the sequence set covered by the PFAM domain PF00850 'Histone-deacetylase domain', which is now incorporated into the INTERPRO entry IPR050284 'Histone deacetylase and polyamine deacetylase'.

## Phylogenetic tree and sequence logo generation

The phylogenetic tree is based on a multiple sequence alignment generated with MAFFT (L-Ins-I modus). Before using the alignment for tree construction, columns with more than 90% of gaps were removed. The alignment was used to generate a neighbor-joining tree with the program belvu[192]. This was converted to Newick format and the unrooted phylogenetic tree was visualized with iTOL[193,194]. For the sequence logo representations, these sequences were analyzed by the program WebLogo 3[195].

## Expression and purification of proteins

The classical lysine deacetylases (KDACs) *Vs*HdaH (1b), *Rs*PrpH (1b), *Rw*DmhA (1b), *Kp*HdaH (1b), *Bs*AcuC (2c), *Vc*HdaH (5b), *Ps*ApaH (4), *Lc*ApaH (3) and *Lp*ApaH (3) (Uniprot: A0A2N0XRJ0, G3JWV8, A0A067XIQ6, A0A377Z5F6, P39067, A0A395TF31, A0A1Y0KY79, A0A0W0SGS1, A0A2S6EWV0), in their wild-type and catalytically inactive form, and *Bs*AcsA (Uniport: P39062) as well as *Bs*AcuA (Uniport: P39065) were expressed as *N*-terminal His$_6$-tagged fusion proteins in *E. coli* BL21 (DE3) cells (Supplementary Table 1). For expression in *E. coli* BL21 (DE3) codon optimized synthetic genes were used (pET-45b(+); BioCat GmbH, Heidelberg). PA1409 (Uniprot: Q9I3T5) was expressed as His$_6$-SUMO-fusion protein using the pOPIN-S vector with a SENP1 cleavage site[196]. Truncated proteins were expressed as His$_6$-tagged fusion proteins (pRSF-Duet1; Merck/Sigma-Aldrich, Novagen) in *E. coli* BL21 (DE3) cells. The protein expressions were conducted in 2 L LB media supplemented with 0.2 μM ZnCl$_2$. The cells were cultivated to an OD$_{600}$ of 0.5 (37 °C; 150 rpm) and expressed by inducing with 0.4 μM of isopropyl-β-D-thiogalactopyranoside (IPTG). *Vs*HdaH (1b), *Rs*PrpH (1b), *Rw*DmhA (1b), *Kp*HdaH (1b), *Bs*AcuC (2c), *Vc*HdaH (5b), *Ps*ApaH (4), *Bs*AcsA, *Bs*AcuA and PA1409 were expressed for 12–16 h (18 °C; 150 rpm) whereas *Lc*ApaH (3) and *Lp*ApaH (3) were expressed for 4 h (37 °C; 150 rpm). The cells were harvested by centrifugation (4000 × *g*, 20 min) and resuspended in resuspension buffer (*Vs*HdaH (1b), *Rs*PrpH (1b), *Rw*DmhA (1b), *Kp*HdaH (1b), *Bs*AcuC (2c), *Ps*ApaH (4): 50 mM Tris-HCl pH 8, 100 mM NaCl, 50 mM KCl, 1 mM β-mercaptoethanol; *Bs*AcsA, *Bs*AcuA: 50 mM Tris-HCl pH 7.4, 100 mM NaCl, 2 mM MgCl$_2$, 1 mM β-mercaptoethanol; *Vc*HdaH (5b): 100 mM K$_2$HPO$_4$/KH$_2$PO$_4$ pH 8, 100 mM NaCl, 50 mM KCl, 5% glycerol, 1 mM β-mercaptoethanol; PA1409: 50 mM Tris-HCl pH 8, 50 mM NaCl, 5 mM imidazole, 5% glycerol, 1 mM β-mercaptoethanol; *Lc*ApaH (3), *Lp*ApaH (3): 50 mM HEPES pH 8, 500 mM KCl, 1 mM MgCl$_2$, 5% glycerol 1 mM β-mercaptoethanol) containing 0.2 mM Pefabloc protease inhibitor cocktail. HEPES buffer pH was adjusted with NaOH. Cell lysis was done by sonication (3 × 3 min 1.5 s pulse/3 s pause), the cleared lysate (20,000 × *g*, 45 min) was applied to the equilibrated Ni$^{2+}$-NTA-column (resuspension buffer plus 10–20 mM imidazole). Washing was done with high-salt buffer (equilibration buffer with 500 mM NaCl). Elution from the Ni-NTA-column was performed over a gradient of 20–500 mM imidazole. Only PA1409 was dialyzed in resuspension buffer after elution and the His$_6$-SUMO fusion tag cleaved (0.2 mL, 2 mg/ml SENP1$_{415-644}$ protease) overnight at 4 °C. After cleavage a second Ni$^{2+}$-NTA was performed and the flow-through used. The eluted protein was concentrated by ultrafiltration and applied to an appropriate size-exclusion chromatography column (HiLoad 16/600 Superdex 75 or 200 pg; Cytiva). The concentrated fractions were shock frozen in liquid nitrogen and stored at −80 °C. Protein concentrations were determined measuring the absorption at 280 nm using the proteins' extinction coefficients.

## Plasmids and enzymes

For expression in bacterial cells synthetic codon optimized genes in pET-45b(+) (BioCat, Heidelberg) were used. Mutations for the catalytically inactive forms were introduced by site-directed mutagenesis according to the PCR Protocol for Phusion High-Fidelity DNA Polymerase (New England Biolabs). For cloning *E. coli* DH5α cells were used (Supplementary Table 1). PA1409 was cloned into a pOPIN-S vector and the truncated constructs of *Lc*ApaH (3) and *Lp*ApaH (3) in a pRSF-Duet1 vector (Merck/Sigma-Aldrich, Novagen) using the Gibson assembly Kit (New England Biolabs)[196]. For cloning, primers (Sigma-Aldrich), Phusion-DNA-polymerase, Taq-DNA-ligase, T5 exonuclease, and restriction enzymes were used (New England Biolabs). The oligonucleotides used for cloning and mutagenesis are listed in Supplementary Table 2.

## Analytical size-exclusion chromatography

To analyze the oligomeric states of the KDACs, analytical size-exclusion chromatography runs were performed on a calibrated Superdex 200 10/300 GL column (Cytiva) (Supplementary Fig. 1a). The SEC column was equilibrated with two column volumes of potassium phosphate buffer (100 mM K$_2$HPO$_4$/KH$_2$PO$_4$ pH 8, 100 mM NaCl, 50 mM KCl, 5% glycerol, 1 mM β-mercaptoethanol). Before injecting the protein into the column, the samples were diluted with potassium phosphate buffer to a concentration of 5 mg/ml, and 100 μl were applied to the column. The protein-containing fractions were identified by following the absorption at 280 nm and verified using SDS-PAGE. To calculate the molecular weights of the proteins a calibration curve was used (standard proteins in phosphate buffer: ribonuclease A (13.7 kDa,), carbonic anhydrase (29 kDa), ovalbumin (44 kDa), conalbumin (75 kDa), aldolase (158 kDa) and ferritin (440 kDa)) according to the Cytiva low and high molecular weight calibration kit (Supplementary Fig. 2b). A run with blue dextran was performed to calculate the void volume of the column. The partition coefficients (K$_{av}$) were obtained from the elution volumes (V$_e$), the column void volume (V$_0$), and geometric column volume (V$_c$), calculated using the following equation: K$_{av}$ = (V$_e$ - V$_0$)/(V$_c$ - V$_0$). The K$_{av}$ values were plotted as a function of the log molecular weight (MW) and fitted using a linear equation. The resulting calibration equations are shown with the

coefficient of determination R², showing the accuracy of the fit (Supplementary Fig. 2b).

## Deacetylation of *Bs*AcsA^K549Ac and Immunoblotting

For deacetylation of *Bs*AcsA^K549Ac, purified *Bs*AcsA was first enzymatically acetylated with *Bs*AcuA and Ac-CoA as described earlier[197]. Afterward, an SEC was performed (50 mM Tris-HCl pH 8, 100 mM NaCl, 2 mM MgCl₂) and the acetylated *Bs*AcsA^K549Ac was concentrated by ultracentrifugation to a concentration of 100 μM. The deacetylation reaction was performed with 30 μM *Bs*AcsA^K549Ac and 10 μM of the respective DAC for 3 h at 30 °C. The reactions were stopped by adding SDS-PAGE sample buffer (5×) and incubating the samples for 10 min at 95 °C. Before immunoblotting, the samples were separated by SDS-PAGE and afterward the proteins were transferred (transfer buffer: 25 mM Tris base, 150 mM glycine, 10% (v/v) methanol) to a PVDF membrane (0.2 μm, SERVA Electrophoresis GmbH, cat no. 42515.01) using a semi-dry immunoblotting system (90 min, 150 mA). Afterward, the membrane was stained with Ponceau S-red solution (VWR Chemicals/Sigma-Aldrich, cat no. 6226-79-5) to analyze the completeness of the transfer and as a loading control. The membrane was blocked with 3% (w/v) semi-skimmed milk PBS-T buffer (30 min, room temperature; PBS-T: 10 mM K₂HPO₄, 2 mM KH₂PO₄ pH 7.4, 140 mM NaCl, 2.7 mM KCl, 0.1% (v/v) Tween-20) and afterward incubated with the primary anti-acetyl-lysine antibody (rabbit anti-AcK-AB; abcam, cat no. ab21623, 1:2000 in 3% (w/v) semi-skimmed milk PBS-T; overnight, 4 °C). The membrane was washed three times with PBS-T buffer (5 min, room temperature) before the HRP-coupled secondary antibody (goat anti-rabbit-AB: Abcam, cat no. ab6721 (1:10,000 in 3% (w/v) semi-skimmed milk PBS-T)) was incubated for 1 h at room temperature. After washing the membrane three times with a PBS puffer (5 min, room temperature), the detection was done by using enhanced chemiluminescence (Roth, cat no. P078.2; OctopulsQPLEX, iNTAS science Imaging Instruments GmbH). For the subsequent detection of His₆-tagged proteins, the PVDF membrane was washed once for 5 min with PBS-T buffer, stripped (3 ml 30% (w/w) H₂O₂ incubated for 15 min at 37 °C), and blocked with 3% (w/v) semi-skimmed milk PBS-T buffer (30 min, room temperature). The PVDF membrane was then incubated with the primary mouse 6×-His tag monoclonal antibody (HIS.H8; Invitrogen, cat no. MA1-21315; 1:2000 in 3% (w/v) semi-skimmed milk PBS-T), the secondary HRP-coupled polyclonal rabbit anti-mouse IgG H&L antibody (Abcam, cat no. ab6728; 1:5000 in 3% (w/v) semi-skimmed milk PBS-T) and chemiluminescence was detected as previously described. For the validation of the immunoblotting, the program ImageJ was used to analyze the chemiluminescent signals[198]. The detected acetyl-lysine signals were normalized to the signals of the 6×-His Tag blot and afterward, the highest normalized acetylation signal was set to 100%.

## Acetyl-polyamine deacetylase assay

To determine acetylpolyamine deacetylase activity the Acetic Acid (RM) Kit (Megazyme, cat. No: K-ACETRM, LOT:230908-01) was used. First the acetylated-polyamines (*N*-acetylputrescine (Sigma-Aldrich; cat No A8784), *N*^1-acetyl-cadaverine (Combi-Blocks, cat no. QH-3990), *N*^1-acetyl-spermine (Sigma-Aldrich, cat No 01467), *N*^1/*N*^12-acetyl-spermine (MERK/Sulpelco, cat No 91423), *N*^1-acetyl-spermidine (Cayman chemical, cat no. 9001535-10) and *N*^8-acetyl-spermidine (MERK/Sulpelco, cat No A3658)) and KDACs were diluted in phosphate buffer (100 mM K₂HPO₄/KH₂PO₄ pH 8, 100 mM NaCl, 50 mM KCl, 1 mM β-mercaptoethanol). In the deacetylation assay, a concentration of 3 mM of acetylated polyamine and 100 nM PA1409 (positive control), 200 nM PsApaH, or 3 μM of the respective KDACs were used. The samples were incubated for 20 min at 37 °C and the reaction stopped by heating (2 min, 95 °C). Afterward, the samples were measured at 340 nm according to the protocol of the Acetic Acid (RM) Kit using 70 μl UV-cuvettes micro (BRAND). The kinetic for *Ps*ApaH (450 nM) with acetyl-putrescine was conducted in phosphate buffer using various concentrations of acetyl-putrescine (0.2 mM, 0.5 mM, 1 mM,

1.5 mM, 2 mM, 2.5 mM, 3 mM, 4 mM, 5 mM, 7 mM, and 10 mM) and samples were taken after 0, 1, 2, 4, 6, 10, and 15 min. P-values from the unpaid t-test and Michaelis–Menten kinetic were calculated using GraphPad Prism version 8 and version 9.5.1.

## Flour-de-Lys assay

For screening of deacetylase activity toward the Boc-Lys(Ac)-AMC (Sigma-Aldrich, cat no. SCP0168) and peptide S2 and S2a (Sigma-Aldrich; cat no. SRP0306, cat no. SRP0303) the Flour-de-Lys assay was used. The relative comparison was performed by using 20 nM of the respective KDAC and 20 μM of Boc-Lys(Ac)-AMC or 10 μM S2 or S2a peptide in Tris buffer (50 mM Tris/HCl pH 8, 100 mM NaCl, 50 mM KCl). All samples were incubated at 37 °C (Boc-Lys(Ac)-AMC for 30 min, S2 for 45 min, S2a for 30 min) in black med.-binding 96-well microtiter plates (F-bottom, chimney well, Greiner Bio-One, cat no. 655076) and the reaction stopped by adding stop-solution (0.2 mM trypsin (Sigma-Aldrich, cat no T4799), 100 μM TSA (Sigma-Aldrich, cat no T8552)) and incubated for 30 min at room temperature. Samples were measured at ex = 340 nm/em = 460 nm for Boc-Lys(Ac)-AMC and S2a (peptide 2a), and S2 (peptide 2) was measured at ex = 358 nm/em = 440 nm (Infinite® 200 PRO, TECAN). As controls human *Hs*HDAC1 (full length, *C*-terminal His-FLAG-Tag, BPS Bioscience, cat no. 50051), *Hs*HDAC6 (full length, N-terminal GST-Tag, Sigma-Aldrich, cat no.382180), *Hs*HDAC7 (a.a. 518-end, N-terminal GST-Tag, BPS Bioscience, cat no. 50007), *Hs*HDAC8 (full length, *C*-terminal His Tag, Sigma-Aldrich, cat no. 382184), *Hs*HDAC9 (a.a. 604-1066, *C*-terminal His Tag, BPS Bioscience, cat no. 50009) and *Hs*HDAC11 (full length, BPS Bioscience, cat no. 50021) were used.

## Assay materials

7-Amino-4-methylcoumarin (AMC) fluorescence assays were performed in black low-binding 96-well microtiter plates (Corning half-area wells, Fischer Scientific, cat. # 3686), with duplicate series and control wells without enzyme within each plate, and each assay performed at least twice. All experiments were performed in assay buffer [50 mM Tris/HCl, 137 mM NaCl, 2.7 mM KCl, 1.0 mM MgCl₂, 0.5 mg/mL bovine serum albumin (BSA), pH 8.0]. β-Nicotinamide adenine dinucleotide hydrate (NAD⁺) for sirtuin assays (Sigma-Aldrich, cat. #N7004), trypsin (Sigma-Aldrich, cat. #: T8003), HDAC3/NCoR2 (full length, HDAC3 C-terminal His tag, NCoR2_DAD N-terminal GST-tag, BPS Bioscience, cat. #: 50003, lot #: 190327), SIRT2 (a.a. 50–356, C-terminal His tag, BPS Bioscience, cat. # 50013, lot #: 190701-2), and SIRT5 (full length, N-terminal GST-tag, BPS Bioscience, cat. # 50016, lot #: 140813-1) were of commercial source. The following inhibitors were of commercial source: entinostat (MS-275, Sigma-Aldrich, cat. #: EPS002), SAHA (Sigma-Aldrich, cat. #: SML0061), trapoxin A (Sigma-Aldrich, #T8552) and TSA (Tokyo Chemical Industry, cat. #T2477); and apicidin A was synthesized as described[199]. The peptides used in this study are listed in the Supplementary Information (Supplementary Table 2). Stocks were prepared in DMSO (10–40 mM), and the concentration of peptide substrates was determined based on absorbance [ε₃₂₆(Ac-Lys-AMC) = 17,783 M⁻¹·cm⁻¹] using a Thermo Scientific NanoDrop^C instrument. Enzyme stocks were centrifuged after thawing (2 min, 16,200 × *g*, 4 °C), and the concentration of enzyme in the supernatant was determined based on absorbance and the corresponding ε₂₈₀. Assay concentrations were obtained by dilution from stock solutions in a buffer. Fluorescence recordings were performed in a FLUOstar Omega plate reader (BMG Labtech). Data analysis was performed using GraphPad Prism 9.

## Crystallization

For crystallization, the sitting drop method was used. The crystallization of N-terminally His₆-tagged *Rs*PrpH (1b), *Rw*DmhA (1b), and *Vc*HdaH (5b) was performed with the robotic platform CyBi™-HTPC (CyBio AG, Jena, Germany) in 96-well CrystalQuick LP plate, PS, Square

(Greiner Bio-One, cat no 609180). Therefore, 0.3 µl protein solution was mixed with 0.3 µl reservoir solution and the reservoir was filled with 40 µl solution of the condition. The storage temperature for the plates was 20 °C. Before crystallization, *Rs*PrpH (1b) was dialyzed in 25 mM HEPES pH 8, 100 mM NaCl, and concentrated to 10.7 mg/ml. To crystallize *Rw*DmhA (1b) (in 50 mM Tris/HCl pH 8, 100 mM NaCl, 50 mM KCl) a concentration of 10.5 mg/ml was used. For crystallization of *Vc*HdaH (5b), the protein was purified in phosphate buffer (100 mM $K_2HPO_4/KH_2PO_4$ pH 8, 100 mM NaCl, 50 mM KCl, 5% (v/v) glycerol, 1 mM β-mercaptoethanol) and concentrated to 10 mg/ml. The following *C*-terminal His$_6$-tagged proteins were crystallized using the robotic platform Crystal Gryphon (Art Robbins Instruments, USA): *Lp*ApaH (3), *Lc*ApaH$_{22-409}$ (3), *Ps*ApaH$_{1-342}$ (4). The crystallization was performed in 96-well Intelli-plates (Art Robbins Instruments, cat no. 102-0001-03) in which the compositions of the three different drops per well were 0.15 µl protein + 0.15 µl reservoir, 0.15 µl protein + 0.1 µl reservoir and 0.1 µl protein + 0.2 µl reservoir and the reservoir solution had a volume of 80 µl. The storage temperature of the plates was 18 °C. After purification of *Ps*ApaH$_{1-342}$, the protein was dialyzed in 50 mM Tris/HCl pH 7.5, 50 mM NaCl and 25 mM KCl, concentrated to 22 mg/ml, and crystallized. For crystallization *Lp*ApaH (3) was dialyzed in 50 mM HEPES pH 8, 400 mM KCl, 2.5% (v/v) glycerol and crystallized with 10 mg/ml. To crystallize *Lc*ApaH (3) the truncated protein (aa 22-409) was used (in 50 mM HEPES pH 8, 300 mM KCl, 5% (v/v) glycerol) with a concentration of 11 mg/ml.

For crystallization inhibitor proteins were co-crystallized. *Rw*DmhA (1b) and *Vc*HdaH (5b) were incubated in the buffers as described previously overnight at 4 °C with 10-fold higher concentrations of SAHA compared to protein (Cayman chemical, cat no. 10009929), concentrated on the next day and crystallized with the robotic platform CyBi™-HTPC. *Rs*PrpH (1b)•SAHA was concentrated to 14 mg/ml, *Rw*DmhA (1b)•SAHA was crystallized with 16 mg/ml and *Vc*HdaH (5b)•SAHA was crystalized with a concentration of 10 mg/ml. For crystallization of *Kp*HdaH (1b) with inhibitors SAHA and TSA the enzyme was crystallized (present in 50 mM Tris/HCl pH 8, 100 mM KCl, 2 mM β-mercaptoethanol) *Kp*HdaH (1b) in the apo form and subsequently crystals were soaked with 200 µM SAHA or TSA for 1 h at 20 °C. The crystallization conditions are shown in Supplementary Table 4. The crystals were measured at the BESSY II Synchrotron in Berlin, Germany, and at the DESY Synchrotron in Hamburg, Germany. Data sets were processed with the XDS package and the structures were solved using the molecular replacement program Phaser (Basic Molecular Replacement) from the CCP4i2 package[200,201]. As a search template, Alphafold2 models were used and the structure solution was improved/refined using Coot and Refmac5 from the CCP4i2 package[202-207]. The data collection and refinement statistics, values for structure validation, and geometry are listed in Supplementary Tables 5–10. The coordinates and structure factors of the structures were deposited in the PDB with accession codes as stated in the "Data availability" section.

## Sequence- and structure alignments
The sequence alignments were performed with the program T-Coffee/Expresso v.9.86 and analyzed using the software ESPript3.0 version 3.0.10[208-213]. As structure models x-ray structures or Alphafold2 models were used. For the structural alignment, if possible, x-ray structures otherwise Alphafold2 Monomer v2.0 models were used. The alignment was performed using Pymol[214].

## Substrate screening assays
Substrate (50 µM) and enzyme (100 nM) were incubated in assay buffer (50 mM Tris/HCl, 137 mM NaCl, 2.7 mM KCl, 1 mM MgCl$_2$, 0.5 mg/mL BSA, pH 8.0) for 60 min at 37 °C in a final volume of 25 µL. Experiments with SIRT2 and SIRT5 were supplemented with NAD$^+$ (500 µM). Thereafter, a solution of trypsin (25 µL, 5.0 mg/mL; final concentration

of 2.5 mg/mL) was added, and the development of the assay was allowed to proceed for 90 min at room temperature before fluorescence analysis. Background fluorescence data were subtracted, and the resulting fluorescence data were transformed into [AMC] using a standard curve measured under similar conditions. AMC fluorescence does not follow a linear trend at high concentrations, which explains calculated conversions >100%. To this end, a single category is given to all conversions >48% on the heatmap. The following control enzymes were employed for the screening: HDAC3/NCoR2 for K$_{ac}$, K$_{pro}$, K$_{but}$, K$_{cr}$, K$_{(L-la)}$, and K$_{(D-bhb)}$ modifications; SIRT2 for K$_{hex}$, K$_{oct}$, K$_{dec}$, K$_{lau}$, K$_{myr}$, K$_{pal}$, and K$_{bio}$ modifications; and SIRT5 for K$_{suc}$ and K$_{glu}$ modifications. For the H4-derived peptide LGK$_{bio}$ no positive control was available. Follow-up screening for *Vc*HdaH (5b) was performed in a similar manner, at 300 nM and 1 µM enzyme concentration; and de-D-/L-lactylase selectivity assays were performed at 100 nM (*Vs*HdaH (1b)) or 300 nM (*Lc*ApaH (3), *Lp*ApaH (3)) enzyme concentration, and both at 50 µM and 5 µM substrate concentration.

## Determination of enzyme kinetic parameters
Discontinuous method: enzyme kinetics with QPKK-based substrates were determined by incubation of enzyme (see concentrations below) and substrate (150–8.8 µM, or 337.5–8.8 µM for *Lc*ApaH (3) and *Lp*ApaH (3)) for 15, 30, 45, or 60 min at 37 °C in a final volume of 25 µL in assay buffer (50 mM Tris/HCl, 137 mM NaCl, 2.7 mM KCl, 1 mM MgCl$_2$, 0.5 mg/mL BSA, pH 8.0). Then, a solution of trypsin containing the inhibitor TSA [25 µL, 5.0 mg/mL with 1 µM TSA (100 µM TSA for *Rw*DmhA (1b)); final concentrations of 2.5 mg/mL and 0.5 µM TSA (50 µM TSA for *Rw*DmhA (1b))] was added, and the development of the assay was allowed to proceed for 90 min at room temperature before fluorescence analysis. The following enzyme concentrations were employed: *Kp*HdaH (1b), 8 nM; *Lc*ApaH (3), 0.1 nM (QPKK$_{ac}$) or 10 nM (QPKK$_{pro}$); *Lp*ApaH (3), 0.2 nM (QPKK$_{ac}$) or 10 nM (QPKK$_{pro}$); *Ps*ApaH (4), 20 nM; *Rw*DmhA (1b), 40 nM; and *Vs*HdaH (1b), 5 nM (QPKK$_{ac}$) or 100 nM [QPKK(L-la)].

Continuous method: enzyme kinetics with LGK-based substrates were determined by incubation of enzyme (see concentrations below), substrate (150–8.8 µM) and trypsin (10 µg/mL for *Bs*AcuC (2c), 7 µg/mL for *Lc*ApaH (3) and *Lp*ApaH(3)) for 8–35 min at 25 °C in a final volume of 50 µL in assay buffer (50 mM Tris/HCl, 137 mM NaCl, 2.7 mM KCl, 1 mM MgCl$_2$, 0.5 mg/mL BSA, pH 8.0), and with fluorescence read every 30 s. Trypsin concentration was optimized prior to the assay to verify fast fluorophore release and maximize enzyme stability[215]. The following enzyme concentrations were employed: *Bs*AcuC (2c), 200 nM (LGK$_{ac}$, LGK$_{pro}$) or 400 nM (LGK$_{cr}$); *Lc*ApaH (3), 0.4 nM (LGK$_{ac}$), 40 nM (LGK$_{pro}$) or 150 nM [LGK$_{(L-la)}$, LGK$_{(D-la)}$]; LpApaH: 1.5 nM (LGK$_{ac}$) or 75 nM (LGK$_{pro}$).

For all enzyme kinetics $v_0$ is the initial reaction rate, i.e. [AMC] in nM converted per units of time in s$^{-1}$, i.e. $v_0 = $ [AMC]*s$^{-1}$ (unit: nM*s$^{-1}$). Plotted are the values of $v_0/[E_0]$ (unit: s$^{-1}$), with [$E_0$] as the total enzyme concentration (unit: nM). Data from both methods were adjusted for background and transformed into [AMC], and the linearity of the data was verified before fitting either to the Michaelis−Menten equation (Eq. 1), where [$E$]$_0$ is the initial concentration of the deacylase and [$S$] is the initial concentration of substrate or the Michaelis−Menten equation with substrate inhibition (Eq. 2) where the $K_i$ inhibitory constant of the substrate could also be calculated.

$$\frac{v_0}{[E]_0} = \frac{k_{cat}[S]}{K_M + [S]} \tag{1}$$

$$\frac{v_0}{[E]_0} = \frac{k_{cat}[S]}{K_M + [S]\left(1 + \frac{[S]}{K_i}\right)} \tag{2}$$

## Inhibitor screening assays

Substrate (40 μM LGK$_{ac}$ for $Bs$AcuC (2c), $Lc$ApaH (3), $Lp$ApaH (3); or 60 μM QPKK$_{ac}$ for $Kp$HdaH (1b), $Ps$ApaH (4), $Rw$DmhA (1b), $Vs$HdaH (1b)), inhibitor (10 μM or 1 μM) and enzyme were incubated in assay buffer (50 mM Tris/HCl, 137 mM NaCl, 2.7 mM KCl, 1 mM MgCl$_2$, 0.5 mg/mL BSA, pH 8.0) for 60 min at 37 °C in a final volume of 25 μL. Then, a solution of trypsin (25 μL; 0.4 mg/mL for LGK$_{ac}$ assays, final concentration of 0.2 mg/mL; or 5.0 mg/mL for QPKK$_{ac}$ assays, final concentration of 2.5 mg/mL) was added, and the assay was allowed to develop for 15 min (LGK$_{ac}$ assays) or 90 min (QPKK$_{ac}$ assays) at room temperature. Data was transformed to relative enzyme activity (%) compared to control wells without inhibitors.

## Dose-response inhibition assays

Substrate (40 μM LGKac for $Bs$AcuC (2c), $Lc$ApaH (3), $Lp$ApaH (3); or 60 μM QPKKac for $Kp$HdaH (1b), $Ps$ApaH (4), $Rw$DmhA (1b), $Vs$HdaH (1b)), inhibitor (30.0 μM–1.52 nM) and enzyme (see concentrations below) were incubated in assay buffer (50 mM Tris/HCl, 137 mM NaCl, 2.7 mM KCl, 1 mM MgCl$_2$, 0.5 mg/mL BSA, pH 8.0) for 30 min (LGK$_{ac}$ assays) or 60 min (QPKK$_{ac}$ assays) at 37 °C in a final volume of 25 μL. Then, a solution of trypsin (25 μL; 0.4 mg/mL for LGKac assays, final concentration of 0.2 mg/mL; or 5.0 mg/mL for QPKK$_{ac}$ assays, final concentration of 2.5 mg/mL) was added, and the assay was allowed to develop for 15 min (LGK$_{ac}$ assays) or 90 min (QPKK$_{ac}$ assays) at room temperature. The following enzyme concentrations were employed: $Bs$AcuC (2c), 200 nM; $Kp$HdaH (1b), 15 nM; $Lc$ApaH (3), 0.5 nM; $Lp$ApaH (3), 0.5 nM; $Ps$ApaH (4), 40 nM; $Rw$DmhA (1b), 30 nM; $Vs$HdaH (1b), 1.5 nM.

Data was transformed to relative enzyme activity (%) compared to control wells without inhibitor, and fitted to sigmoidal functions with variable slope (Eqs. 3, 4 parameters, $h$: Hill slope) to afford IC$_{50}$ values. Assuming fast-on/fast-off competitive inhibition, the Cheng–Prusoff equation (Eq. 4) was employed to transform IC$_{50}$ data into inhibitory constants ($K_i$), with $K_M$ values as determined previously.

$$\text{Res. activity} = \text{Res. activity}_{bottom} + \frac{\text{Res. activity}_{top} - \text{Res. activity}_{bottom}}{1 + 10^{(\log \text{IC}_{50} - \log[I])h}}$$

$$(3)$$

$$K_i = \frac{\text{IC}_{50}}{1 + \frac{[S]}{K_M}}$$

$$(4)$$

## Data analysis and visualization

Fiji (ImageJ 2.0.0-rc-68/1.52h) was used for quantitative analysis of the immunoblots[216]. Raw data from most experiments was processed using Microsoft Excel 2011. Data was visualized and statistically analyzed in GraphPad Prism version 8 or version 9.5.1. Fitting of data was also performed in GraphPad Prism version 8 or version 9.5.1. SnapGene Viewer 5.1.4.1 was employed for DNA sequence handling and generation of plasmid maps (SnapGene software from Insightful Science; available at snapgene.com). PyMOL version 2.5.3 and PyMOL version 2.3.4 were used to generate visual representations of protein structures and protein-inhibitor complexes[217]. ChemDraw version 23.0.1 was used to draw chemical structures. Adobe Photoshop 22.3.1 and Adobe Illustrator 25.4.1 were used to create figures.

## Statistics and reproducibility

All assays were performed in independent replicates as indicated resulting in similar results. For bar graphs, the standard deviations (SD) and mean values were depicted. No statistical method was used to predetermine the sample size. For all kinetic experiments, a threshold of substrate conversion (typically 10%) was set and any data points over that threshold were excluded for analyses to ensure measurement of the initial rates of the respective enzyme at the initial concentration of substrate[215]. In the continuous kinetic experiments assessment of enzyme activity in the coupled assay format requests to obtain equilibration of trypsin activity. To this end, the first data points were not linear for some measurements and were therefore excluded from the evaluation[215]. No further data were from the analyses. Unpaired, two-tailed student's t-tests or an ordinary one-way ANOVA test (Tukey's multiple comparison test) were performed to assess statistical significance with significance levels as indicated.

## Reporting summary

Further information on research design is available in the Nature Portfolio Reporting Summary linked to this article.

## Data availability

The coordinates and structure factors for the structures of 9GLB ($Kp$HdaH 1b), 9GN1 ($Kp$HdaH H144A 1b), 9GN7 ($Kp$HdaH 1b•TSA), 9GN6 ($Kp$HdaH 1b•SAHA), 9GKU ($Rs$PrpH 1b), 9GKW ($Rw$DmhA 1b), 9GKY ($Vc$HdaH (5b), 9GL0 ($Lp$ApaH 3), 9GL1 ($Lc$ApaH$_{22-409}$ 3), 9GKZ ($Ps$ApaH$_{1-342}$ 4), 9GKX ($Rw$DmhA•SAHA 1b) and 9GKV ($Vc$HdaH 5b•SAHA) were deposited in the PDB (http://www.rcsb.org). Reference structures used in this work are available in the PDB under accession codes 7LTG (human HDAC2•apicidin A), 5VI6 (human HDAC8•trapoxin), 3Q9C, and 3Q9E. Source data are provided with this paper.

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

## Acknowledgements

We thank the group of Prof. Marcus Krüger/CECAD, Institute for Genetics, CECAD, University of Cologne for support in conducting pull-down experiments, conducting mass spectrometry experiments, and analyzing the data. This work was supported by the German Research Foundation (DFG, Deutsche Forschungsgemeinschaft) grants No. INST 292/156-1 FUGG (M.L.), INST 292/154-1 FUGG (M.L.), LA2984-5/1 (Project: 389564084; M.L.), LA2984-6/1 (Project: 449703098; M.L.), and a LEO Foundation Open Competition Grant (LF-OC-21-000901; C.A.O.). We thank HZB/BESSY, Berlin, and EMBL/DESY, Hamburg for continuous support in X-ray data collection.

## Author contributions

L.G.G. and C.M.Y. performed most biochemical experiments. C.M.Y. performed enzyme kinetics on peptides with various acyl-chains and inhibition assays. C.Q. and Sa.S. contributed to biochemical experiments, and O.S. performed pull-down experiments. N.W., D.H., and L.J. performed genomic deletion, conducted growth experiments, and assays on biofilm formation and infection experiments. M.J. and D.Z. grew bacterial strains and supplied cell lysates for pull-down experiments and bacterial growth experiments. D.A., D.Z., and O.S. performed mass spectrometry experiments and performed bacterial growth experiments. L.G.G., Sa.S, C.Q. G.J.P., and M.L. collected data for X-ray crystallography and solved the structures. L.B., B.G., B.D., and D.M.W. supported the expression and purification of proteins. C.A.O. supervised enzyme kinetics. R.A.S. supervised experiments with genomic deletion strain. K.H. performed bioinformatic analyses. Su.S. supervised proteomics experiments. M.L. initiated, designed, and supervised the study. L.G.G., Sa.S., and M.L. wrote the manuscript. All authors contributed to data analysis and gave comments on writing the manuscript.

## Funding

## Competing interests

The authors declare no competing interests.
