## [Transparent Peer Review file · Nature Communications]

Distribution and diversity of classical deacylases in bacteria

Corresponding Author: Professor Michael Lammers

Version 0:

Reviewer comments:

Reviewer #1

(Remarks to the Author)

The manuscript by Graf and colleagues describes the comprehensive discovery and analysis of bacterial deacetylases belonging to the arginase-deacetylase family. In short, this is brilliant work that should be published without delay in Nature Communications. As summarized in the Abstract, the authors have discovered “thousands of uncharacterized classical deacetylases in bacteria, which were grouped into five clusters”. The authors selected representative examples from each cluster and performed exhaustive enzymological measurements, accompanied by 12 protein crystal structures of native proteins and selected inhibitor complexes. Data collection and refinement statistics for all structures are recorded in Supplementary Tables 5–10; all structures are well determined with excellent statistics, although the separation between R and R_{free} for PsApaH (Supplementary Table 9) and for LpApaH (Supplementary Table 10) is a bit high for each, and may indicate overfitting at the modest resolutions of these structure determinations. Regardless, I expect that this will be a very highly cited paper. I would ask the authors to consider the following points prior to publication:

1. Line 73: the polyamine spermidine is in fact present in bacteria; perhaps the authors meant spermine here? Even so, while spermine had not been thought to be widely present in bacteria, the identification of bacterial spermine synthases suggests that spermine plays a greater role in bacteria than previously appreciated [Li et al. (2024) J. Biol. Chem. 300, 107281].
2. Line 97: the authors introduce the term “classical HDACs” to refer to the class I, II, and IV deacetylases, but in lines 110–112 and also 120–121 it is not clear that all of the cited papers actually refer to “classical HDACs” (line 110) or “classical deacetylases” (line 120). For example, ref. 124 pertains to NMR studies of the bacterial deacetylase LpxC, which adopts a protein fold quite different from that of the arginase-deacetylase fold (distinctive folds of many deacetylases can be compared in Figure 1 of ref. 105). The authors might clarify their use of the term “classical”, of course allowing for the fact that enzymes that do not belong to the arginase-deacetylase superfamily can also catalyze deacetylation chemistry.
3. Lines 104–106: the sentence “Sirtuins are structurally and regarding...” is awkward and should be reworded for clarity.
4. Line 131: ref. 93 reports the first discovery that HDAC10 is an N8-acetylspermidine deacetylase and would be more appropriate to cite here.
5. Line 216, and throughout the manuscript: the term “primary sequence” should be replaced by either “amino acid sequence” or “primary structure”.
6. Lines 511–512: the GGGGY sequence of HDAC8 has been probed by mutagenesis to show that this sequence is required for optimal catalytic function, to confer conformational flexibility for the catalytic tyrosine [Porter et al. (2016) Biochemistry 55, 6718].
7. Line 673: ref. 93 contains the substrate specificity studies of HDAC10 and is better cited here instead of ref. 94.
8. Lines 675–678: The glutamate gatekeeper of HDAC10 forms two water-mediated hydrogen bonds with the secondary ammonium group of N8-acetylspermidine and thereby confers substrate specificity over N1-acetylspermidine. Ref 94 does not contain this information, which instead is found here: Herbst-Gervasoni & Christianson (2021) Biochemistry 60, 303.
9. Line 686: ref. 93 and not ref 94 contains the enzymological measurements mentioned here.

10. Typographical errors: line 155, "diacylation" should be "deacylation"; line 699, "obtain" should be "contain"; line 768, "Prussof" should be "Prusoff".

Reviewer #2

(Remarks to the Author)

The work entitled "Distribution and diversity of classical deacylases in bacteria" presents a large and deep study of the currently known (as inferred from sequence analysis) classical deacylases. i.e enzymes that remove "acyl" groups from other biomolecules. The work draws attention from well known Zn²⁺ deacylases of eukaryotes, and based on sequence analysis, i.e building of a profile and phylogenetic tree, identifies 5 main groups of bacterial deacylases. The authors extensively and deeply characterize representative members of each group, in terms of substrate selectivity (polyamines, peptides, acyl-chain length etc.), enzyme kinetics, structural analysis and response to known inhibitors.

Through the extensive analysis, authors propose the structural requirements for substrate selectivity, and discovered the first bacterial de-D-/L-lactylases and long-chain deacylases. Interestingly, the observe effect of known inhibitors they suggest could set the ground for drug re purposing strategies to fight bacterial infections.

The amount of performed experimental work is impressive and of high quality, and the topic is definitively relevant, since this a group of very relevant enzymes. The presented conclusions are well supported by the data, and the authors have, in my opinion, gained deep and crucial insight into key issues related to these proteins sequence to structure to function relationships. In this regards my opinion is that the work is relevant and has merits to be published Nat. Comm.

My main critic, which I nonetheless think can be corrected, is related to the presentation in terms of length and detail of the descriptions for a wide audience, which is not necessarily specialist in the details of acylases. Concerning the methods I have only few comments related to particular computational methods. Specific comments follows:

1) Concerning the presentation I believe the work is too long and goes to much into detailed description, particular when describing the substrate specificity results in relation to other known, I understand, mostly eukaryotic deacylases. Also, it seems like the authors try to justify the relevance of these bacterial proteins only in the context of their eukaryotic cousins and not as relevant by themselves. I suggest to try to reduce the length of both introduction (by only briefly commenting eukaryotic enzymes) an the results, particularly those related to substrate specificity.

2) Authors also could reduce the number of figures and keep tables of the results and the more general results of the comparison if the main groups, subgroups and selected enzymes, while avoiding going into the detail of each case. All the removed information, which is nonetheless relevant could be moved to SI for the reader who is interested in that level of detail.

Concerning the methods:

3) clusters are first presented (Figure 1) based on pairwise comparison, and only later in terms of phylogeny. I think it is makes much more biological sense to use as man results the phylogenetic tree, clusters as comparison can go into SI.

4) How do the GP (or model) compare to known (sub) family and/or domain and/or clans available in protein family databases such as interpro (which now includes PFAM among others). Are those sequences retrieved by the authors comparable to those found in the commonly used databases?

5) Although the MSA is nice, it is always biased by the selection of the shown sequences. I think I would be more interesting to see sequence logos, at least of the whole family, and possibly key elements of each group.

6) How was the tree rooted? The authors do not specific if (and which in this case) they used an outgroup.

Reviewer #3

(Remarks to the Author)

Leonie G. Graf et al. provided a detailed structure-function analysis of bacterial deacylases. They discussed how to classify these prokaryotic enzymes and their similarity/differences with eukaryotic deacylases. Based on the biochemical assays and structural data, the authors discussed the determinants for their substrate specificity, acyl chain preference, oligomerization states and inhibitor selectivity. This work not only provide a nice overview of bacterial deacylase but also suggest the possibility to targeting these enzymes in a drug repurposing strategy to develop novel therapeutics to fight bacterial infections.

Overall this is a very interesting paper generally well formulated, but we have following remarks and questions on the proposed text.

Comments:

Line 252-254: "Accordingly, we observed strong activity for HsHDAC7 (1g) and HsHDAC9 253 (1g) but neither for the class I enzyme HsHDAC1 (2a) nor the class IIb enzyme HsHDAC6 (1g), 254 supporting the validity of the assay and of the clustering (Fig. 2c; Fig. 1a,b; Supplementary Data 1-3)." However, the assay data of HsHDAC6 were not presented in the Fig 2c.

Line 335-337: Original text was as follows "However, we found the peptide LGKoct being most efficiently hydrolyzed by BsAcuC (2c) compared to the other LGKacyl peptides (Fig. 3a)." In Fig 3a, LGKoct was not found. Further explanation

should be given to support your conclusion.

Line 344-347: "This agrees well with our data in which we observed for BsAcuC (2c) high deacylase activity for the peptides containing a Gly-Lys (Fig. 3a,d). Notably, the deacylation efficiency for the physiological substrate BsAcS A might be substantially higher as the peptides analyzed here differ in their substrate sequences." From Fig 3a and supplementary Fig 6, it is clear that QPKKhex, QPKKoct, QPKKdec and TARKdec showed similar or better extent activity as LGK. It is difficult to get the conclusion as stated in Line 344-346. Further quantitative assay should be performed.

Additionally, there is no data supporting the hypothesis stated in Line 346-347. The biochemical assays should be measured for the native peptide sequence of BsAcS A, if possible.

Line 492-493: The original text is "In bacterial deacylases of cluster 3, LpApaH (3) and LcApaH (3), the first His is replaced by an Asn (Fig. 4b)." However, based on the sequence alignment in Fig 1, the two His residues seem highly conserved in bacterial deacylases including cluster 3. Therefore, the description is kind of controversial.

Line 505: The interacting residues described in text "(KpHdaH: Val189, Tyr192, Arg195, Tyr227)" is different from the residues shown in Fig 4c. Please correct the text or Fig 4c.

Minor comments:

1. Line 16: The affiliation corresponding to number 5 has not been linked to any author in the author list. Please check it.
2. Supplementary Data should be cited according to the order they described in the main text.
3. Line 244-245: The corresponding references for peptide 2 and peptide 2a should be added.
4. In table 1, "VsvHdaH" should be corrected to VSHdaH.
5. Line 472: In the text "(5b)and by X-ray crystallography", the word "and" should be deleted.
6. The abbreviations of amino acids (one letter or three letters) are better to keep consistent in the whole text and figures.
7. Fig 5e is a redundant with Fig 1c.
8. The position of the reference numbers (in front of/behind the punctuation marks) should be consistent within the manuscript.
9. In supplementary table 4-10, the PDB entry should be added. Additionally, the crystallization information of KpHdaH (1b) H144A should be added in Supplementary table 4.
10. The description in line 861 is likely linked to Fig. 8d not 8c.
11. Line 1129: The word "shwos" needs to be corrected.
12. What kind of base was used for the pH adjustment of "HEPES"? The information needed to be provided in the method section.
13. Line 1180: "E. coli DH5 cells" needs to be corrected.
14. Line 1256: "50 mM Tris/HCl pH 8" needs to be corrected.
15. Line 1326: Supplementary table should be cited correctly.

Reviewer #4

(Remarks to the Author)

Version 1:

Reviewer comments:

Reviewer #1

(Remarks to the Author)

The manuscript has been revised to the satisfaction of this reviewer.

Reviewer #2

(Remarks to the Author)

The revised version of the work entitled "Distribution and diversity of classical deacylases in bacteria" presents a corrected and improved version of the original manuscript. The authors have correctly addressed all my concerns as well as those raised by the other reviewers. I therefore recommend the manuscript for publication. Specific comments follow:

- 1) The authors have shortened the manuscript where possible. Specifically, they removed descriptions of eukaryotic classical deacetylases in the introduction and discussion and emphasized the work on the bacterial enzymes.
- 2) The authors showed that the set of sequences covered by their clustering analysis is virtually identical to the sequence set covered by the PFAM domain PF00850 'Histone deacetylase domain', which is now incorporated into the INTERPRO entry IPR050284 'Histone deacetylase and polyamine deacetylase'. This information should be explicitly mentioned in the manuscript.
- 3) As requested they have in the revised version prepared sequence logos for the individual clusters, which are shown in the SI section of the revised version.
- 4) The authors state that the tree is not rooted since no suitable outgroup is available. Therefore I would recommend to

remove the root to avoid possible confusion of its biological meaning.

Reviewer #3

(Remarks to the Author)

All my comments have been addressed accordingly. I think this manuscript is suitable to be published in this journal.

Reviewer #4

(Remarks to the Author)

Point-by-point response to the reviewers' comments

Reviewer 1:

We thank reviewer 1 for carefully reading our manuscript and giving this valuable input. We concisely worked on the comments and integrated these into the revised version of the manuscript. Find the answers to all points raised by the reviewer in the subsequent section:

Reviewer #1 (Remarks to the Author)

The manuscript by Graf and colleagues describes the comprehensive discovery and analysis of bacterial deacetylases belonging to the arginase-deacetylase family. In short, this is brilliant work that should be published without delay in Nature Communications. As summarized in the Abstract, the authors have discovered “thousands of uncharacterized classical deacetylases in bacteria, which were grouped into five clusters”. The authors selected representative examples from each cluster and performed exhaustive enzymological measurements, accompanied by 12 protein crystal structures of native proteins and selected inhibitor complexes. Data collection and refinement statistics for all structures are recorded in Supplementary Tables 5–10; all structures are well determined with excellent statistics, although the separation between R and R_{free} for PsApaH (Supplementary Table 9) and for LpApaH (Supplementary Table 10) is a bit high for each, and may indicate overfitting at the modest resolutions of these structure determinations. Regardless, I expect that this will be a very highly cited paper. I would ask the authors to consider the following points prior to publication:

Point 1:

1. Line 73: the polyamine spermidine is in fact present in bacteria; perhaps the authors meant spermine here? Even so, while spermine had not been thought to be widely present in bacteria, the identification of bacterial spermine synthases suggests that spermine plays a greater role in bacteria than previously appreciated.

Response:

We agree, we added the reference as suggested and modified the text as suggested to abolish the confusion. (lines 71-74)

Point 2:

2. Line 97: the authors introduce the term “classical HDACs” to refer to the class I, II, and IV deacetylases, but in lines 110–112 and also 120–121 it is not clear that all of the cited papers actually refer to “classical HDACs” (line 110) or “classical deacetylases” (line 120). For example, ref. 124 pertains to NMR studies of the bacterial deacetylase LpxC, which adopts a protein fold quite different from that of the arginase-deacetylase fold (distinctive folds of many deacetylases can be compared in Figure 1 of ref. 105). The authors might clarify their use of the term “classical”, of course allowing for the fact that enzymes that do not belong to the arginase-deacetylase superfamily can also catalyze deacetylation chemistry.

Response:

We agree, we altered the text (lines 102-107). Both, sirtuins and classical deacetylases are known to be active in removing acylations others than acetylations from lysine side chains. To clarify that from an ancestor enzyme, deacetylases with different substrate preferences developed during evolution, we rewrote the sentence from line 120 to:

“From this precursor, classical Zn²⁺-dependent protein lysine-, polyamine- and small molecule-deacetylases divergently developed during evolution¹.”

Furthermore, we agree with reviewer 1 that LpxC is no classical deacylase with arginase/deaceylase fold and removed the reference.

Point 3:

3. Lines 104–106: the sentence “Sirtuins are structurally and regarding...” is awkward and should be reworded for clarity.

Response:

We agree, we rewrote the sentence (lines 103-104):

“Sirtuins are structurally and mechanistically, regarding the catalytic strategies used to achieve substrate deacylation, not related to classical HDACs²⁻⁴.”

Point 4:

4. Line 131: ref. 93 reports the first discovery that HDAC10 is an N8-acetylspermidine deacetylase and would be more appropriate to cite here.

Response:

We agree and checked carefully if the references 93 and 94 were used appropriately throughout. At this position we removed the sentence as it does not contain information that is essential to understand the story and another reviewer asked to shorten the manuscript.

Point 5:

5. Line 216, and throughout the manuscript: the term “primary sequence” should be replaced by either “amino acid sequence” or “primary structure”.

Response:

We agree and replaced the term “primary sequence” with “amino acid sequence” throughout the manuscript.

Point 6:

6. Lines 511–512: the GGGGY sequence of HDAC8 has been probed by mutagenesis to show that this sequence is required for optimal catalytic function, to confer conformational flexibility for the catalytic tyrosine [Porter et al. (2016) *Biochemistry* 55, 6718].

Response:

We agree and added the reference and rewrote the text:

“This sequence confers conformational flexibility for substrate binding and catalysis in agreement with earlier studies performed on human HDAC8 (Fig. 1c).”

Point 7:

7. Line 673: ref. 93 contains the substrate specificity studies of HDAC10 and is better cited here instead of ref. 94.

Response:

We corrected the citation as suggested.

Point 8:

8. Lines 675–678: The glutamate gatekeeper of HDAC10 forms two water-mediated hydrogen bonds with the secondary ammonium group of N8-acetylspermidine and thereby confers substrate specificity over N1-acetylspermidine. Ref 94 does not contain this information, which instead is found here: Herbst-Gervasoni & Christianson (2021) *Biochemistry* 60, 303.

Response:

We altered the references as suggested.

Point 9:

9. Line 686: ref. 93 and not ref 94 contains the enzymological measurements mentioned here.

Response:

We altered as suggested.

Point 10:

10. Typographical errors: line 155, “diacylation” should be “deacylation”; line 699, “obtain” should be “contain”; line 768, “Prussof” should be “Prusoff”.

Response:

We corrected the typos.

Reviewer 2:

We thank reviewer 2 for carefully reading our manuscript and giving this important and valuable input. We concisely worked on the comments and integrated these into the revised version of the manuscript. Find the answers to all points raised by the reviewer in the subsequent section:

Reviewer #2 (Remarks to the Author)

The work entitled "Distribution and diversity of classical deacylases in bacteria" presents a large and deep study of the currently known (as inferred from sequence analysis) classical deacylases. i.e enzymes that remove "acyl" groups from other biomolecules. The work draws attention from well known Zn²⁺ deacylases of eukaryotes, and based on sequence analysis, i.e building of a profile and phylogenetic tree, identifies 5 main groups of bacterial deacylases. The authors extensively and deeply characterize representative members of each group, in terms of substrate selectivity (polyamines, peptides, acyl-chain length etc.), enzyme kinetics, structural analysis and response to known inhibitors. Through the extensive analysis, authors propose the structural requirements for substrate selectivity, and discovered the first bacterial de-D-/L-lactylases and long-chain deacylases. Interestingly, the observe effect of known inhibitors they suggest could set the ground for drug re purposing strategies to fight bacterial infections. The amount of performed experimental work is impressive and of high quality, and the topic is definitively relevant, since this a group of very relevant enzymes. The presented conclusions are well supported by the data, and the authors have, in my opinion, gained deep and crucial insight into key issues related to these proteins sequence to structure to function relationships. In this regards my opinion is that the work is relevant and has merits to be published Nat. Comm. My main critic, which I nonetheless think can be corrected, is related to the presentation in terms of length and detail of the descriptions for a wide audience, which is not necessarily specialist in the details of acylases. Concerning the methods I have only few comments related to particular computational methods. Specific comments follows:

Point 1:

1) Concerning the presentation I believe the work is too long and goes to much into detailed description, particular when describing the substrate specificity results in relation to other known, I understand, mostly eukaryotic decaylases. Also, it seems like the authors try to justify the relevance of these bacterial proteins only in the context of their eukaryotic cousins and not as relevant by themselves. I suggest to try to reduce the length of both introduction (by only briefly commenting eukaryotic enzymes) an the results, particularly those related to substrate specificty.

Response:

We agree with reviewer 2 that the manuscript is quite long but we present a huge amount of data which we think is important to show. However, we shortened the manuscript where possible. We also removed descriptions of eukaryotic classical deacetylases in the introduction and discussion to emphasize our work on the bacterial enzymes. However, with respect to the detailed descriptions on molecular determinants underlying substrate specificity we think the level of detail is important.

Point 2:

2) Authors also could reduce the number of figures and keep tables of the results and the more general results of the comparison if the main groups, subgroups and selected enzymes, while avoiding going into the detail of each case. All the removed information, which is nonetheless relevant could be moved to SI for the reader who is interested in that level of detail.

Response:

We agree with reviewer 2 that we described members of enzymes representing each cluster. However, we do not focus on members of the sub-groups to enable to get an overview on the main clusters. It might be possible that there are differences in substrate specificity comparing enzymes of each sub-group/sub-subcluster, and indeed we described this for sub-cluster 1b enzymes, but to avoid to give too much information we focused on the main clusters. From our point of view the level details given in the main text on selected enzymes of each main cluster is important to describe their structural features, activities and substrate specificities. As an example, for the structural data we presented the main features in one section and only described structural features unique to the individual sub-clusters in separate paragraphs. Along that line, for the clusters that are specific to bacteria we did not find any sub-clusters at all. We think we wrote the manuscript in a way that allows to allow to be able to follow the experiments and results and moving huge parts of the results to the SI section will be on cost of readability and understanding.

Point 3:

3) clusters are first presented (Figure 1) based on pairwise comparison, and only later in terms of phylogeny. I think it makes much more biological sense to use as main results the phylogenetic tree, clusters as comparison can go into SI.

Response:

One should consider that the phylogenetic tree is based on the clustering analysis, since for every cluster two representative sequences have been chosen for the tree construction. A tree using all sequences as input would not have been suitable. Given that the clustering was an essential part of the analysis pipeline and figure 1a does not use that much space, we would prefer to keep it in the main text.

Point 4:

4) How do the GP (or model) compare to known (sub) family and/or domain and/or clans available in protein family databases such as InterPro (which now includes PFAM among others). Are those sequences retrieved by the authors comparable to those found in the commonly used databases?

Response:

The set of sequences covered by our clustering analysis is virtually identical to the sequence set covered by the PFAM domain PF00850 'Histone deacetylase domain', which is now incorporated into the INTERPRO entry IPR050284 'Histone deacetylase and polyamine deacetylase'.

Point 5:

5) Although the MSA is nice, it is always biased by the selection of the shown sequences. I think I would be more interesting to see sequence logos, at least of the whole family, and possibly key elements of each group.

Response:

We agree with this proposal and have now prepared sequence logos for the individual clusters, which are shown in the SI section.

Point 6:

6) How was the tree rooted? The authors do not specify if (and which in this case) they used an outgroup.

Response:

This is an unrooted tree since no suitable outgroup is available. For purpose of display in iTOL, the root was arbitrarily placed between cluster 5 and the other sequences. This placement does not imply a biologically meaningful root.

Reviewer 3:

We thank reviewer 3 for carefully reading our manuscript and giving this valuable and important input. We concisely worked on the comments and integrated these into the revised version of the manuscript. Find the answers to all points raised by the reviewer in the subsequent section:

Reviewer #3 (Remarks to the Author):

Leonie G. Graf et al. provided a detailed structure-function analysis of bacterial deacylases. They discussed how to classify these prokaryotic enzymes and their similarity/differences with eukaryotic deacylases. Based on the biochemical assays and structural data, the authors discussed the determinants for their substrate specificity, acyl chain preference, oligomerization states and inhibitor selectivity. This work not only provide a nice overview of bacterial deacylase but also suggest the possibility to targeting these enzymes in a drug repurposing strategy to develop novel therapeutics to fight bacterial infections. Overall this is a very interesting paper generally well formulated, but we have following remarks and questions on the proposed text.

Point 1:

Line 252-254: "Accordingly, we observed strong activity for HsHDAC7 (1g) and HsHDAC9 253 (1g) but neither for the class I enzyme HsHDAC1 (2a) nor the class IIb enzyme HsHDAC6 (1g), 254 supporting the validity of the assay and of the clustering (Fig. 2c; Fig. 1a,b; Supplementary Data 1-3)." However, the assay data of HsHDAC6 were not presented in the Fig 2c.

Response:

We agree with reviewer 3 and added the data for HsHDAC6 and updated the Source Data file and the figure legend.

Point 2:

Line 335-337: Original text was as follows "However, we found the peptide LGK_{oct} being most efficiently hydrolyzed by BsAcuC (2c) compared to the other LGK_{acyl} peptides (Fig. 3a)." In Fig 3a, LGK_{oct} was not found. Further explanation should be given to support your conclusion.

Response:

We agree with reviewer 3. This was a mistake, we refer to the peptide QPKK_{oct}. We corrected it in the manuscript and wrote:

"However, we found the peptide QPKK_{oct} being most efficiently hydrolyzed by BsAcuC (2c) compared to all other peptides tested including the the LGK_{acyl} peptides (Fig. 3a)."

Point 3:

Line 344-347: "This agrees well with our data in which we observed for BsAcuC (2c) high deacylase activity for the peptides containing a Gly-Lys (Fig. 3a,d). Notably, the deacylation efficiency for the physiological substrate BsAcsA might be substantially higher as the peptides analyzed here differ in their substrate sequences." From Fig 3a and supplementary Fig 6, it is clear that QPKK_{hex}, QPKK_{oct}, QPKK_{dec} and TARK_{dec} showed similar or better extent activity as LGK. It is difficult to get the conclusion as stated in Line 344-346. Further quantitative assay should be performed. Additionally, there is no data supporting the hypothesis stated in Line 346-347. The biochemical assays should be measured for the native peptide sequence of BsAcsA, if possible.

Response:

We agree with reviewer 3 and altered the text accordingly:

“Our data show BsAcuC (2c) is capable to deacylated Gly-Lys-containing peptides with similar efficiency to peptides containing a positively-charged residue at -1 position (Fig. 3a,d).”

Concerning the conclusion drawn in lines 344-346 we analysed the experimental results which show that BsAcuC is much less efficient in deacylating the substrate peptides compared to highest activities observed for other enzymes. Having said this the lower efficiency observed for BsAcuC in our assay goes along with the highest degree of substrate promiscuity of all enzymes compared. We want to say that BsAcuC might not generally be less efficient compared to the other enzymes tested but might be much more efficient for the natural substrate. We show in Supplementary Fig. 17 that BsAcuC is highly efficiently deacetylating the acetylated full length enzyme AMP-forming acetyl-CoA synthetase BsAcsA. We published a paper on regulation of BsAcsA activity by lysine acetylation in July 2024 in Nature Communications containing data on acetylation and BsAcuC-catalysed deacetylation of BsAcsA (Qin et al. (2024), Nature Communications, 10.1038/s41467-024-49952-0). In *B. subtilis* the deacetylase AcuC is encoded in an operon together with the acetyltransferase BsAcuA and another protein, AcuB, of unknown function. This operon is reversely transcribed downstream from the gene encoding the acetyl-CoA synthetase BsAscA. This support that under physiological conditions BsAcuC might mainly be targeting BsAcsA as a substrate. From our perspective it would be interesting to quantitatively compare the catalytic efficiency of BsAcuC towards different substrates including an BsAcsA peptide but this is beyond the scope of the current study and would not add any essential information nor is it important for support of the major findings of the study.

Point 4:

Line 492-493: The original text is “In bacterial deacylases of cluster 3, LpApaH (3) and LcApaH (3), the first His is replaced by an Asn (Fig. 4b).” However, based on the sequece alignment in Fig 1, the two His residues seem high conserved in bacterial deacylases including clsuter 3. Therefore, the discription is kind of controversial.

Response:

The double-His motif containing the catalytic base (*KpHdaH*: H143/H144 (base)) is totally conserved. However, there is a second double-His motif (*KpHdaH*: H183/H184), in which the first His (*KpHdaH*: H183; *LpApaH/LcApaH*: N221/N218) is indirectly involved in catalysis by orienting and coordinating the catalytic Zn²⁺-ion. Maybe Fig. 4b is misleading as the N186 is highlighted, however, it is not N186 which is replacing any His in *LpApaH/LcApaH* (3). In the second His-motif, the H183 of *KpHdaH* is replaced by N221/N218 *LpApaH/LcApaH* both being functionally conserved.

We added labelling of the N221/N218 of *LpApaH/LcApaH* in Fig. 4b for a better overview and altered the text:

“These enzymes mediate catalysis using a penta-coordinated catalytic Zn²⁺ ion coordinated by two aspartates and a histidine (*KpHdaH* (1b): Asp181, Asp269, His183) (Fig. 4c). In bacterial deacylases of cluster 3, *LpApaH* (3) and *LcApaH* (3), the *KpHdaH* H183 is replaced by an Asn both being functionally conserved (Fig. 4b).”

Point 5:

Line 505: The interacting residues described in text "(KpHdaH: Val189, Tyr192, Arg195, Tyr227) " is different from the residues shown in Fig 4c. Please correct the text or Fig 4c.

Response:

The potassium ion is hexacoordinated by the main chain carbonyl oxygens of Val189, Tyr192, Arg195, Tyr227 and two water molecules. The Tyr192 is not shown in Fig. 4c to enable a better visibility.

We rewrote the text:

"The second potassium ion, K^{+2} , binds distantly from the active site and has a pure structural role (Fig. 4c). It is hexa-coordinated by the main chain carbonyl oxygens of several residues (KpHdaH: Val189, Tyr192 (not shown in Fig. 4c), Arg195, Tyr227) and by two additional water molecules (Fig. 4c)."

Point 6:

1. Line 16: The affiliation corresponding to number 5 has not been linked to any author in the author list. Please check it.

Response:

The reviewer is correct, we corrected it as requested.

Point 7:

2. Supplementary Data should be cited according to the order they described in the main text.

Response:

We corrected the order of citations as requested.

Point 8:

3. Line 244-245: The corresponding references for peptide 2 and peptide 2a should be added.

Response:

The peptides were commercial and the company and order numbers are listed in the Material and Methods section (peptide S2 and S2a (Sigma-Aldrich; cat no. SRP0306, cat no. SRP0303)).

Point 9:

4. In table 1, "VsvHdaH" should be corrected to VsHdaH.

Response:

We corrected the typo as requested.

Point 10:

5. Line 472: In the text "(5b)and by X-ray crystallography", the word "and" should be deleted.

Response:

We deleted the "and" as requested.

Point 11:

6. The abbreviations of amino acids (one letter or three letters) are better to keep consistent in the whole text and figures.

Response:

We use the one-letter code for amino acids in the figures and the three-letter code in the text. We think that creates a better readability of the figures and the text.

Point 12:

7. Fig 5e is a redundant with Fig 1c.

Response:

The alignment in Fig. 1c contains the sequences of the enzymes we studied in the paper and additional sequences of the human enzymes and bacterial enzymes that were studied before. In contrast, Fig. 5e shows an alignment containing only the bacterial enzymes and the known representatives that are used for a direct comparison. That means, these alignments are not the same. In Fig. 5e a closeup is shown in which the important sequence motifs are color-coded.

Point 13:

8. The position of the reference numbers (in front of/behind the punctuation marks) should be consistent within the manuscript.

Reference:

We checked the position of the references and think it is consistent now.

Point 14:

9. In supplementary table 4-10, the PDB entry should be added. Additionally, the crystallization information of KpHdaH (1b) H144A should be added in Supplementary table 4.

Reference:

We added the PDB codes as requested.

Point 15:

10. The description in line 861 is likely linked to Fig. 8d not 8c.

Response:

The reviewer is correct, we altered it as requested.

Point 16:

11. Line 1129: The word “shwos” needs to be corrected.

Response:

We corrected the typo as requested.

Point 17:

12. What kind of base was used for the pH adjustment of “HEPES”? The information needed to be provided in the method section.

Response:

We added to the methods section: “HEPES buffer pH was adjusted with NaOH.”

Point 18:

13. Line 1180: "E. coli DH5 α ells" needs to be corrected.

Response:

We thank the reviewer and corrected the typo.

Point 19:

14. Line 1256: "50 mM Tis/HCl pH 8" needs to be corrected.

Response:

We thank the reviewer and corrected the typo.

Point 20:

15. Line 1326: Supplementary table should be cited correctly.

Response:

We thank the reviewer and corrected the citation.

Reviewer 4:

We thank reviewer 4 for carefully reading our manuscript and giving this constructive feedback on our manuscript. We think all comments improve our manuscript and are valuable. Thank you very much for your work and help.

Point-by-point response to the comments of reviewer #2 (Remarks to the Author):

Reviewer #2 (Remarks to the Author):

The revised version of the work entitled "Distribution and diversity of classical deacylases in bacteria" presents a corrected and improved version of the original manuscript. The authors have correctly addressed all my concerns as well as those raised by the other reviewers. I therefore recommend the manuscript for publication. Specific comments follow:

Point 1:

1) The authors have shortened the manuscript where possible. Specifically, they removed descriptions of eukaryotic classical deacetylases in the introduction and discussion and emphasized the work on the bacterial enzymes.

Response:

We thank the reviewer for appreciating the revisions done in the manuscript.

Point 2:

2) The authors showed that the set of sequences covered by their clustering analysis is virtually identical to the sequence set covered by the PFAM domain PF00850 'Histone deacetylase domain', which is now incorporated into the INTERPRO entry IPR050284 'Histone deacetylase and polyamine deacetylase'. This information should be explicitly mentioned in the manuscript.

Response:

We added the following sentence in the Material and Methods section:

"The set of sequences covered by the clustering analysis is virtually identical to the sequence set covered by the PFAM domain PF00850 'Histone deacetylase domain', which is now incorporated into the INTERPRO entry IPR050284 'Histone deacetylase and polyamine deacetylase'."

Point 3:

3) As requested the have in the revised version prepared sequence logos for the individual clusters, which are shown in the SI section of the revised version.

Response:

We thank the reviewer for appreciating the revisions done in the manuscript.

Point 4:

4) The authors state that the tree is not rooted since no suitable outgroup is available. Therefore I would recommend to remove the root to avoid possible confusion of its biological meaning.

Response:

We thank the reviewer for this comment. We added in the figure legend the information that this is an unrooted phylogenetic tree to avoid any confusion:

"The unrooted phylogenetic tree was created with iTOL using a multiple sequence alignment of the catalytic domains (deleted >90% of gaps) created by MAFFT."

We also added a paragraph in the Material and Methods section:

"The alignment was used to generate a neighbor-joining tree with the program *belvu*¹⁹². This was converted to newick format and the unrooted phylogenetic tree was visualized with iTOL^{193,194}."